# Cross-regional real-time visualization of systemic physiology and dynamics with 3D panoramic photoacoustic computed tomography (3D-PanoPACT)

Xuanhao Wang[1,2], Yuqian Meng[1,2], Mingli Sun[1], Xiali Gao[1,3], Yuqi Wang[1,2], Shaobo Wang [1], Kaiyue Wang[1], Ruofan Wang[1,2], Danyang Ren[1,2], Yonggang Yin[1,2], Chiye Li[1,2], Ruimin Chen[1,2], Lihong V. Wang [4,5] ✉ & Junhui Shi [1,2,3] ✉

Non-invasive systemic visualization is crucial in biomedical research. Photoacoustic imaging transcends the optical scattering limitations, capturing optical absorption from deep tissue at high resolution. Nevertheless, the full exploitation of its physical advantages faces challenges due to unsatisfactory field of view (FOV), spatiotemporal resolution, and imaging quality. Here, we demonstrate that 3D panoramic photoacoustic computed tomography (3D-PanoPACT) mitigates these limitations by combining dynamical and functional contrasts, high spatiotemporal resolution, and broad three-dimensional FOV, allowing for visualizing cross-regional dynamics from organ to whole-body level. Leveraging its single-pulse-volumetric imaging capability, 3D-PanoPACT imaged the whole liver at 25 Hz, facilitating the extraction of arterial networks. Real-time monitoring of whole-brain vasculatures and functional hemodynamics induced by pharmacological action and physical stimulation was demonstrated, especially in the Circle of Willis. The dynamic visualization of the whole trunk at 10 Hz further substantiates its capacity for systemic imaging. We tracked the metabolic pathways of a small-molecule probe across multiple organs using 3D-PanoPACT, highlighting its value in whole-body, cross-regional dynamic studies. 3D-PanoPACT, being a potent novel imaging tool, exhibits substantial potential for advancing biotechnology.

Systemic imaging stands as a pivotal tool in advancing life sciences, offering various modes for comprehensive development[1,2]. Non-invasive three-dimensional (3D) visualization of intrinsic physiology and dynamics allows a macroscopic understanding of organ-level biological mechanisms. However, in-vivo systemic imaging grapples with challenging compromises involving spatiotemporal resolution, limited field of view (FOV) or penetration depth, and acceptable image quality amidst the presence of abundant contrast sources. Presently, the widespread use of magnetic resonance imaging (MRI) allows researchers to obtain quality 3D whole-body images, albeit with

[1]Research Center for Computing Sensing, Zhejiang Lab, Hangzhou, China. [2]State Key Laboratory of Ocean Sensing, Hangzhou, China. [3]College of Biomedical Engineering and Instrument Science, Zhejiang University, Hangzhou, China. [4]Caltech Optical Imaging Laboratory, Andrew and Peggy Cherng Department of Medical Engineering, California Institute of Technology, Pasadena, CA, USA. [5]Caltech Optical Imaging Laboratory, Department of Electrical Engineering, California Institute of Technology, Pasadena, CA, USA. ✉e-mail: LVW@caltech.edu; junhuishi@outlook.com

construction costs and strong magnetic fields over minutes due to physical limitations[3,4]. X-ray computed tomography (X-CT) lacks soft tissue contrast and is unsuitable for dynamic imaging due to low temporal resolution and prolonged radiation exposure[5]. Positron emission tomography (PET) specializes in invasive diagnoses of cardiac-cerebral vascular diseases and cancer screening but suffers from poor spatial resolution[6]. Ultrasonic imaging (USI) non-invasively provides structural images but lacks endogenous molecular contrasts, impairing functional imaging capabilities[7–9]. These deficiencies hinder deeper insights into biological studies. Although optical imaging (OI) offers multifarious functional visualizations, conventional OI faces challenges caused by strong optical scattering in biological tissue, limiting systemic imaging quality beyond the optical scattering limit of 1–2 mm in depth[10]. Overcoming these challenges is crucial for advancing systemic imaging in life sciences.

Photoacoustic (PA) Computed Tomography (PACT), as an emerging imaging modality, stands out for its distinctive ability to visualize optical absorption in deep tissues[11]. Utilizing light pulse illumination and acoustic detection, PA signals carrying information on light absorption propagate from tissues several centimeters deep, creating images with acoustic resolution and optical contrast[12]. This approach holds great promise for whole-body imaging of small vertebrates, with various PACT systems demonstrating their capacity to generate high-fidelity structural images of cross-sections of the trunk or brain[13–15]. The unique combination of high-velocity and multiwavelength imaging abilities in two-dimensional (2D) PACT previously extended its performance to the tracking of metabolism dynamics[16], monitoring physiological parameters, and recording exogenous probes for specialized applications[17–19]. For example, existing studies have already been able to achieve dynamic assessment of metabolic functions in specific organs using 2D-PACT[20]. Despite these advancements, PACT systems designed for 3D imaging face challenges, including low spatiotemporal resolution, confined FOV, and image quality without optimization. In one approach, a PACT system equipped with 128 ultrasonic transducers integrated into a hemispherical surface was introduced[21], however, artifacts persist outside the FOV due to angular sparse sampling. To address these challenges, some researchers scanned a dome-type array with 256 elements in a helical line, successfully obtaining 3D images of the trunk with clear vessels and recording the pharmacokinetics in the relatively shallow tissue[21]. As a further optimization, other researchers adopted the 512-element dome-type array with higher center frequency for raster and angular scanning to achieve whole-trunk imaging[22]. However, the application in functional imaging of this system was limited by slow imaging speed and shallow penetration depth. In addition, from the perspective of invivo imaging quality, the increase of the center frequency of the transducers did not appear to achieve higher spatial resolution as expected. Another significant development involved a 3D PACT system with superior performance achieved by rotating an arc-shaped transducer array at high speed, allowing for a large FOV and a relatively high frame rate[22]. While this design was successfully demonstrated in rat brain and human breast imaging, the system's molecular imaging capability remains unproven, necessitating further exploration and validation through multispectral unmixing techniques[23,24]. These advancements underscore the evolving landscape of PACT, showcasing its potential for transformative contributions to biomedical imaging. Overall, the inherent trade-off between the FOV and spatiotemporal resolution limits the application of PACT in biomedical research, particularly in studying the dynamics of multiple organs and cross-regional systems throughout the body.

In this study, we present a state-of-the-art 3D panoramic photoacoustic computed tomography (3D-PanoPACT), representing a cutting-edge advancement in PACT technology. Leveraging a high-density transducer array arranged according to the Fibonacci grid and rigorous engineering design and calibration, 3D-PanoPACT achieves high-fidelity, high-spatiotemporal-resolution imaging within scalable FOVs, allowing analysis of physiological dynamics from whole-organ to whole-body level. We used 3D-PanoPACT to clearly image the whole liver at a frame rate of 25 Hz, extracted the heartbeat and respiratory signals, and mapped the 3D arterial network and the phase gradient of the liver lobe related to the pulse wave, employing a single wavelength. We further obtained the authentic whole-brain vascular anatomy, with particular emphasis on the vivid depiction of pivotal basicranial arterial structures, the Circle of Willis. Based on this, whole-brain functional dynamics, from the Circle of Willis to the cortex, can be recorded and analyzed at a frame rate of up to 10 Hz. To underscore the 3D-PanoPACT's imaging capacity at whole-body scale, we introduced 10 Hz dynamic reconstruction with high quality of the whole trunk, from the thoracic cavity to the reproductive system, visually presenting the 3D anatomy with elaborate vascular networks as well as rhythmic organ motions in nature, with a penetration depth beyond 20 mm. We further tracked the metabolic pathways of a small-molecule probe across multiple organs with high spatiotemporal resolution using 3D-PanoPACT, thereby demonstrating its significant practical value in cross-regional dynamic studies at the whole-body scale. Therefore, we illustrated the comprehensive imaging performance of 3D-PanoPACT by extending from single-organ to whole-body FOV and from structural to functional imaging. Given that high-speed 3D imaging of the whole brain and whole body are significant research areas, presenting the whole-brain and whole-body imaging results separately helps to highlight the broad applicability of 3D-PanoPACT in addressing diverse imaging needs and its potential for solving key biological questions. For a clear illustration, we further provided a detailed description of the implementation and effectiveness of the spatiotemporal-integration (STINT) method, which is extensively used in this study. Consequently, 3D-PanoPACT not only explores but fully harnesses the superiority of the PA modality, attracting increasing attention from biomedical practitioners.

## Results

### Development and calibration of 3D-PanoPACT

The constructed 3D-PanoPACT system is illustrated in Fig. 1a. The foundational framework of the imaging system comprises four modules: (1) 1024 piezoelectric ultrasonic transducers are integrated into a hemispherical housing with a diameter of 20 cm (Fig. 1a). Each transducer element is non-focusing with a central frequency of 3.16 MHz and a one-way bandwidth of 60% (Fig. S1, see Supplementary Information for details). The transducer elements are arranged according to the Fibonacci grid[25] to form a high-density 3D array with uniform acoustic detection field. Considering the mechanical geometrical deviations, the actual position of each element was calibrated so as to affect the image reconstruction quality (see "Methods"). (2) Four sets of encapsulated 256-channel parallel analog-digital conversion modules are attached to four connectors of the ultrasonic array (Fig. 1a). Each module contains 256-channel pre-amplification circuits and 256-channel data acquisition (DAQ) circuits interlinked with an adapter plate for facilitating interface correspondence. This direct plug-in architecture is conducive to minimizing coupling noise. (3) The illumination light from the lasers is delivered through spatial light paths. One of the Nd:YAG lasers provides fundamental 1064 nm output and the other one equipped with an optical parametric oscillator (OPO) generates 670-900 nm tunable output. The 3D-PanoPACT offers two illumination modes. 1064 nm output reaches a maximum laser pulse repetition rate of 25 Hz when single-wavelength imaging mode is applied (Fig. 1c). The tunable and 1064 nm outputs work in pairs to provide a 10 Hz dual-wavelength imaging mode, whose pulse repetition rate is limited by the damage threshold of the crystal inside OPO (Fig. 1c). The laser pulse interval for each group was set to 20 μs, preventing aliasing of the back-and-forth arriving PA wavefronts. To achieve a 20 μs delay in the output of laser 2 (1064 nm output) relative

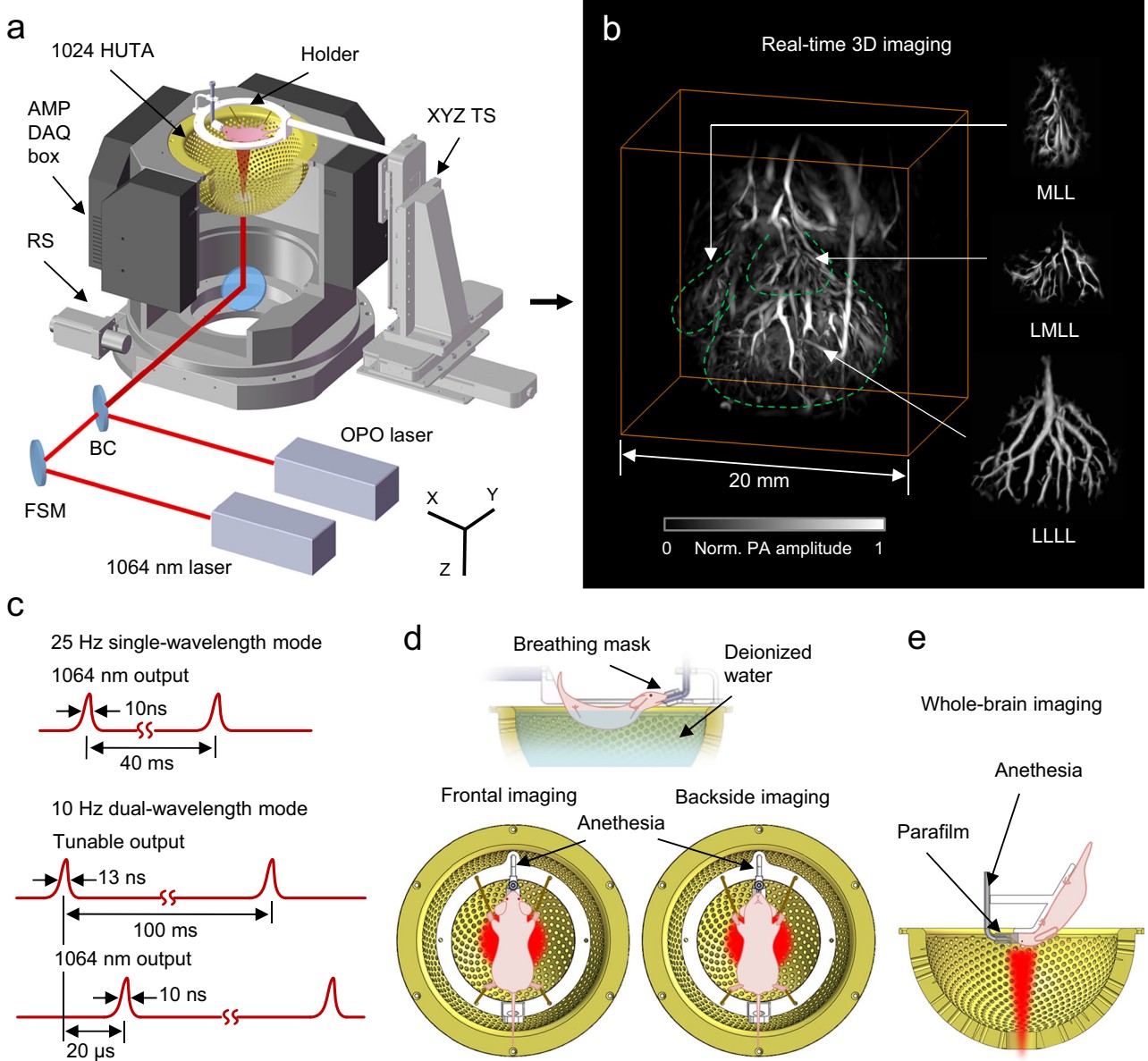

**Fig. 1 | System design and imaging examples of 3D-PanoPACT. a** The schematic diagram of the 3D-PanoPACT system. HUTA hemispherical ultrasonic transducer array, AMP amplification circuits, DAQ data acquisition module, RS rotation stage, BC beam combiner, FSM front-silvered mirror, ED engineered diffuser, OPO Optical Parametric Oscillator, TS translation stage. **b** Real-time 3D imaging of the whole liver anatomy produced by the 3D-PanoPACT with separate displays of the different liver lobes attached. The image was acquired with a single laser pulse. MLL middle liver lobe, LMLL left middle liver lobe, LLLL left lateral liver lobe. **c** The sequence diagram of 25 Hz single-wavelength mode and 10 Hz dual-wavelength mode. **d** The schematic diagrams of in-vivo trunk imaging in 3D-PanoPACT. **e** The schematic diagrams of in-vivo brain imaging in 3D-PanoPACT.

to laser 1 (tunable output), we delayed the flash signal of laser 1 by 50 μs relative to that of laser 2, resulting in a laser output interval of $160 + 50 − 230 = −20$ μs, corresponding to a 20 μs delay. Two outputs are beam-combined and pass through an engineered diffuser sealed on the bottom of the hemispherical housing to obtain a compatible illumination inside the imaging region (Fig. 1a). (4) The customized animal holders accommodate the systemic imaging tasks. For frontal and backside trunk imaging, the anesthetized animals are secured to the holder via limb restraints, and the holder is mounted onto the triaxial translation stage, allowing precise movement of the entire body in space. The animal's trunk is submerged in deionized water, with its head above the surface, connected to a gas anesthesia apparatus to maintain respiration (Fig. 1d). For brain imaging, the animal is placed with the brain region adjusted to the center of the array and, the interface between the mask and nose is sealed by parafilm (Fig. 1e).

Considering that acoustic detection is crucial for imaging quality, the determination of the array element size has undergone special design and validation. We first used the k-Wave toolbox to investigate the relationship between detection sensitivity and transducer element size. We placed an ideal PA point source at the center of FOV with the transducer set to the center frequency and bandwidth used and observed the relationship between the detected PA amplitude and the radius of the transducer element (Fig. S2a). The simulation results indicate that the PA amplitude exhibits a quadratic relationship with the radius (i.e., is linearly proportional to the area) when the radius is small. However, when the radius is large, the simulated values fall below the ideal values, and the quadratic relationship is no longer maintained (Fig. S2b). The critical point where these two relationships intersect is around Radius = 2.5 mm. In other words, designing the transducer element radius to be 2.5 mm can maximize the detection

sensitivity at the highest cost-effectiveness ratio. We further used Field II to simulate the receiving aperture angles for different element radii. The simulations show that while a transducer with a radius of 0.5 mm has a large aperture angle (Fig. S2c), its sensitivity is 25 times lower than that of a transducer with a 2.5 mm radius (Fig. S2b), which is not conducive to deep tissue imaging. For comparison, the transducer with a radius of 1.5 mm has an aperture angle comparable to that of the transducer with a 2.5 mm radius, but its sensitivity is 2.78 times weaker. After evaluating both detection sensitivity and receiving aperture angle, we determine that a transducer element radius of 2.5 mm (i.e., 5 mm diameter) offers the optimal balance for 3D-PanoPACT.

In 3D-PanoPACT, the deviation of the detection coordinates is influenced by both the element position deviations of the hemispherical array itself and the deviations of the rotation axis, which can affect the quality of image reconstruction[26,27]. To calibrate the spatial position of each element, we used a point absorber (50 μm diameter black polystyrene microspheres) embedded in an agarose phantom. The array housing, machined with high precision for an exact 20 cm diameter and ideal Fibonacci grid distribution, still introduced radial deviations due to individual element insertion, necessitating calibration. The calibration procedure involved three steps (see "Methods" for details). First, we centered the FOV at the origin and adjusted the point absorber's position while repeatedly imaging it at 690 nm until it was centered in the field with PA raw data acquired (Fig. S3a). Second, we measured the water temperature with a high-precision thermocouple and calculated the speed of sound in the medium using the classic temperature-velocity formula[28]. Third, we positioned the signals from the transducer surface and the point absorber in the raw data, identified the sample points where the PA signals peaked (Fig. S3b), and calculated the radial deviation for each element using the travel time and acoustic velocity[29] (see "Methods" for details). The corrected positions of the elements were then determined in Cartesian coordinate with the radial deviations kept within ±0.5 mm (Fig. S3c). We then calibrated the system rotation axis due to assembly errors. To correct this, we positioned a point absorber 15 mm from the origin and imaged it while rotating the array by 70°. By co-reconstructing 15 evenly-spaced frames, we formed a 3D image where the center of the reconstructed points clearly defined a circle (Fig. S3d). The normal through the center was identified as the calibrated rotation axis of the system, allowing us to correct the misalignment and improve image quality. The calibration revealed that the actual rotation axis was parallel to the ideal axis but shifted by 0.07 mm along the X-axis and 0.29 mm along the Y-axis (Fig. S3d). Applying these calibrations, we observed improvements in resolution and contrast, particularly in regions away from the center of the FOV (Fig. S3e, f).

To enhance the quantitative accuracy of the PA images, we addressed the issue of limited bandwidth in the transducers and subsequent circuits, which affects the fidelity of the acquired signals (see "Methods"). We employed the edge-emission method to measure the system's electrical impulse response (EIR)[30,31], using a 10 μm thick polyimide film cut into a 2 × 15 mm rectangle as the PA signal source (Fig. S4a, b). This method produced a pair of negative and positive unipolar PA pulses at the edges of the film, which were used to determine the system EIR (Fig. S4c). To demonstrate the effectiveness of EIR deconvolution, we imaged three square polyimide films with side lengths of 3 mm, 6 mm, and 9 mm. The deconvolution process significantly improved the reconstruction of low-frequency components (Fig. S4d, e), which are crucial for accurate unmixing of oxyhemoglobin saturation ($sO_2$) and visualizing neurovascular coupling responses in the cortex.

In PA imaging, accurate optical fluence compensation is crucial for enhancing image fidelity and ensuring reliable quantitative analysis[32], especially for multi-wavelength unmixing. We employed empirical testing to estimate optical attenuation in freshly excised tissue for subsequent compensation. A simple setup was first constructed using two quartz plates, a collimated laser beam, a 1:9 beam splitter, and a pair of optical energy meters. One optical energy meter was used to detect the light passing through the quartz plates, while the other was used to monitor the energy fluctuations of the split beam as a normalization reference (Fig. S5). By comparing these measurements with and without a sample, we calculated the optical attenuation coefficient ($\mu_{eff}$) for brain and liver tissues at different wavelengths based on the Lambert-Beer law[33], which relates optical attenuation to the properties of the material (see "Methods" for details). For brain tissue, $\mu_{eff}$ was 0.065 mm$^{-1}$ at 1064 nm and 0.077 mm$^{-1}$ at 800 nm, which were utilized for imaging whole-brain anatomies and dynamic functions. For liver tissue, $\mu_{eff}$ was 0.075 mm$^{-1}$ at 1064 nm and 0.133 mm$^{-1}$ at 690 nm, which were applied in dynamic liver imaging and whole-body imaging. These coefficients were then used in image processing to apply exponential depth compensation, enhancing the PA amplitude from deep tissues.

The spatial resolution of the 3D-PanoPACT system was tested with a 10-μm-diameter tungsten wire, which was imaged at various radial positions along the Z-axis (0 mm, 5 mm, 10 mm, and 15 mm) to quantify the variation in spatial resolution across different regions in the FOV (Fig. S6a–d). Gaussian fitting of the profiles of the X-Z and X-Y sections was performed, and the full width at half maximum (FWHM) values were used to characterize lateral and axial resolutions (Fig. S6e, f). It was observed that, as moving radially from the center, the lateral resolution degraded from 178.6 to 312.3 μm, while the axial resolution remained stable at approximately 250 μm (Fig. S6g). This phenomenon was attributed to the angular sampling in the transverse plane becoming sparser with increasing radial distance. On the other hand, within a radial distance of 15 mm (corresponding to an FOV of 30 mm diameter), the resolution could be regarded as near-isotropic, thereby benefitting the imaging quality across a broad FOV. In addition, the high-density acoustic detection enables an ample volumetric imaging field captured by each single laser excitation, empowering the 3D-PanoPACT with the ability to perform real-time volumetric imaging for living animals. By utilizing the weak scattering properties of 1064 nm laser output in tissues, 3D-PanoPACT generated high-fidelity whole-liver vascular network at 25 Hz within the effective FOV size of ~6 cm$^3$ and an imaging depth of ~15 mm (Fig. 1b). For ease of observation, individual lobes of the liver are separately displayed, showcasing rich vascular morphology and details (Fig. 1b). The FOV and the imaging depth of the 3D-PanoPACT are further extended through the spatiotemporal-integration (STINT) method, enabling dynamic visualization of deep tissue ranging from organ to whole-body scales with high fidelity.

## Real-time visualization of liver dynamics in 3D-PanoPACT

The 3D-PanoPACT possesses considerable single-pulse 3D imaging capabilities within a certain FOV. Its imaging frame rate depends on the laser repetition rate when single-pulse imaging is applied, enabling real-time 3D imaging. Here, we demonstrate that 3D-PanoPACT was capable of achieving real-time 3D imaging of the liver non-invasively, enabling the study of rapid hemodynamic processes at the organ level, without exogenous contrast agents (Movie S1, see Supplementary Information for details). At an imaging frame rate of 25 Hz (the 1064 nm laser repetition rate was set to 25 Hz), respiratory motions and heartbeats were fully captured under 1064 nm excitation at well above the Nyquist sampling rate. By placing the left lateral lobe, which is the largest lobe of the liver, at the center of the imaging FOV of the hemispherical array, obvious displacement can be observed during the expansion and contraction of the thoracic cavity (Fig. 2a). By recording the signal changes inside the selected voxel of the vessel (the blue arrow in Fig. 2a), in addition to the low-frequency signal changes caused by respiratory motion, the high-frequency heartbeat signal was also captured by the 3D-PanoPACT (Fig. 2b). Fourier analysis showed that two main frequency components existed in the recorded signal of

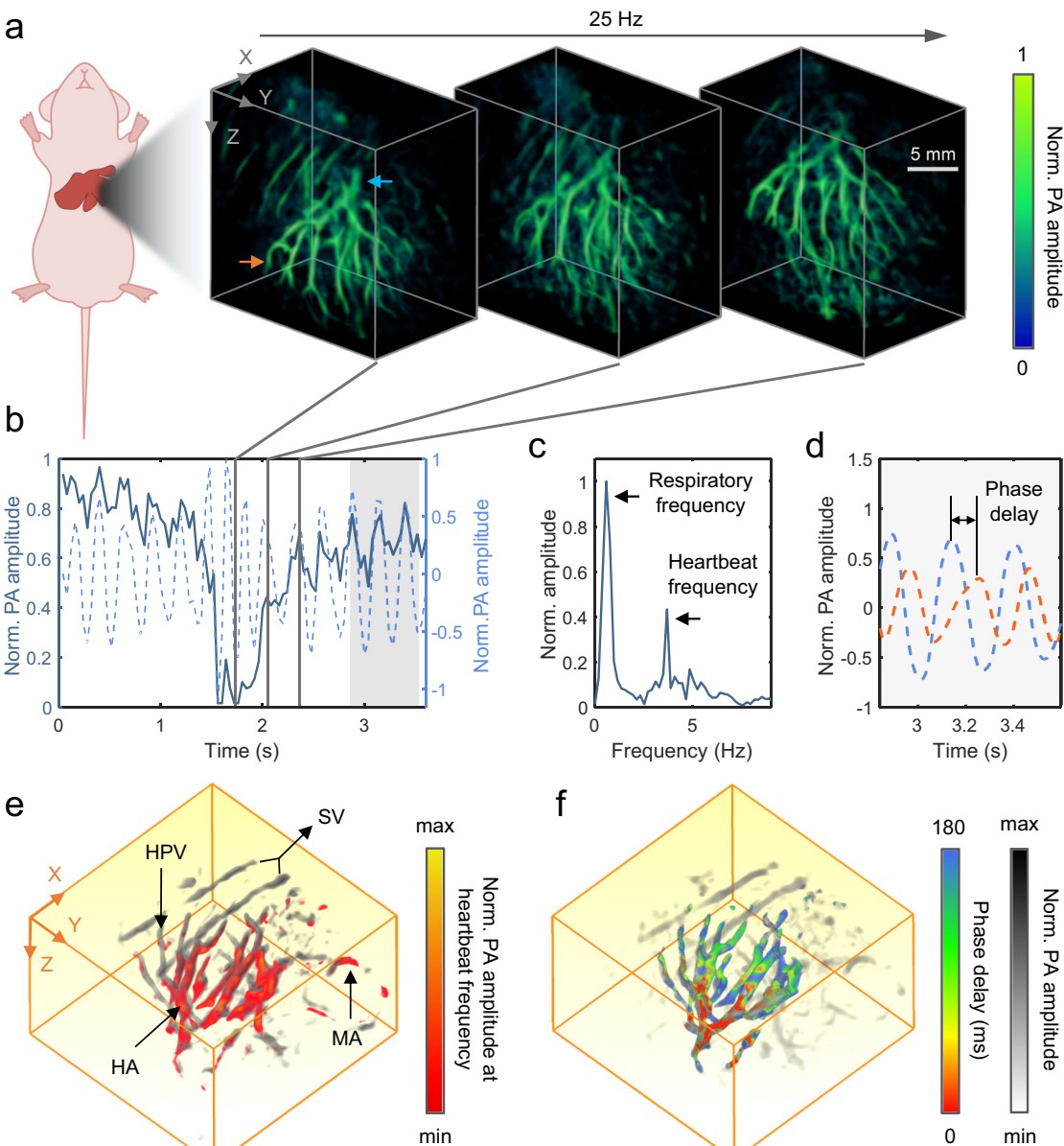

**Fig. 2 | Real-time visualization of liver dynamics in 3D-PanoPACT. a** The 1064 nm real-time 3D imaging of the liver at 25 Hz. The images are taken at different points in a respiratory cycle, which was labeled in (**b**) with gray lines. **b** The raw signal of PA amplitude change (dark blue solid line) of the blood vessel (blue arrow in a) and the extracted heartbeat signal (light blue dashed line). **c** The Fourier transform of the raw data showing the respiratory frequency and heartbeat frequency, respectively. **d** The comparison of heartbeat signals extracted from blood vessels at different locations (the blue and orange arrows in a) with the phase delay clearly visible. The corresponding time windows are indicated by shaded areas in (**b**). **e** Heartbeat-encoded arterial network mapping overlaid on the anatomy background in gray scale. **f** Mapping of pulse wave phase delay overlaid on the gray-scale anatomy background. SV superficial blood vessels, HPV hepatic portal vein, HA hepatic artery, MA mesenteric artery.

the selected voxel, corresponding to the heartbeat frequency (around 3.9 Hz) and respiratory frequency (around 0.8 Hz), respectively. By applying a finite-impulse-response high-pass filter on the recorded signal, the heartbeat component can be extracted while keeping the phase constant, which was shown as a quasi-sine wave (dash line in Fig. 2b).

Additionally, we chose another vessel voxel that was far away from the previous location (the orange arrow in Fig. 2a) and performed a similar analysis on the recorded signal. By comparing the heartbeat component waves obtained from the two locations, a stable phase delay was revealed in the zoomed-in panel (Fig. 2d). In one heartbeat cycle, the contraction of the ventricles leads to the rapid influx of blood into the aorta, causing the vessel walls to contract, subsequently

generating pressure waves that propagate throughout the arterial network, which may be an explanation for the phase delay between two locations. With the imaging sequence, voxel-wise extraction of the amplitude at heartbeat frequency was performed (see "Methods") and the arterial network can be mapped on the 3D liver anatomy (Fig. 2e). If equipped with a higher-repetition-rate pulsed laser, the 3D-PanoPACT could potentially capture cardiac substructure signals and extract information using more complex filters, as these signals often require a higher sampling frame rate for accurate analysis[34]. It is intuitively seen that, although the superficial vessels have stronger amplitude, they cannot be identified as arteries by this method, which is consistent with the fact that the superficial vessels are almost entirely composed of venous branches. On the contrary, the hepatic artery and the

mesenteric artery are highlighted by heartbeat encoding. Further, by extracting the phase of each voxel's time sequences at the cardiac frequency component, we are able to map the 3D phase gradient of hepatic vasculature (Fig. 2f) (see "Methods"), which may be related to the spatial gradient of blood pressure[35]. This unique mapping relies on the high-quality video-rate 3D imaging enabled by 3D-PanoPACT, which may serve as a reference for assessing overall hepatic hemodynamics or as an indicator for certain liver diseases, such as acute liver injury[36,37]. Benefiting from the imaging capability and high spatiotemporal resolution of 3D-PanoPACT, we employed single-wavelength imaging to non-invasively map the 3D arterial networks and quantify the relative phase difference of pulse waves between arteries within an FOV covering the entire organ. This example demonstrates the utility of the 3D-PanoPACT in the label-free imaging and diagnosis of chronic cardiovascular diseases and liver lesions.

## Cerebrovascular network anatomy and real-time functional dynamics in 3D-PanoPACT

Whole-brain imaging has become an indispensable approach for conducting in-depth research in neuroscience[38], and 3D-PanoPACT proves to be competent in providing high-quality whole-brain anatomy and dynamic functional images non-invasively. To minimize acoustic loss and wavefront distortion, we removed a portion of the parietal bone with a diameter of ~8 mm and sutured the scalp without adding replacement. During the modeling process, to avoid overpressure, we carefully removed only the skull without causing any damage to the dura mater and employed low-dose mannitol to further alleviate brain swelling that may be induced by static overpressure (see "Methods")[39,40]. The reliability of this modeling method for short-term brain imaging studies has been confirmed through structural imaging and pathological sections of the control groups[41]. A 10-Hz dual-wavelength illumination of 800 nm and 1064 nm was adopted to reveal more abundant cerebrovascular structures and obtain a larger penetration depth. To attain high spatiotemporal resolution within whole-brain FOV, we employed a method termed STINT, which involves the continuous and rapid rotation of the transducer array at small angular increments to synthesize an equivalent high-density detection array (refer to the final part of the Results section for details). With this, a comprehensive 3D representation of the whole brain anatomy was reconstructed, showcasing distinct main vessels and an abundance of branches (Movie S4). For visual clarity, the entire cerebrovascular network was categorized into cortical and deep-brain regions with multiple perspectives provided (Fig. 3a). Subsequently, a cut-open view of the deep-brain-region vessel network was depth-encoded along the Y-axis (Fig. 3b), and the image of each region was depth-encoded along the Z-axis (Fig. 3c, d), revealing intricate details with the names of vascular structures marked[39,40]. More strikingly, the 3D-PanoPACT provided clear observation of the Circle of Willis, which was representative basicranial vessels and acted as a crucial hub for ensuring blood supply and balancing blood flow throughout the entire brain. The arterial vascular composition (Fig. 3e) and the 3D location (Fig. 3f) of the Circle of Willis were distinctly observable at the base of the brain[42,43], an application unattainable with other optical imaging modalities in living animals. The imaging depth of the whole brain in the real sense can be illustrated more intuitively by displaying different coronal sections from imaging results (Fig. 3g). These imaging results underscore the capability of 3D-PanoPACT in imaging the deep-brain anatomy, providing a high-quality structural reference for functional imaging.

Furthermore, to demonstrate the superiority of imaging functional dynamics, the 3D-PanoPACT was applied in monitoring real-time hemodynamic response introduced by sodium nitroprusside (SNP) administration in the whole brain (Movie S2). As a commonly-used clinical antihypertensive hypotensor, SNP dilates the systemic vascular wall to alleviate the cardiac load[44,45]. In this experiment, a 10 Hz laser output (800 nm and 1064 nm) was adopted throughout the imaging process for unmixing operation (see "Methods"). To improve the signal-to-noise ratio (SNR) of deep-brain signals for high-precision unmixing results, the STINT method was adopted and the dual-wavelength image sequence at 0.5 Hz was produced, which fully met the requirements for capturing non-ultrafast changes in real time (refer to the final part of the Results section for details). Given that low-frequency information often accompanies functional responses such as cerebral blood oxygenation changes, we adopted the precisely measured EIR and applied deconvolution to the signals, enabling more accurate reconstruction of PA image amplitude (see "Methods"). A fixed dose of SNP was delivered from the caudal vein with a retention syringe over 180 s, and the whole brain was constantly monitored for 20 min. The estimation of sO₂, as a pivotal physiological indicator, were demonstrated from the cortex to the Circle of Willis and displayed in parts acquired at 1 s, 290 s, and 1000 s after SNP injection (corresponding to Fig. 4a, b, and c respectively), to facilitate observation. To highlight the advantages of imaging deep-brain functional dynamics, two parameters, the sO₂ and the total hemoglobin (HbT), were calculated in parallel for the Circle of Willis. In the whole-brain view, the average sO₂ change for each major vessel is respectively plotted in Fig. 4d. Veins exhibited varying degrees of sO₂ decrease, reaching a nadir around 300 s after SNP injection. The caudal rhinal vein (CRV) showed the maximum percentage decrease, approximately 40%, with sO₂ values recovering to normal conditions in ~200 s. The reduction in cerebral sO₂ resulted from an increase in oxygen extraction fraction caused by a simultaneous reduction in capillary perfusion[46,47]. In contrast, sO₂ was maintained at almost 100% in the anterior cerebral artery (ACA), azygos of the anterior cerebral artery (AACA), and middle cerebral artery (MCA) due to their direct connection to the carotid artery. For the monitoring of the hemodynamics in the Circle of Willis, the ACA consistently maintained a high sO₂ value, accompanied by a more pronounced change in HbT value (Fig. 4e). This could be attributed to the greater elasticity of the arterial wall. In contrast, the posterior communicating artery (PCoA) exhibited a relatively steep decline in sO₂ values, while the HbT displayed a more gradual alteration (Fig. 4e). This observation suggested a potential role of the Circle of Willis in balancing and coordinating blood supply between the anterior and posterior parts of the whole brain. Therefore, the capability of imaging whole-brain functional dynamics in 3D-PanoPACT was underscored. Additionally, upon discontinuation of the anesthetic input, we monitored changes in sO₂ in the major cerebral vessels during the recovery process to normal conditions (Fig. S7a–c), utilizing similar system setup and imaging methods. As illustrated, venous sO₂ values gradually rebounded from hypoxia due to the progressively reawakening neural activities throughout the body, while the AACA consistently maintained a hyperoxic state, supporting fundamental cerebral activities (Fig. S7d). The inverse correlation between the average gradient of each sO₂ curve and the physiological significance of the corresponding brain region may provide insights into the functional importance of the vessel's location. In general, the 3D-PanoPACT demonstrates competence in noninvasively elucidating vessel-specific hemodynamic mechanisms throughout the entire brain, owing to its deep penetration and high spatial resolution.

To further showcase the capabilities of capturing high-velocity functional dynamics, the 3D-PanoPACT visualized rapid hemodynamic changes in the responsive brain regions induced by electrical stimulation (Movie S3). Referring to the principle of neurovascular coupling[48–50], neural activities can be reflected via changes in blood oxygen, which was previously applied in blood-oxygen-level-dependent functional magnetic resonance imaging (BOLD-fMRI)[51,52]. During the experiment, a sequence of electrical stimuli with a 10-s duration was applied onto the left and right forelimb respectively (see "Methods"). The 10 Hz dual-wavelength imaging using 800 nm and 1064 nm was carried out to record the whole-brain sO₂ change and the

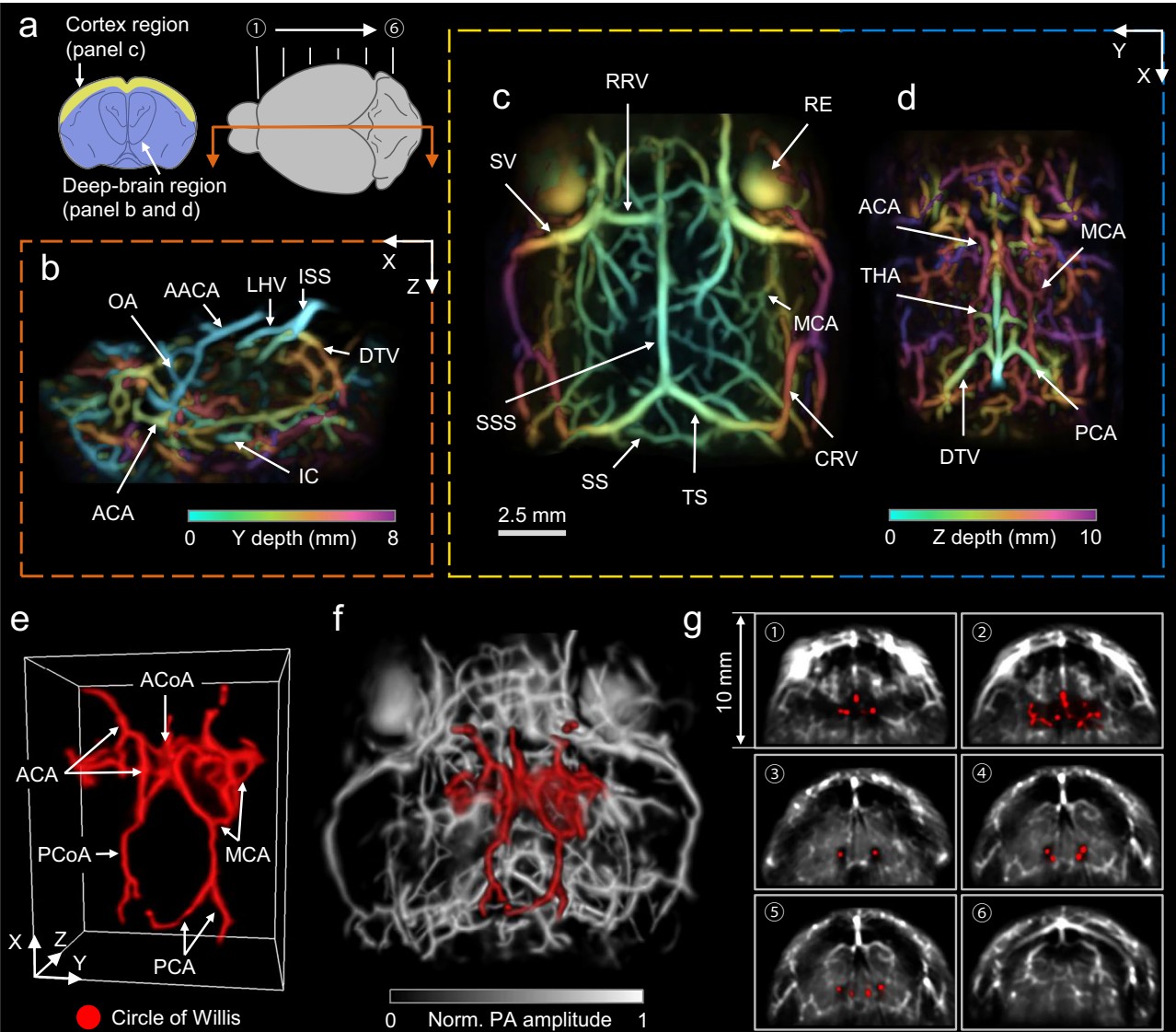

**Fig. 3 | Label-free whole-brain anatomy in 3D-PanoPACT. a** Schematic diagrams illustrating regional segmentation of whole-brain imaging results from various observation perspectives. **b** The cut-open view of the deep brain with the cutting position and view orientation labeled in **a** (orange arrow). The image was depth-encoded along the Y-axis with the cutting position set to zero for better visualization. **c** The image of the cortical region depth-encoded along the Z-axis (yellow region in **a**). **d** The image of the deep-brain region depth-encoded along the Z-axis (blue region in **a**). **e** The 3D view of the Circle of Willis (displayed in red for emphasis) along with its constituent vascular structures. **f** The in situ location of Circle of Willis from the perspective of the base of the brain. **g** Six coronal sections from the image (labeled in **a**) showing the imaging depth in 3D-PanoPACT. Each cross-section was obtained by averaging a 1 mm thick 3D image in the thickness direction. OA olfactory artery, AACA azygos of the anterior cerebral artery, LHV longitudinal hippocampal vein, ISS inferior sagittal sinus, DTV dorsal thalamic vein, ACA anterior cerebral artery, IC internal carotid, RRV rostral rhinal vein, SV supraorbital vein, SSS superior sagittal sinus, SS sigmoid sinus, TS transverse sinus, CRV caudal rhinal vein, MCA middle cerebral artery, RE right eye, THA transverse hippocampal arteries, PCA posterior cerebral artery, ACoA anterior communicating artery, PCoA posterior communicating artery.

monitoring lasted for 30 s. To pinpoint the responsive brain regions, we first produced the convolved hemodynamic response function (CHRF) by convolving stimulation sequence with the standard hemodynamic response function (HRF)[53,54] (Fig. 4h). Then the responsive regions were extracted through voxel-wise $sO_2$ sequence correlations (see "Methods"). It can be observed that a pronounced $sO_2$ change happened in the contralateral somatosensory cortex when certain stimuli were carried out (Fig. 4f, g). Discriminately, a faintish response existed in the bilateral motor cortex when the right forelimb was provoked, which may be a result of limb movement under mild anesthesia (Fig. 4f, g). Averaged $sO_2$ change curve (%baseline) from six regions (labeled in Fig. 4f, g) were plotted, and notably, the change in somatosensory cortex peaked lagging behind 5–10 s after the stimuli stopped (Fig. 4i-k). Moreover, a slight $sO_2$ decrease, known as the

initial dip[55,56], happened at the onset of the stimulation and recovered within seconds, which was theoretically supported by dynamics of neurovascular coupling and was previously hard to observe in PACT systems (Fig. 4j, k). This indicates the 3D-PanoPACT is adequate for capturing rapid physiological changes taking advantage of the high imaging velocity and sensitivity. By employing high-repetition-rate lasers, the frame rate of 3D imaging can be further augmented, facilitating the capture of more rapid activities, such as the firing of nerve signals.

## Label-free visualization of whole-trunk dynamics in 3D-PanoPACT

In preclinical studies, the imaging of the whole trunk, which houses the majority of vital organs, holds significant importance. In addition to

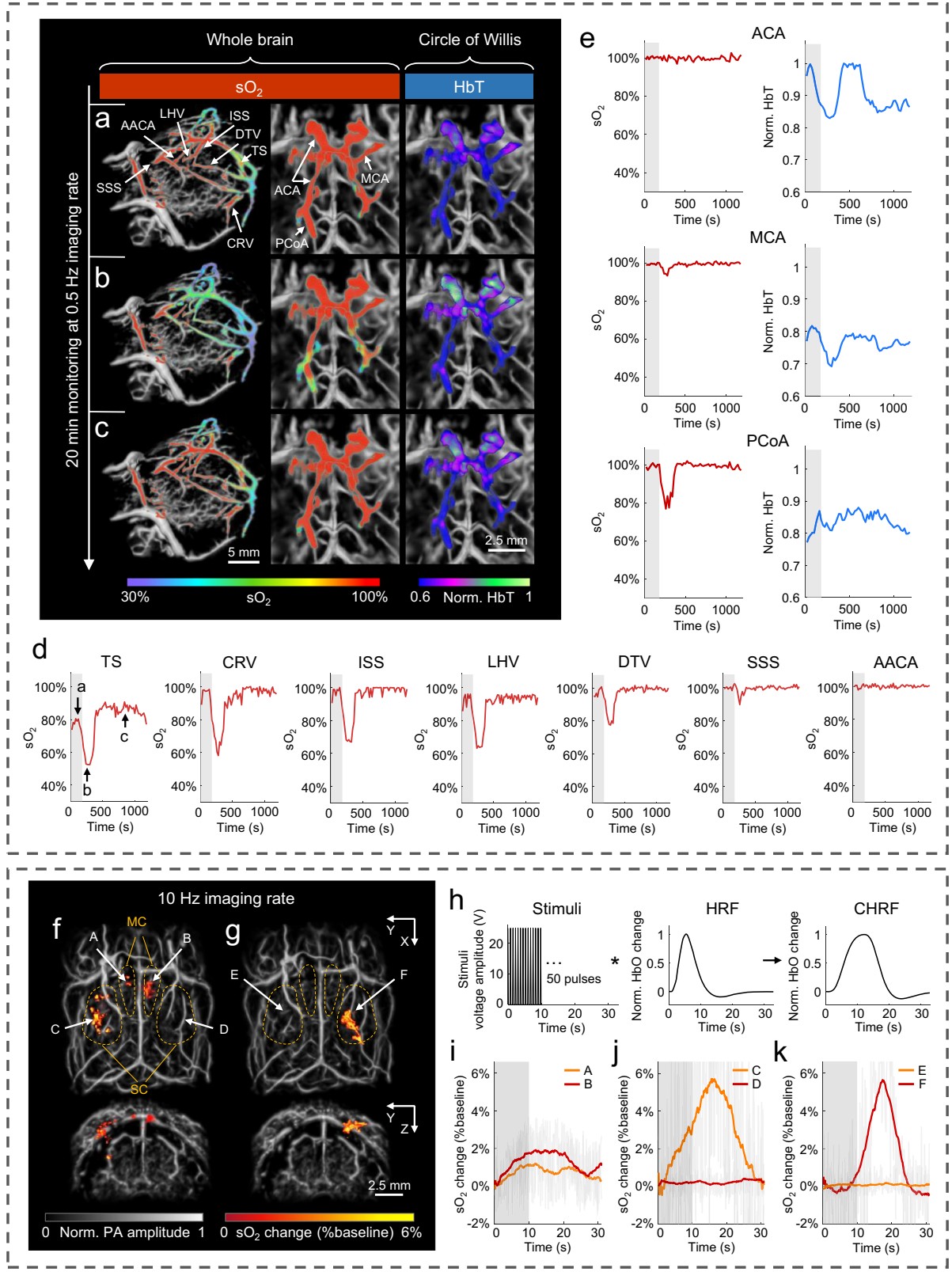

cross-regional whole-organ-level imaging, the 3D-PanoPACT can achieve high-quality dynamic imaging of the whole trunk non-invasively, providing a comprehensive view of in-vivo physiological research. Two imaging perspectives of the front and back were provided here with the experimental setup illustrated in Fig. 1d. During the imaging process, 690 nm and 1064 nm dual-wavelength laser outputs at 10 Hz were used to provide component information from more

diverse optical absorptions. To preserve high resolution while achieving a broad FOV, the STINT method was applied through small-angle rotation of the hemispherical array combined with a two-dimensional translation of the animal holder (Fig. 5a) within approximately 15 s (refer to the final part of the Results section for details). Through the STINT method, dual-wavelength dynamic whole-trunk images can be reconstructed (Fig. 5a) and, a 10-Hz-frame-rate 3D image

**Fig. 4 | Real-time monitoring of whole-brain functional dynamics in 3D-PanoPACT. a–e** The 0.5-Hz-frame-rate imaging of cerebral hemodynamic response to sodium nitroprusside (SNP) administration. **a–c** The first column: the oxyhemoglobin saturation (sO$_2$) map in the whole brain in response to SNP administration at 1 s, 290 s, and 1000 s (labeled in **d**) after SNP injection superimposed on the anatomy background; the second and third columns: the sO$_2$ and total hemoglobin (HbT) maps of the Circle of Willis at the corresponding time point. **d** The sO$_2$ change in whole-brain vascular (labeled in **a**) over 20 min following SNP administration (the administration time window is indicated by the shaded gray area). **e** The sO$_2$ and HbT change in the Circle of Willis (labeled in **a**) over 20 min following SNP administration. TS transverse sinus, CRV caudal rhinal vein, ISS inferior sagittal sinus, LHV longitudinal hippocampal vein, DTV dorsal thalamic vein, SSS superior

sagittal sinus, AACA azygos of the anterior cerebral artery, SS sigmoid sinus, ACA anterior cerebral artery, MCA middle cerebral artery, PCoA posterior communicating artery. **f–k** The 10-Hz-frame-rate imaging of whole-brain neurovascular coupling response under external electrical stimulation. **f, g** The map of sO$_2$ change (%baseline) introduced by right and left forelimb stimulation, respectively. The top view and rear view of the responsive brain regions are displayed. MC motor cortex, SC sensory cortex. **h** The schematic curve of the electrical stimuli, the hemodynamic response function (HRF), and the convoluted hemodynamic response function (CHRF) used for extracting the responsive regions. **i–k** The averaged sO$_2$ changing curve in the region A-F labeled in (**f** and **g**), correspondingly. The time window for applying electrical stimulation is indicated by the shaded gray area.

for the whole-trunk with rich structural information was obtained by superimposing the two images.

The 3D-PanoPACT visualized clear and spectacular 3D vasculature network distribution and organ morphology throughout the whole trunk. The single-frame imaging results, from the front (Fig. 5b) and back (Fig. 5d) perspectives, were displayed separately using maximum intensity projection (MIP) and depth encoding methods. Melanin and deoxyhemoglobin dominate 690 nm optical absorption while adipose and oxyhemoglobin have stronger PA signals at 1064 nm, which leads to a striking difference in the dual-wavelength images (Fig. S8). From the anatomical features, images at 690 nm provided strong superficial vascular signals and organic contours, while images at 1064 nm revealed deeper structures due to the weaker scattering in biological tissues. From the frontal view, the major organs and their accessory vascular networks were clearly visible. The digestive system was observed with block-like morphology, benefiting from the applicable central frequency of ultrasonic transducers. More noticeably, the entire male reproductive system and urinary system were imaged in detail, showing crisscrossing tubular structures (Fig. 5b, Movie S4). From the back view, a spleen and two kidneys filled with vascular circulations were visualized (Fig. 5d, Movie S4). In 3D-PanoPACT, the movement of the trunk viscera in a living mouse was visualized intuitively at a 10 Hz frame rate (Movie S5). Respiratory motions were captured while other organs were squeezed and displaced during each respiratory motion cycle. To show the imaging depth and the dynamic imaging properties of the 3D-PanoPACT more directly, three cross-sections were selected from the reconstruction results of the two views, respectively, and their dynamic changes over time during one respiratory cycle were shown (Fig. 5c, e). The imaging depth reached at least 20 mm, with deep tissue exhibiting a relatively high SNR. This also suggested that the signal in MIP did not originate from superficial blood vessels, but rather from deep tissues (Fig. 5b, d). The results indicate that 3D-PanoPACT is a powerful tool to reveal systemic physiological dynamics and study multi-organ synergistic mechanisms with near-isotropic resolution, broad FOV, and high imaging speed.

### High-spatiotemporal-resolution tracking of small molecule metabolic pathways at whole-body scale

To further illustrate the practical value of 3D-PanoPACT in biological applications, we tracked a small molecule probe, A1094, through its metabolic pathways in the whole body, allowing us to capture the intricate changes and developments occurring across different organs over an extended period. Tracking probes at the whole-body scale necessitates that the imaging system concurrently exhibits high spatiotemporal resolution and an extensive imaging FOV. From the perspective of signal excitation, an engineered diffuser with a divergence angle of 60° was installed at the bottom of the array. This setup enables uniform illumination within an approximate 12 cm diameter range at the center of the array, thereby allowing a single laser pulse to simultaneously excite the entire mouse body and collect PA signals.

Considering the inverse square relationship between energy density and spot diameter, the laser operated at its maximum output energy (single-pulse energy of 500 mJ @1064 nm) to ensure a high SNR in the raw data, which also limited it to operate at a repetition rate of 10 Hz to prevent damage. For acoustic detection, we adopted a strategy similar to that described in the aforementioned brain functional imaging. High-quality whole-body images can be obtained by simply rotating the array through a small angle, without the need for additional translational movement (Fig. 6a). In each whole-body imaging session, the transducer array rotated 15° around the central axis within 5 s, acquiring 50 frames of raw PA data, which were then reconstructed into a 3D image using the STINT method (refer to the final part of the Results section for details). The array subsequently rotates in the opposite direction within 5 s to complete the next imaging session. Utilizing small-angle reciprocating rotation of the array and the STINT method, 3D-PanoPACT achieved high temporal resolution by completing 12 whole-body imaging sessions within 1 min (equivalent to a frame rate of 0.2 Hz), thus fulfilling a broad spectrum of biological application requirements. It should be emphasized that to minimize the stochastic vibrations induced by mechanical rotation, the rotational velocity of the array was deliberately reduced. Consequently, this approach alleviated the spatial resolution degradation at the peripheral regions of the extensive FOV during STINT reconstruction. The exceptional quality of the whole-body 3D imaging with clear vascular networks and organ morphologies corroborated the efficacy of this method (Fig. 6b). Upon comparison, it was observed that the imaging results achieved high spatial resolution and were largely consistent with those in Fig. 5b across most of the trunk region, with resolution degradation only occurring at the peripheral edges (such as the limbs, neck, and tail), which was attributed to the spatial impulse response of the ultrasound transducer (Fig. 6c). Overall, by utilizing wide-field illumination and the Fibonacci grid arrangement of the array elements, 3D-PanoPACT achieves high spatiotemporal resolution imaging at whole-body scale with only a small angular rotation of the detection array. Concurrently, the streamlined scanning approach and reconstruction methods enable long-duration continuous imaging and batch data processing, making it highly suitable for biological applications involving the tracking of small molecule probes.

In this study, we selected a small molecule probe, A1094, as the tracer to visualize its metabolic pathways throughout the whole body. A1094 is a stable and rapidly cleared NIR-II imaging agent that has shown potential for clinical translation due to its sufficient blood solubility, good interference resistance, and pH/oxidative/metabolic stability[57]. In addition, its absorption peak is close to the fundamental output wavelength of the Nd:YAG laser (i.e., 1064 nm), making it suitable for deep-tissue PA imaging (Fig. 6c). Therefore, the imaging wavelength was selected to be 1064 nm in this application (Fig. 6d). During the experimental procedure, a syringe pump was utilized to administer 0.4 ml prepared A1094 solution into the animal via a tail vein indwelling needle within 90 s (refer to "Methods" section for details). The onset of injection was designated as the zero point of

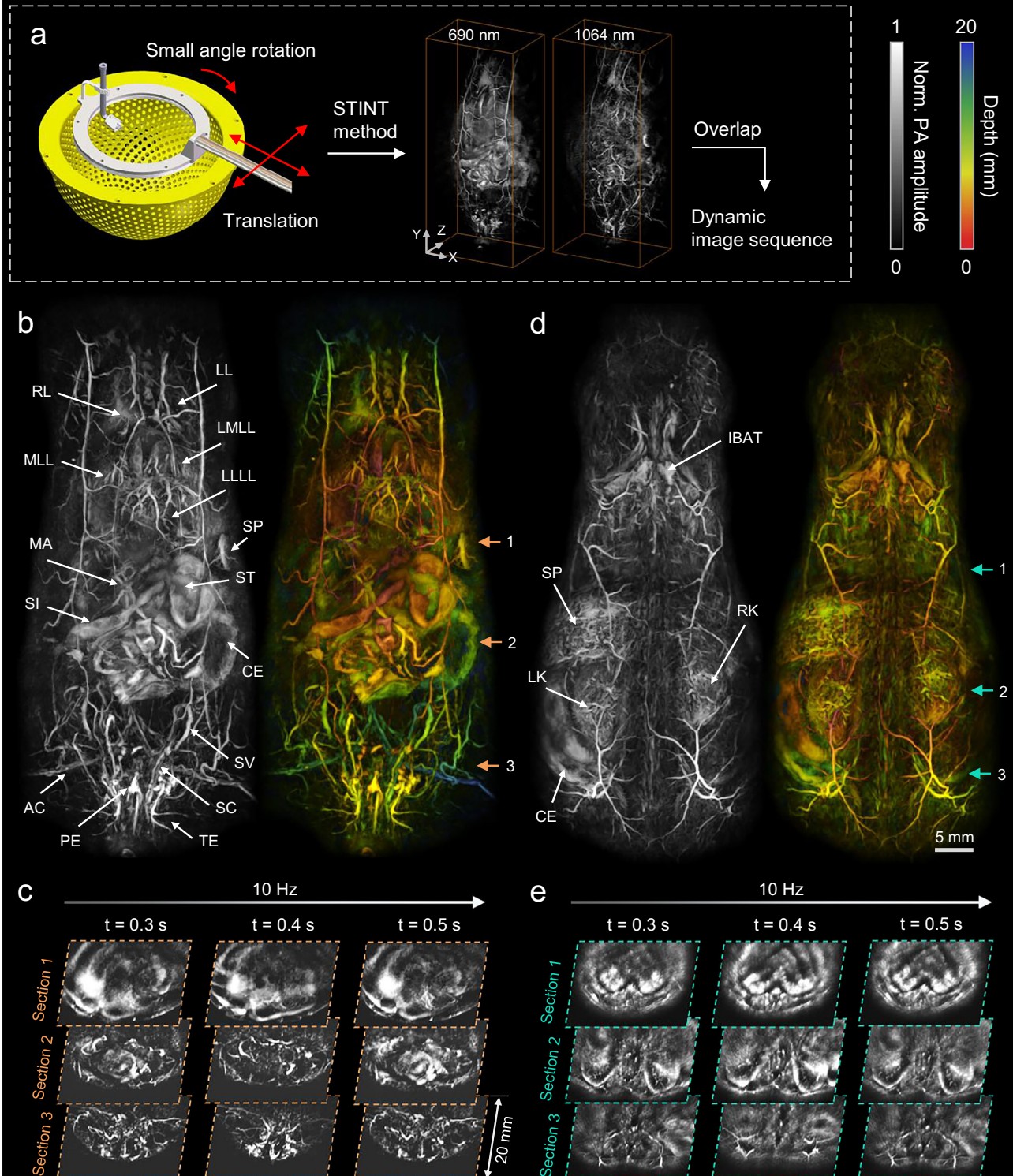

**Fig. 5 | Label-free visualization of whole-trunk dynamics in 3D-PanoPACT. a** The steps to reconstruct dynamics of the whole trunk at 10 Hz using the spatiotemporal-integration (STINT) method. **b** The MIP and depth-encoded images from the frontal view of the whole trunk. RL right lung, LL left lung, MLL middle liver lobe, LMLL left middle liver lobe, LLLL left lateral liver lobe, MA mesenteric artery, SI small intestine, ST stomach, CE cecum, SV seminal vesicle, SC spermatic cord, TE testicle, PE penis, AC arteria cruralis. **c** The cross-sections from the frontal view of the whole-trunk image (orange arrows in **b**) at different time points, indicating that the imaging depth extends to a minimum of 20 mm. Each cross-section was obtained by averaging a 1 mm thick 3D image in the thickness direction. **d** The maximum intensity projection (MIP) and depth-encoded images from the backside view of the whole trunk. IBAT interscapular brown adipose tissue, SP spleen, LK left kidney, RK right kidney. **e** The cross-sections from the backside view of the whole-trunk image (cyan arrows in **d**) at different time points.

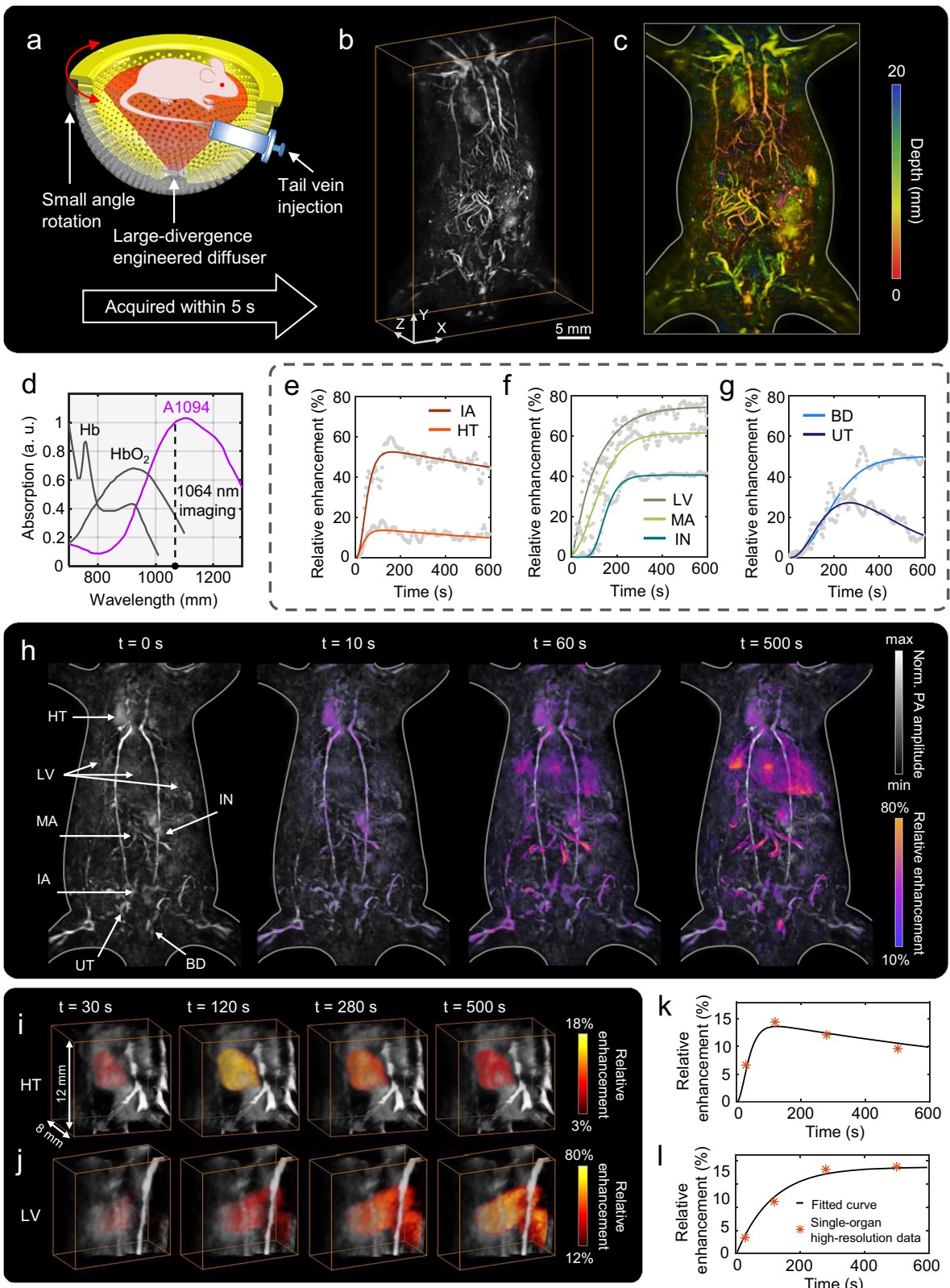

tracking time, and continuous whole-body tracking consisting of 120 frames was completed within the following 10 min. Fig. 6h illustrates the whole-body tracking results at time points 0 s, 10 s, 60 s, and 500 s (refer to "Methods" section for details), thereby evidencing the capacity of 3D-PanoPACT to record the metabolic trajectory of the A1094 probe with high spatiotemporal fidelity across the whole body. To validate the A1094 probe tracking methodology, we first conducted

phantom experiments prior to in-vivo test. Serial dilutions of A1094 (50, 60, 70, and 80 μM) were infused into Teflon tubes and imaged under a 60 mm FOV (Fig. S9a). Regional analysis revealed that while absolute pixel values failed to linearly correlate with probe concentrations due to non-uniform light illumination and spatial impulse response variations (Fig. S9b), the relative percentage changes remained consistent across the regions (relative standard deviation

**Fig. 6 | High-spatiotemporal-resolution tracking of small molecules at whole-body scale using 3D-PanoPACT. a** A schematic of the experimental setup and imaging strategy. **b** High-quality whole-body 3D imaging acquired within 5 s under this imaging strategy using the spatiotemporal-integration (STINT) method. **c** Depth-encoded results. **d** Comparison of the absorption spectra of A1094 with hemoglobin (HbO$_2$) and deoxygenated hemoglobin (Hb). Imaging wavelength: 1064 nm. **e–g** Relative enhancement (t = 0 s referenced) in signal intensity across different organs following A1094 administration. Raw data are shown in grayscale, while fitted curves are depicted in colors. IA iliac artery, HT heart, LV liver, MA mesenteric artery, IN intestines, UT ureter, BD bladder. **h** Mapping of whole-body cross-regional enhancement at 0 s, 10 s, 60 s, and 500 s post-A1094 administration, revealing molecular metabolic pathways. **i** High-resolution reconstructions of the heart at representative time points (t = 30, 120, 280, 500 s), with pseudo-colored overlay indicating relative signal enhancement (t = 0 s referenced). **j** Reconstruction results of the liver. **k** Comparison of heart high-resolution data at specific time points with the fitted curve of corresponding organ signal changes in whole-body probe tracking results. **l** Comparison of liver data.

<5%) (Fig. S9c), confirming that relative changes could reliably reflect metabolic patterns, a principle subsequently applied to in-vivo studies. To quantify the interregional variations of A1094, registrations were performed on the reconstructed image sequence, and the percentage changes in voxel amplitude relative to the initial state were calculated for different organs, thereby generating corresponding time courses. Employing different empirical models to fit the time courses enables the extraction of variation of A1094 from the fitted curves (refer to "Methods" section for details). From the fitted curves, we can extract several valuable parameters, such as the earliest time to reach the peak $t_p$, and the corresponding maximum percentage enhancement $C_m$. In the circulatory system, A1094 first reached the heart after injection, leading to the earliest signal changes, with $t_p \sim 120$ s. The iliac artery, being a branch of the aorta supplying the lower trunk, exhibited slightly slower signal changes, with $t_p \sim 145$ s (Fig. 6e). The $C_m$ value of the heart (~14%) was weaker than that in the iliac artery (~51%), possibly due to the heart's relatively stable coronary arterial blood supply system, which made it less susceptible to changes in blood composition compared to peripheral vessels[58] (Fig. 6e). The relatively slow signal decay may be attributed to the extended administration time coupled with increased blood flow associated with abdominal agent metabolism. The sustained enhancement in the liver, with $t_p \sim 505$ s and $C_m \sim 75\%$, further underscored the liver's pivotal role in the metabolism of A1094 (Fig. 6f). Additionally, the observed ~55 s delay of intestine signal change compared to the liver suggested A1094's metabolic route through the liver and then via the biliary tract to the intestine[16]. The $t_p \sim 300$ s and $C_m \sim 41\%$ indicated that the intestine, although involved in A1094 metabolism, did not play a dominant role (Fig. 6f). We also found that the mesenteric artery, with $t_p \sim 434$ s and $C_m \sim 61\%$, facilitated the hepatoenteral circulation mechanism, serving as a crucial link in maintaining the metabolic activities between the liver and the intestine[16,59] (Fig. 6f). Given that the ureter was directly connected to the kidneys, the observed signal changes in ureter ($t_p \sim 263$ s and $C_m \sim 26\%$) indicated that the kidneys were also involved in the metabolism of A1094 (Fig. 6g). This metabolic pathway was distinct from that of another commonly used probe, ICG[60]. In addition to the ureter signal, which initially increased and subsequently decreased, the bladder signal exhibited a sustained increase, with $C_m \sim 49\%$, approximately double that of the ureter $C_m$ value (~26%). This observation may be attributed to the bilateral ureters collectively channeling urine containing metabolites into the bladder, which, to some extent, underscores the efficacy of this analytical approach (Fig. 6g). The dynamic cross-regional tracking of A1094 across the whole body was visually demonstrated using 3D-PanoPACT (Movie S6). To validate the methodological robustness of our approach, high-resolution single-organ reconstruction (FOV = 12 mm) were performed of two organs - the heart (Fig. 6i) and liver (Fig. 6j). Organ-specific sound speeds were fine-tuned to ensure imaging fidelity. PA relative enhancement was quantified at four critical time points (t = 30, 120, 280, and 500 s) and compared with whole-body tracking results. The high-resolution organ measurements showed strong agreement with whole-body metabolic trends (maximum deviation: 8.26% at t = 500 s in heart), confirming the accuracy of 3D-PanoPACT for whole-body metabolic monitoring (Fig. 6k, l).

Compared to PET, MRI, and fluorescence imaging, 3D-PanoPACT offers unique advantages for small-molecule probe tracking. PET involves ionizing radiation, limiting its use for repeated imaging and long-term tracking, and has poor spatial resolution. MRI has lower sensitivity for certain molecular targets and often suffers from low temporal resolution and high cost. Fluorescence imaging is limited by shallow penetration and lower spatial resolution due to light scattering and can suffer from photobleaching and phototoxicity. In contrast, 3D-PanoPACT provides non-invasive, real-time, high-resolution imaging at deeper tissue depths with abundant and highly specific optical absorption contrast, making it a powerful tool for tracking small-molecule probes and studying systemic pharmacokinetics. These results not only highlight the high spatiotemporal resolution imaging capabilities of 3D-PanoPACT for large-scale 3D FOV but also further underscore its utility and potential for studying whole-body biological processes in practical applications.

## The spatiotemporal-integration (STINT) method

Theoretically, the transducer array composed of 1024 elements is still not the optimal approach for providing a large enough FOV radius (denoted by $r$), which is determined by the following equation[61]:

$$r = \frac{v}{4f_{uc}}\sqrt{\frac{2N}{\pi}} \tag{1}$$

where $v$ denotes the speed of sound in the medium and $N$ denotes the total number of elements. $f_{uc}$ denotes the upper cutoff frequency of the transducer where the frequency-dependent SNR decreased to one above the central frequency. With calculation, $r = 1.5$ mm when $N = 1024$ and, if $r$ grows to 20 mm, $N$ will reach a number of approximately 175000 ($v$ is set to 1500 m/s, and $f_{uc}$ is calculated as 6.26 MHz by definition)[61]. The STINT method serves to physically augment detection density and broaden the imaging FOV (especially in the deep tissue) with an appropriate reduction in temporal resolution. Here, the arrangement of transducer elements in a Fibonacci grid constitutes the essential hardware foundation for implementing the STINT method. The Fibonacci grid provides an elegant description of "uniform" angular sampling on the surface of a sphere, while notably lacking any central symmetry. This unique property allows us to rotate the array by small angles to generate an equivalent high-density detection array with near-uniform element distribution. In the phantom validations, rotating the original array by 1° for 10 consecutive times yielded a detection array comprising 10240 transducers (1024 × 10), and the reconstruction of leaf vein details using the equivalent high-density array was significantly superior to that achieved with the original array within the same FOV (Fig. 7a, b). In this study, the STINT method was further implemented in two different modes to accommodate various imaging targets and desired outcomes. This feature makes it suitable for imaging non-ultrafast physiological processes (for example, the hemodynamics in the Circle of Willis and the whole-body tracking of small molecules). For some exceptionally large FOV (for example, whole-trunk imaging), the STINT method can also reconstruct 3D image sequences, achieving the imaging of dynamics.

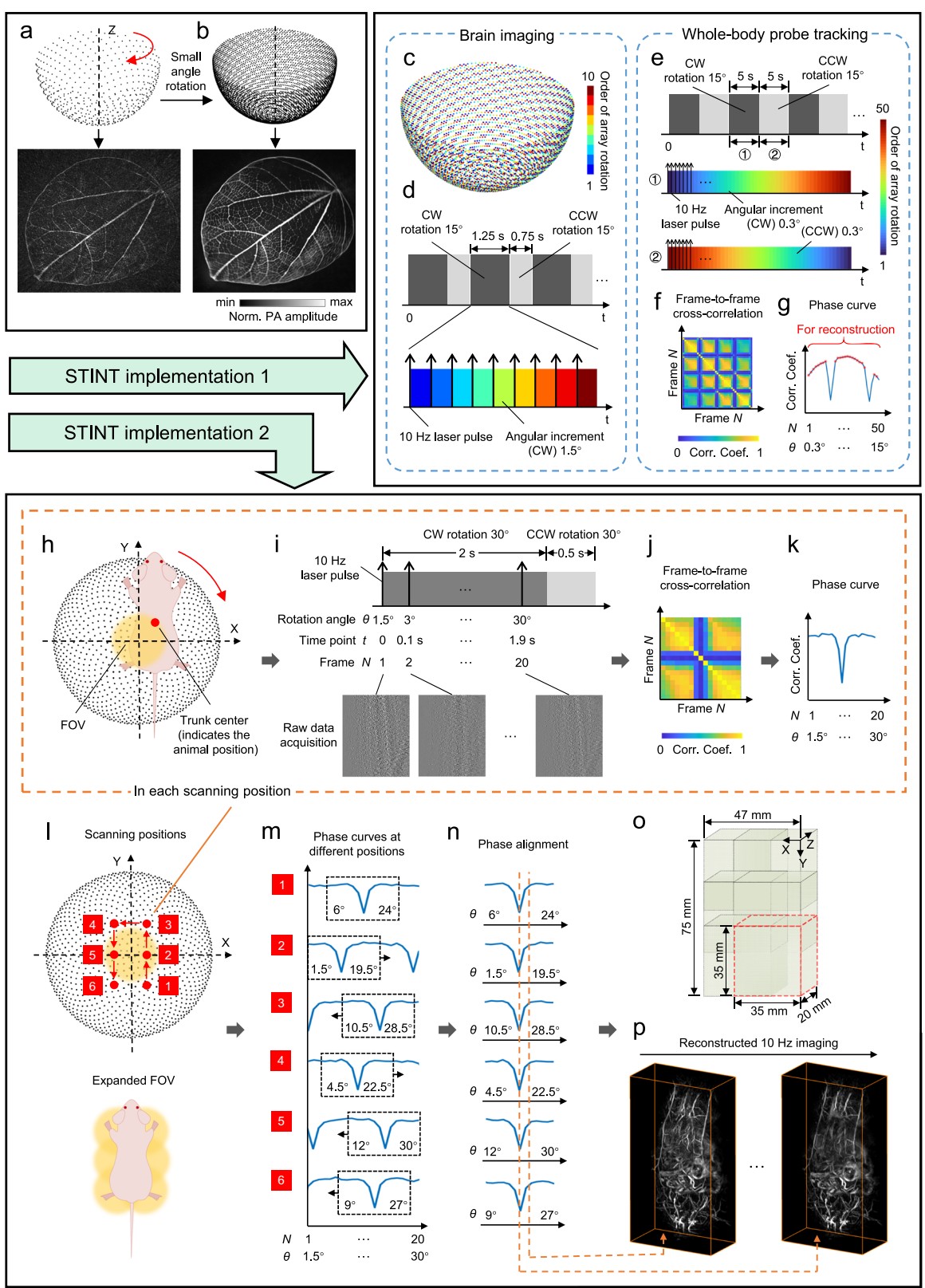

We employed the first implementation of the STINT method in brain imaging and the whole-body probe tracking applications, achieving large-FOV, high-spatiotemporal-resolution 3D reconstruction solely through rapid, small-angle rotations of the transducer array during each imaging cycle (Figs. 3, 4, 6). In brain imaging, we utilized the spherical array to integrate into an equivalent dense array with 10240 elements by rotating it 10 times at an equidistant angular increment of 1.5°, ensuring the uniformity of the array element density after integration according to this method (Fig. 7c). During each imaging cycle, the array was synchronized with a 10 Hz dual-wavelength laser for triggering. After each group of laser pulses and raw signal acquisition, the array rotated clockwise (CW) by 1.5° (Fig. 7d, one black arrow represents a group of temporal closely-spaced dual-wavelength laser outputs). The 10 rotations took a total of 1.25 s (including the time

**Fig. 7 | The spatiotemporal-integration (STINT) method. a** The reconstruction of the original hemispherical array with 1024 elements. **b** The schematic diagram and reconstruction results of the STINT method. The leaf skeleton was used as phantoms for a more intuitive illustration. **c–g** The first implementation of STINT. **c** The array motion timing in brain imaging. The colorbar indicates the order of array rotation. **d** The overall timing diagram of array rotations and the synchronization of the laser pulse with the array motion in each cycle in brain imaging. CW clockwise, CCW counterclockwise. **e** The overall timing diagram of array reciprocating rotations and the synchronization of the laser pulse with the motions in whole-body probe tracking. **f** The frame-to-frame cross-correlation matrix. **g** The phase curve generated from the correlation matrix. Red star symbols indicate the static frames used for reconstruction. **h-p** The second implementation of STINT. **h** The relative position of the animal to the FOV at a certain scanning position in the X-Y plane. The red point indicates the position of the animal. **i** Synchronization sequence diagram

of dual-wavelength laser pulse outputs, array rotation motion, and data acquisition at a certain scanning position (only single-wavelength raw data is shown here). **j–k** The frame-to-frame cross-correlation matrix and the generated phase curve. **l** Schematic diagram of the scanning process in the X-Y plane (intervals of 12 mm along the X-axis and 20 mm along the Y-axis) and the expanded FOV (the red point represents the position of the animal). **m** The phase curves at different scanning positions. The complete respiratory cycle is enclosed by black dashed rectangles, with the range of array rotation angles annotated. **n** The aligned phase curves from different positions. The orange dashed line indicates the raw data used for single-frame image reconstruction. **o** The process of integrating FOVs from different scanning positions into a large FOV in single-frame whole-trunk image reconstruction. The red dashed box represents the FOV at a certain scanning position. **p** 10 Hz imaging results of the whole-trunk dynamics using the STINT method.

for motor start-up and stabilization), followed by a counterclockwise (CCW) rotation of 15° to the initial position within 0.75 s (Fig. 7d), ready for the next imaging cycle. Owing to the rigid fixation of the skull by the animal holder, data acquired within each imaging cycle can be utilized for spatiotemporal-integrated reconstruction without the need for respiratory gating. In whole-body probe tracking, to achieve a large FOV, the array rotated CW by 15° within 5 s, acquiring 50 frames of raw data with a 10 Hz laser pulse, with each frame corresponding to an angular increment of 0.3° of the detection array (Fig. 7e). The array then rotated CCW by 15° over the following 5 s to return to its initial position, during which a similar data acquisition procedure was executed, with the sole difference being the reversed order of array positions. To elaborate, the reciprocating small-angle rotation of the array enabled continuous data acquisition during both CW and CCW rotations, thereby maximizing imaging temporal resolution (Fig. 7e). Due to the influence of thoracic respiratory motion, the 50 frames of raw data require respiratory-gating screening[62,63]. The cross-correlation matrix was then obtained by calculating the correlation coefficient between frames. It could be seen that some of the frames were obviously different from other frames, containing the respiratory movement information (Fig. 7f). Due to symmetry, averaging the cross-correlation matrix along any dimension yielded the phase curve (Fig. 7g). By combining the derivative of the phase curve with threshold segmentation, frames with high correlation coefficients and low derivative values were identified as static frames, which were then used for spatiotemporal-integrated reconstruction (Fig. 7g). In STINT, the raw data corresponding to each array angular position was reconstructed according to the dual-speed-of-sound universal back-projection algorithms, in which each data frame was normalized using the detector surface signal to eliminate the influence of laser energy fluctuations. The detection coordinate used for each data frame was obtained by rotating the corresponding angle of the original coordinates around the rotation axis that has been calibrated. The images from all angular positions were then superimposed to obtain the reconstructed images with the STINT method (Supplementary Methods). It should be noted that reconstructing images directly from all-angle data is equivalent to reconstructing separately and then summing the results, as the universal back-projection[64] is a linear reconstruction process. However, the former approach requires higher GPU memory, which can be addressed with a more advanced GPU that supports larger memory and parallel computations.

We then employed the second implementation of the STINT method in reconstructing the high-resolution dynamics of the whole trunk of a living animal (Fig. 5), allowing a large 3D FOV to be obtained in a short period of time and reducing artifacts caused by spatial undersampling. This implementation utilized a more complex synthesis of array rotation and translation, achieving spatial resolution comparable to that at the field center across the entire FOV, while alleviating the resolution degradation in peripheral regions caused by

spatial impulse response (SIR)[65,66]. To apply the STINT method in this case, the motorized translation stages drove the animal to scan six positions in the X-Y plane, rotating the array at a small angle around Z-axis at each fixed position, and collecting all the raw data for image reconstruction. In each X-Y scanning position, part of the animal's trunk was inside the FOV of the original array (Fig. 7h). The output of a 10 Hz dual-wavelength laser pulse was triggered synchronously with the rotating motion of the hemispherical array around the calibrated axis, and each group of laser pulses triggered the array to rotate CW by an angular spacing of 1.5° (Fig. 7i, one black arrow represents a group of temporal closely-spaced dual-wavelength laser outputs). In this process, 20 frames of dual-wavelength raw data sequence were acquired, after which the array was rotated CCW back to the initial position, taking a total of about 2.5 s (note that only single-wavelength raw data was shown here for clarity since the time interval of the dual-wavelength laser pulse was almost negligible relative to the laser repetition period, and, the processing methods of the dual-wavelength data in this paragraph were the same). The cross-correlation matrix as well as the phase curve were then obtained using the approach analogous to that described previously (Fig. 7j, k). Throughout the imaging procedure for the whole trunk, there existed relative scanning between the animal and the FOV. We conducted scans at six positions in the X-Y plane, with intervals of 12 mm along the X-axis and 20 mm along the Y-axis, which enables the creation of a comprehensive expanded FOV that encompasses the whole trunk (Fig. 7l). The coordinate system of the translation stages had been meticulously aligned with that of the hemispherical array, ensuring they shared the same Cartesian coordinate system to avoid coordinate mismatch. Corresponding raw data sequences and phase curves were generated at each scanning position and, by arranging the phase curves together, it became apparent that each contained at least one complete respiratory cycle with a frequency of approximately 1 Hz (Fig. 7m). We selected the complete respiratory cycle composed of 11 frames from position 1 as a reference, and performed sliding correlation on the phase curves from other positions to complete phase alignment, generating a new sequence with 11 frames (Fig. 7n, the frame rate was still 10 Hz). Within these curves, identical frame points represented the in-phase states within one respiratory cycle, albeit corresponding to different array detection angles (Fig. 7n). For each scanning position, 11 frames of raw data were individually reconstructed using the array coordinates of the corresponding angle of rotation with an FOV of $35 \times 35 \times 20$ mm³ (the voxel size $0.1 \times 0.1 \times 0.1$ mm³). According to the scanning intervals, the FOV for the whole trunk was set to $75 \times 47 \times 20$ mm³ (Fig. 7o). It was noted that different FOVs overlapped, which inhibited artifacts to a certain extent and improved the imaging SNR. By reconstructing the raw data corresponding to identical frame points from different scanning positions and integrating them according to the scanning intervals (orange dashed line in Fig. 7n), a single-frame 3D image of the whole trunk was obtained (Fig. 7n, p). It

should be noted that the reconstruction results from each position needed Gaussian blur (filter size = 13, σ = 13) along the edge planes perpendicular to the X-Y plane to prevent discontinuities during the superposition process. After traversing all frame points in the aligned phase curves, the 10 Hz whole-trunk dynamics was reconstructed with a total imaging time of approximately 15 s (Fig. 7p). In the whole process, while the STINT method formed a large FOV, an imaging region was reconstructed by spatial sampling from different positions and angles, which weakened artifacts and improved image fidelity.

To evaluate the effectiveness of the STINT method, we imaged a complete leaf vein at 690 nm with an FOV diameter of 60 mm. In the single-pulse imaging results, the outline of the leaf and the larger vein branches can be observed, indicating that 3D-PanoPACT has near-isotropic 3D PA imaging capabilities without motion, as evidenced by the maximum intensity projections (MIP) in the X-Z and X-Y planes (Fig. S10a). By integrating 20 sets of data within 2 s using the STINT method, 3D-PanoPACT reconstructs images that reveal rich details, with significantly reduced artifacts and markedly enhanced imaging fidelity (Fig, S10b). We first provide feature correlation coefficient (FCC) at the same image location before and after using the STINT method as a metric to quantify the difference in image features. It can be observed that in the main vein, the FCC value is ~ 0.97 in the X-Y plane (Fig. S10c) and ~ 0.98 (Fig. S10d) in the X-Z plane, indicating that the STINT method does not introduce additional image blurring or artifacts. At the edge branches of the leaf veins, the FCC value is 0.39, indicating that the STINT method can indeed reveal detailed features that are almost impossible to reconstruct in single-pulse imaging (Fig. S10e). We further calculate the standard deviation (STD) as a metric of artifact suppression. It is observed that the STD value at the edge of FOV is reduced by 8 times after using the STINT method, indicating that artifacts are significantly weakened, and this effect could be further enhanced by integrating more data for reconstruction (Fig. S10f). We subsequently conducted a comparative analysis between the efficacy of the STINT method and the outcomes of numerical spatial interpolation reconstruction. In the interpolation, each transducer served as a node to construct a Delaunay triangulation mesh. Virtual transducers were placed at the centroids of each triangular element, and their corresponding data were generated by averaging the data from the three vertex transducers. A single spatial interpolation increased the total number of transducers by approximately threefold, thus, the STINT method performed a lateral comparison by synthesizing physical transducer arrays at three angular positions (Fig. S11b, c). By comparing the imaging results of the leaf skeleton, it is evident that spatial interpolation causes significant blurring of the details and degraded image quality (Fig. S11a, b), whereas the STINT method preserves the details, suppresses artifacts, and thereby improves image fidelity (Fig. S11a, c). We further utilized whole-body imaging to evaluate the impact of respiratory motion on the STINT method (the first implementation). Apparently, the reconstruction results using the STINT method, with or without respiratory gating, are superior to those of single-pulse imaging, as the 1024 elements are severely spatially-undersampled for the whole-body FOV (Fig. S12a), which is consistent with the leaf vein results. The benefits of respiratory gating are evident in reconstructing the details, particularly in the small vessels of the liver and the iliac artery, where it significantly improves the visibility and definition of these structures (Fig. S12b). To quantify this, the image sharpness is evaluated using the Sobel operator to detect gradients, and the standard deviation of the gradient magnitudes is calculated as the sharpness metric. Higher values indicate clearer images, while lower values suggest blurriness. Calculations show that the STINT method with respiratory gating improves image sharpness compared with the others (1.375 and 1.27 times for liver vessels, respectively; 1.25 and 1.07 times for the iliac artery, respectively), avoiding image blurring caused by motion (Fig. S12b). Additionally, this method hardly alters the existing image features in

single-pulse results, as evidenced by the surface vessels, with a correlation coefficient for the cross-section of approximately 0.989 (Fig. S12c). In summary, we have verified the performance of the STINT method from multiple aspects.

## Discussion

In this study, we introduced the design philosophy, parameter calibration, and imaging strategies of a PA imaging device—3D-PanoPACT —and demonstrated its broad applicability and significant potential for practical biological research. The 3D-PanoPACT system has been designed to non-invasively image anatomical structures and functional dynamics, offering superior spatiotemporal resolution, high fidelity, and a broad 3D FOV. This system encompasses several advanced features: (1) The design of array element arrangement using a high-density Fibonacci grid provides single-pulse 3D imaging ability. The hemispherical ultrasonic transducer array utilized in 3D-PanoPACT covers all receiving angles in k-space, with a uniform distribution of 1024 elements ensuring spatial sampling density. The array directly forms an isotropic 3D FOV without any scanning, supporting high-quality, real-time imaging, and the frame rate is solely dependent on the laser pulse repetition rate. The single-pulse imaging capability enables 3D visualization of dynamic processes at video frame rates, enhancing its attractiveness (Fig. 2). By synchronously controlling two or more lasers, real-time multi-wavelength imaging can be achieved, which is capable of capturing dynamic functional information that undergoes rapid changes (Fig. 4f–k). (2) The STINT method ensures a wide-field 3D FOV with high spatiotemporal resolution. Based on the Fibonacci grid array, the STINT method only requires a small rotation angle of the array to achieve equivalent spatial upsampling, enabling FOV expansion with minimal sacrifice of temporal resolution. With this method, 3D-PanoPACT achieves pioneering progress by distinctly imaging the anatomy of the important cerebrovascular network—the Circle of Willis—and monitoring its functional dynamics (Figs. 3 and 4a–e). We also use the STINT method to obtain high-fidelity 3D dynamic imaging of the whole trunk (Fig. 5) and effectively track the metabolic pathways of the small-molecule probe A1094 across multiple organs at the whole-body scale (Fig. 6), thereby highlighting the practical utility of our approach. (3) The meticulous engineering implementation. The transducer array design was validated, with sensitivity variation among the 1024 elements kept within 4% and bandwidth variation within 1.5% during fabrication. The incorporation of preamplifiers, suppression of channel crosstalk, and shielding of noise are important factors in improving the SNR of the raw data, thereby enhancing imaging depth. The application of dual-speed-of-sound reconstruction allows the accurate depiction of deep-seated features and also improves imaging depth. The calibration of system array element positions and the rotation axis further optimizes the spatial resolution. The application of EIR deconvolution processing and optical fluence compensation enhances the quantitative capability of images and improves the accuracy of functional parameter estimation in biological applications. All the implementation methods and technical details mentioned above collectively determine the imaging performance of 3D-PanoPACT (Table S1, see Supplementary Information for details).

To more intuitively demonstrate the performance advantages of 3D-PanoPACT, we conducted a detailed comparison with other advanced PACT systems (Table S1–4). A comparison is made primarily between the 3D-PanoPACT system and other PACT systems based on spherical array configurations (Table S1). In terms of ultrasound transducers, 3D-PACT features a larger element size with optimized SNR (3D-PanoPACT: 5 mm; 3D-PACT: 0.7 mm; PAI-04: 1.5 mm; FONT: 1–2.5 mm). The ratio of element size to the central wavelength (ROSW) is calculated to qualitatively characterize the aperture angles of different transducers. The 3D-PACT system, with a ROSW value close to 1, forms an ideal receiving aperture but has limited sensitivity due to its small transducer size and is more affected by the quantization noise of

data acquisition. Other systems have comparable ROSW values (3D-PanoPACT: 10.53; PAI-04: 4; FONT: 3.33-8.33) indicating similar aperture angles, yet 3D-PanoPACT achieves higher sensitivity. The optimization of transducers using materials with wider bandwidth or higher sensitivity may potentially enhance imaging performance, a possibility that warrants further investigation. The metric "acquisition time per voxel per laser pulse" further highlights the order-of-magnitude improvement in the speed of imaging voxels (Table S1). Additionally, by incorporating the STINT method, 3D-PanoPACT fully exploits the geometric properties of the Fibonacci grid distribution of the array elements, further extending the FOV and imaging depth while ensuring real-time functional imaging capabilities. This advantage has been effectively applied in the anatomical visualization and functional monitoring of the Circle of Willis (Table S1, 3). Despite 3D-PACT and FONT systems not partially removing the skull, 3D-PanoPACT focuses on whole-brain imaging depth to capture physiological activities from the brain base (Table S1). By comparing with the state-of-the-art small animal PACT systems, it can be seen that 3D-PanoPACT's advantages in real-time imaging remain significant, greatly expanding its application range in recording ultra-fast physiological activities. Additionally, the whole-brain and whole-body imaging results provided by 3D-PanoPACT exhibit unparalleled detail and fidelity, with fast imaging speeds (Table S4). It is noteworthy that 3D-PanoPACT achieves a considerable imaging depth, a critical capability lacking in other systems (metric "Cross-sections" in Table S4). We also compared the cross-sections obtained by 3D-PanoPACT with those from the most advanced 2D PACT systems which are specifically designed for imaging cross-sections, and found that the imaging quality is comparable, further validating 3D-PanoPACT's deep-tissue imaging capability (Table S2). Additionally, 3D-PanoPACT has observed new physiological processes through a series of functional imaging paradigms, marking a significant advancement in whole-body imaging of small animals (Table S4).

We specifically explore the uniqueness of the STINT method in 3D-PanoPACT, particularly in comparison with the 3D-PACT (Lin et al. 2021), an advanced imaging system based on rotational scanning[22–24]. From the perspective of array design, the angular isotropic nature of the Fibonacci grid enables 3D-PanoPACT to achieve high-quality 3D imaging within a certain FOV without requiring any additional motion (Fig. 2). The imaging frame rate is determined by the laser repetition rate in 3D-PanoPACT, while 3D-PACT must rely on rotation to achieve 3D imaging. This is because its array distribution does not possess the characteristic of 3D spatial isotropy, which limits the ability of 3D-PACT to capture high-speed biological processes, while 3D-PanoPACT is capable of this task. Moreover, 3D-PACT requires a minimum rotation of 90° to cover the 2π solid angle in spatial frequency, which corresponds to a large angular velocity of 0.785 rad/s when performing rapid imaging. With large mechanical load, the high-speed rotation inevitably induces vibrations, causing random deviations of the element positions and thereby degrading spatial resolution, with this phenomenon becoming increasingly pronounced as the transducer's center frequency increases. For comparison, in 3D-PanoPACT, a small rotation angle suffices to achieve uniform spatial upsampling, markedly reducing random errors caused by vibrations and ensuring spatial resolution. Assuming a rotation of 15° in 3D-PanoPACT, the angular velocity is 6 times slower than that of 3D-PACT for the same imaging frame rate, which significantly reduces vibrations. In 3D-PanoPACT, the essence of using the STINT method is the spatial upsampling process, which involves trading off temporal resolution for the imaging FOV. In other words, even without the STINT method, 3D-PanoPACT already possesses an isotropic 3D-FOV (Fig. 2), and, the STINT method further achieves spatially scalable imaging (Figs. 3–6) by integrating multiple temporal data. In contrast, the array rotation in 3D-PACT is more akin to a scanning method (e.g., using linear array translation for 3D image reconstruction[67]), and the system itself does

not possess an isotropic 3D FOV. This leads to some imaging that can be achieved in 3D-PanoPACT being unattainable in 3D-PACT (Figs. 2, 5). Therefore, even though both involve the rotation of the array, the implementation logic of 3D-PanoPACT is significantly different from that of 3D-PACT, and the STINT method is more suitable for describing the imaging approach of 3D-PanoPACT.

While 3D-PACT exhibits superior imaging performance, it still has some limitations and areas for further improvement. We briefly discuss several rational prospects: (1) Improvement of ultrasound transducer materials. The current system employs traditional piezoelectric ceramic composite materials for its transducers, which, despite their high robustness, have limited receiving bandwidth. Since the transducers in PA imaging are used solely for receiving acoustic waves and not for transmission, replacing them with a Polyvinylidene Fluoride (PVDF) array featuring a wider receiving bandwidth may further enhance the system's spatial resolution and imaging quality. (2) Optimization of temporal resolution. Despite its single-pulse 3D imaging capability, the low laser pulse repetition rate limits 3D-PanoPACT's ability to capture ultrafast physiological processes in this study. For instance, in imaging liver dynamics, we successfully extracted the heartbeat frequency from the temporal sequence but the detailed dynamics of cardiac substructures remained elusive. This limitation arises because these dynamics typically manifest as high-frequency harmonics of the heartbeat frequency, and the current imaging frame rate of 25 Hz fails to meet the Nyquist sampling criterion, thereby making it difficult to resolve this information. For reference in echocardiography imaging[34], an imaging frame rate exceeding 100 Hz may significantly enhance the capability to capture rapid physiological phenomena but also impose higher demands on high-energy, high-repetition-rate pulsed laser equipment. With a higher laser repetition rate, increasing the rotation speed of the array without introducing additional vibrations could further optimize the temporal resolution of the STINT method. The implementation involves reducing the rotational stage load by changing the connection mode between the array and the preamplifier and data acquisition modules from direct plug-ins to cabled connections, presenting higher engineering requirements on the electromagnetic shielding. In this way, the rotational stage only supports the weight of the transducer array without incorporating additional housings. (3) SIR correction for large FOV optimizes spatial resolution. The SIR arises from the non-ideal nature of the transducers. In whole-body probe tracking, the edge of FOV is not impacted by undersampling after the application of the STINT method, making the effects of SIR more pronounced, specifically that the resolution decreases closer to the edge of FOV. Accurate calibration of the SIR coupled with spatial deconvolution, or the implementation of model-based image reconstruction incorporating the SIR, may potentially mitigate the resolution degradation, thereby further enhancing the imaging quality[65,68]. (4) Further exploration of imaging depth. Employing omnidirectional laser illumination can significantly enhance the raw data's SNR from deep tissues. On this basis, the development of advanced segment-based iterative solvers[69,70], or other methods for correcting acoustic heterogeneity for 3D reconstruction may substantially improve the imaging SNR of deep-seated features.

To highlight the versatility of 3D-PanoPACT, we have presented several key applications. Real-time imaging of the hepatic vascular network allowed for pulse wave extraction and 3D arterial mapping, serving as indicators for hepatic hemodynamics and liver disease. Imaging neurovascular coupling confirmed 3D-PanoPACT's capability to rapidly capture whole-brain physiological changes, facilitating deep-brain neural activity studies. Monitoring pharmacological responses in the whole-brain vasculature enabled functional imaging of the Circle of Willis and other cerebral vessels, underscoring its significance in cerebrovascular research. Additionally, 3D-PanoPACT demonstrated unique advantages in cross-regional and multi-organ studies through whole-body 3D dynamic visualization and real-time

small-molecule probe tracking, opening pathways for investigating systemic pharmacokinetics, circulation, and tumor metastasis. Here, we envision several potential applications. Through the integration of endogenous light illumination with 3D-PanoPACT, a greater depth of in-vivo features can be unveiled[71–73]. A prominent example involves the in situ visualization of structures surrounding the pancreas[74,75]. This is attributed to the limited optical absorption of the pancreas within the near-infrared imaging spectrum, resulting in reduced contrast and susceptibility to signal interference from adjacent splenic tissues. Another promising illustration pertains to the comprehensive hemo-dynamic surveillance of the carotid artery, Circle of Willis, and the whole brain under specific conditions. This may provide a promising means to elucidate the intrinsic mechanistic interplay between cere-bral and cardiac mechanisms. Addressing the challenge of improving spatiotemporal resolution and detecting sensitivity to capture deep-seated neural activities amid a strong absorption background requires further efforts[76–78] and the utilization of ion-sensitive contrast agents, possibly explored exclusively with the PA modality, holds promise for advancing this endeavor. By further increasing the density of array elements or incorporating deep learning methodologies[79], the single-pulse FOV could potentially be expanded to a whole-body scale. This advancement will not only enable the high-speed tracking of dynamic changes in probe concentration throughout the entire body but, when combined with data analysis techniques such as optical flow methods[80], could also facilitate the generation of 3D vector maps depicting the migration of probes[81]. This potential of 3D-PanoPACT may offer more perspectives for research in fields such as metabo-lomics and oncology, thereby significantly enhancing our under-standing of complex biological processes.

## Methods

### System construction and experiment configuration

In the 3D-PanoPACT system, we employed a hemispherical transducer array with 1024 elements (3.5C3.5-SR100-1024, ULSO TECH Co., Ltd). The hemispheric housing, with a diameter of 20 cm, was made of white resin to avoid generating unwanted PA signals. Each transducer ele-ment was made of piezoceramics and had a central frequency of 3.16 MHz and a bandwidth of 3.67 MHz according to our test (Fig. S1). Each element had a diameter of 5 mm to ensure receiving sensitivity. A layer of gold film was plated onto each element to weaken the surface PA signals. The elements were evenly distributed on the inner surface of the housing following the Fibonacci grid. Integrated operational amplifier chips (PSA4-5043 + , Mini-Circuits) were adopted for the lab-made preamplifier boards. The DAQ boards (40 MHz sampling rate, 9 - 48 dB programmable gain, 12.5 kHz to 25 MHz analog bandwidth, Legion ADC, Photosound) and preamplifier boards were encapsulated into a metal box, which was grounded to realize electromagnetic compatibility. The box was mounted directly onto the interface of the transducer array to reduce coupling noise. A large-sized rotation stage (Y200RA400, Jiangyun Optronics) was installed coaxially with the hemispherical housing at the bottom of the system body with a max-imum angular resolution of 0.0003125°. Two types of nanosecond lasers were applied in the system. A Nd:YAG laser (SGR-S500, 10-Hz pulse repetition rate, Beamtech Optronics Co., Ltd) provided 1064 nm illumination and the other Nd:YAG laser (Nimma-900, 10-Hz pulse repetition rate, Beamtech Optronics Co., Ltd) equipped with an OPO module (Continuum-OPO, Deyang Tech Co., Ltd) provided 670-900 nm tunable output. Two laser outputs, through a series of silver reflectors and beam combiners (Union Optic Tech Co., Ltd), were guided to an engineered diffuser (EDC-10-G-1R, RPC Photonics Inc.) mounted at the bottom of the transducer array, forming a uniform illumination pattern at the array center with a diameter of 4 cm. The diffuser mount was designed to be detachable, enabling replacement with an engineered diffuser featuring a 60° divergence angle (EDC-50, RPC Photonics Inc.) during whole-body probe tracking experiments to

achieve wide-field illumination with a diameter exceeding 12 cm. The two lasers were able to support 25 Hz single-wavelength mode and 10 Hz dual-wavelength mode. Three motorized linear stages (L-509.20sd00, 52 mm travel distance, 0.2 μm positioning accuracy, PI Inc.) were assembled intervertically to construct a triaxial translation stage. The customized animal holder was mounted onto the stage and set right above the array. The lasers, DAQ modules, and mechanical modules were all set to external trigger mode and were synchronized via a digital delay generator (DG645, Stanford Research Systems). The controlling programs were integrated into a graphical user interface, which was created by LabView for convenient operations.

During the imaging process, the inner space of the transducer array was filled with deionized water. The water temperature was maintained at 25ºC by pump circulating and bath heating to avoid hypothermia in animals. In trunk imaging, the mouse holder was composed of a gas pipe, a nose hoop, and a support ring. A small notch was designed at the pipe end to hook the animal's incisor. Then the nose hoop was stuck to the animal's head and the limbs and tail were lightly tied to the support ring. In brain imaging, the support ring was replaced with a small platform. The animal was in an upside-down posture and half of its head was submerged in water. A patch of par-afilm was used to seal the nose hoop to prevent animals from drowning. The maximal laser fluence on the animal's skin approxi-mately reached 5 mJ·cm$^{-2}$ at 1064 nm, with a 25 Hz repetition rate (equivalent to 12.5 mJ·cm$^{-2}$ at 10 Hz), which was below the American National Standards Institute (ANSI) safety limits for laser exposure (100 mJ·cm$^{-2}$ at 10 Hz repetition rate for 1064 nm).

### Geometric calibrations of acoustic detection

First, for the acoustic transducer array, we calibrated the spatial position of each element. From the perspective of polar coordinates, the ele-ment's position was defined by the polar angle, azimuth angle, and radial distance. The array housing was produced with reserved apertures via high-precision machining. Thus, the diameter of the housing was treated as exactly 20 cm, and, element angular positions were rational to be regarded as following the ideal Fibonacci grid distribution. The 1024 transducer elements were individually inserted into the aperture, which introduced deviations in radial position compared to the ideal Fibonacci grid and required calibration. Here, a point absorber (black mono-disperse polystyrene microspheres, 50 μm in diameter) was separated under an optical microscope and then fixed into a phantom (1% agar-ose). The phantom was mounted onto the triaxial translation stage so that the position of the point absorber could be adjusted flexibly. The calibration procedure ran in three steps:

(1) We set the reconstruction field to 100 μm centered at the origin (i.e., the center of the hemisphere). Then the point absorber was adjusted while imaged at 690 nm repeatedly until the recon-structed absorber was located at the field center. Thereupon, the PA raw data was acquired.

(2) The water temperature was immediately measured with a high-precision thermocouple. Then the speed of sound in the medium, denoted as $V_M$, was calculated according to the classic temperature-velocity formula.

(3) In the raw data of the $n$'th channel, we positioned the signals from the transducer surface and the point absorber, and the sample points where the PA signals peaked were recognized (signal saturation was avoided to ensure accuracy). $\Delta s_n$ denotes the result of subtracting two sample points, and the radial deviation for the $n$'th transducer element $\Delta r_n$ is written as:

$$\Delta r_n = \Delta s_n \frac{V_M}{F_S} - R \tag{2}$$

where $F_S$ denotes the DAQ sampling frequency (i.e., 40 MHz) and $R$ denotes the array radius (i.e., 10 cm).

(4) Then the corrected position of $n$'th element $(x_n, y_n, z_n)$ is written in Cartesian coordinates as:

$$x_n = x'_n \left(1 + \frac{\Delta r_n}{R}\right), \, y_n = y'_n \left(1 + \frac{\Delta r_n}{R}\right), \, z_n = z'_n \left(1 + \frac{\Delta r_n}{R}\right) \quad (3)$$

where $(x'_n, y'_n, z'_n)$ denotes the ideal Fibonacci grid of the $n$'th element.

As shown in Fig. S3a, b, the raw data was aligned evidently, and the element radial deviation fluctuated within a range of $\pm 0.5$ mm (Fig. S3c).

Subsequently, we carried out the calibration of the system rotation axis (Fig. S3d), which was important in the implementation of the STINT method (detailed in the following sections). Ideally, the rotation axis should align with the Z-axis of the transducer array. However, deviations in practice occurred and affected image quality. The misalignment arose from an inevitable assembly error between the transducer array and the rotation stage. Here we adjusted the point absorber 15 mm away from the origin point, and the absorber was imaged while the array rotated 70 degrees synchronously. Fifteen frames in raw data were selected at equal time intervals and co-reconstructed, forming the 3D image with a chain of point absorbers. Apparently, the points determined a circle and the normal through the circle center was treated as the actual rotation axis of the system. As shown in Fig. S3d, the ideal and actual motion trails and corresponding centers were labeled in red and green, respectively. As a result of the analytical geometric calculation, it could be concluded that the actual axis was parallel to the ideal one, which was shifted by 0.07 mm along X-axis and 0.29 mm along Y-axis. Applying the calibration methods in leaf skeleton imaging, the resolution and contrast are improved (Fig. S3e, f). The calibration of element position globally enhanced the image quality while the calibration of the rotation axis mainly acted on the region relatively away from the center of FOV, which was easier to be affected by the process of spatiotemporal integration.

### Test of electrical impulse response (EIR) and image deconvolution

On account of the limited bandwidth of the transducers and subsequent series of circuits, the acquired PA signals differ from the initial-pressure-generated signals, which could be regarded as a convolution of the original PA signals and EIR. EIR reduces the spatial resolution to some extent and results in the loss of low-frequency signals, which deteriorates the quantitative ability of PA images. In imaging the whole-brain functional dynamics, the reconstructed amplitude of blood vessels is related to the accuracy of sO$_2$ calculation. Additionally, the neurovascular coupling response in the cortex appeared as regionally concentrated clusters containing low-frequency information. Therefore, to better visualize the whole-brain functional dynamics (corresponding to the contents of Fig. 4), we performed deconvolution on the raw data using the precisely measured system EIR before image reconstruction. The following sections will describe the EIR testing method in 3D-PanoPACT and the effects of image deconvolution.

Here, we applied edge-emission method to obtain the system EIR. A polyimide film with a thickness of 10 μm was cut into a $2 \times 15$ mm rectangle as the PA signal source. Then the flakelet was fixed into a 1% agarose phantom (0.5% intralipid was added for homogenization of optical fluence). The phantom was fixed to the triaxial translation stage and imaged at 690 nm repeatedly until the end face of the flakelet's short side was centered at the origin of the transducer array. From the perspective of signal generation mechanism, the thin-sheet-shaped absorber was uniformly illuminated, and a pair of negative and positive unipolar PA pulses were produced at the head and tail edges, respectively, which seemed to split the bipolar PA pulse from a point absorber. Therefore, with 1000 frames average to realize high SNR, the PA signal received by the transducer element located opposite the

edge could be taken as system EIR 1 (Fig. S4a). As a comparison, a tungsten wire with a diameter of 10 μm was manipulated similarly, and the numerical integration was operated on the acquired PA signal to generate EIR 2, which was applied to compensate for the bipolar-input-introduced waveform distortion (Fig. S4b). As displayed in Fig. S4c, the waveform of EIR 1 was in a more regular shape compared to EIR 2. Notably, the measured EIRs were both averaged 1000 times to obtain a high SNR. To illustrate the improvements with EIR deconvolution, three pieces of square polyimide film with side lengths of 3 mm, 6 mm, and 9 mm, were fixed into 1% agarose and conducted imaging (Fig. S4d). The deconvolution was realized by applying a Wiener filter on the raw data using EIR. The raw data, with and without deconvolutions, were used to reconstruct the images under the same parameters (Fig. S4e). Visibly, hollow regions, as a loss of low-frequency components, existed in the results of initial reconstruction and were practically recovered after EIR 1 deconvolved, while the recovery result was not satisfactory when applying EIR 2. Hence, EIR 1 was used as the system EIR to optimize the imaging of the whole-brain functional dynamics.

### Optical fluence compensation

The gain coefficients used for compensating optical fluence attenuation were estimated through straightforward experimental tests. First, we constructed a simple testing setup. We utilized two quartz plates ($50 \times 50$ mm², 2.5 mm thick), with the lower plate horizontally secured to a support stand and the upper plate horizontally affixed to a precision elevation stage, allowing precise control of the distance $h$ between them. The space between the quartz plates was used for placing the sample to be measured. The collimated laser output with a diameter of ~ 10 mm was split into two beams through a 1:9 beam splitter (Fig. S5). The 90% portion of the energy traveled vertically downward through the quartz plates and reached optical energy meter 1, while the remaining 10% continued to travel horizontally and reached optical energy meter 2, and the laser output was synchronized with data acquisition from the two optical energy meters (Fig. S5). The energy meters had a detection area of ~ 30 mm in diameter, which was sufficient to capture all the energy within the laser beam (Fig. S5). Prior to testing, without placing the sample, $h$ was adjusted to 4 mm, and the average readings of 50 laser pulses on optical energy meters 1 and 2 (denoted as $I_1$ and $I_2$, respectively) were recorded, which can be expressed as:

$$I_1 = \alpha \cdot I_2 \cdot \tau \quad (4)$$

where $\alpha$ represents the total light intensity conversion coefficient, and $\tau$ represents the light attenuation of the quartz glass.

Subsequently, we rapidly excised fresh organs from the experimental animals (selected from the same strain and age as the animals imaged in this study) for use as test samples. The upper quartz plate was elevated, the sample was inserted, and then the distance $h$ was readjusted to 4 mm, ensuring that the sample was compressed and fully occupied the interspace between the quartz plates. The procedure was repeated to obtain the readings from the two optical energy meters (denoted as $\acute{I}_1$ and $\acute{I}_2$, respectively), which can be expressed according to Lambert-Beer law[33]:

$$\acute{I}_1 = \alpha \cdot \acute{I}_2 \cdot \tau \cdot e^{-\mu_{\text{eff}} \cdot h} \quad (5)$$

By solving the two equations above, the optical attenuation coefficient of the sample can be calculated as:

$$\mu_{\text{eff}} = \frac{1}{h} \ln\left(\frac{\acute{I}_2 I_1}{\acute{I}_1 I_2}\right) \quad (6)$$

Using this method, we estimated $\mu_{\text{eff}}$ for the liver and brain. For each organ, measurements were taken from three animals and

averaged to obtain the results. For the brain tissue, the estimated $\mu_{eff}$ for 1064 nm and 800 nm laser are 0.065 mm$^{-1}$ and 0.077 mm$^{-1}$, respectively. These parameters were utilized for optical fluence compensation in imaging whole-brain anatomies and dynamic functions. For liver tissue, the $\mu_{eff}$ at 1064 nm and 690 nm were estimated to be 0.075 mm$^{-1}$ and 0.133 mm$^{-1}$, respectively. The coefficients were applied for compensation in dynamic liver imaging and whole-body imaging. In the image processing, depth compensation was applied as $e^{\mu_{eff} \cdot depth(mm)}$ for different wavelengths to enhance the PA amplitude from deep tissues, with depth = 0 at the surface. It should also be noted that the empirical testing here serves merely as a rough estimate for optical fluence compensation.

### 3D image reconstruction

The PA images were reconstructed by the universal back-projection (UBP)[64] algorithms implemented in MATLAB 2018 (computer configuration: Intel Xeon Silver 4210 R, 40 CPUs @ 2.4 GHz, RAM 384GB). Considering the transducers' central wavelength (475 μm for 3.16 MHz), the voxel size for all image reconstructions was set to 0.1 × 0.1 × 0.1 mm$^3$. To accelerate the image reconstruction, the program was run on the GPU (CUDA version: 11.6, Quadro RTX 8000, NVIDIA Co.). To mitigate the artifacts induced by the acoustic inhomogeneity for better imaging quality, we incorporated a dual-speed-of-sound algorithm into the reconstruction process. We first used a uniform sound speed (1500 m/s) for rough reconstruction reference. Then we segmented the boundary between the imaging object and the water environment and fitted it to a part of an ellipsoidal surface. The deionized water environment was maintained at a constant temperature of 25 °C with a uniform sound speed of 1496.9 m/s, which was used for the region outside the ellipsoid. The sound speeds set for the phantoms (1% agarose concentration), animal trunk, and brain imaging were 1496.9 m/s, 1535 m/s, and 1520 m/s, which was used for the region inside the ellipsoid, respectively. During the reconstruction, the coordinates of intersection points between the ellipsoidal surface and the lines connecting the transducer elements and the FOV voxels were analytically calculated. Then the time–of–flight was calculated in two segments for back-projection.

In the image post-processing, optical fluence compensation was applied to enhance the PA amplitude from the deep tissue. We then utilized the ellipsoidal surface from the dual-speed-of-sound UBP as the starting position of the compensatory. When imaging the anatomies, we further applied a Hessian-based Frangi vesselness filter to the images to enhance the contrast of blood vessels. For each image, the applied filter size ranged from 2 to 8 voxels, ensuring coverage of all vascular features that could be imaged. Then we added the self-normalized filtered images (with a weighting factor of 0.3) back to the self-normalized unfiltered images (with a weighting factor of 0.7) and generated the final images. This processing primarily involves Figs. 1b, 2a, 3, 5b, and 5d, 6b, c. This operation enhanced the anatomical clarity but concurrently changed the original PA amplitude. Hence, all the computational analysis using image reconstruction values was based on the original images without any postprocessing. To display the 3D image, the MIP images and depth-encoded images were generated by MATLAB 2018. In addition, we rendered and superimposed the functional information onto the anatomy background using the 3D visualization software, Amira 2019, for an intuitive display.

### Dual-wavelength unmixing

In brain imaging, the sO$_2$ and HbT maps were estimated with dual-wavelength images at 800 nm and 1064 nm. To correct the optical influence deposited on the surface, a tube filled with 50-time-diluted India ink was placed on the surface of the object and imaged together. Its absorption spectrum was measured in advance via a spectrophotometer (Cary 6000i UV-Vis-NIR, Agilent Technologies, Inc.). The

formula can be written as:

$$PA_0' = PA_0 \frac{Ab_T}{PA_T} \tag{7}$$

Where $PA_0$, $Ab_T$, $PA_T$, and $PA_0'$ denote the initial multispectral images of the object, the absorption spectrum of the ink tube, the PA amplitude averaged in a specific area of the ink tube, and the surface-fluence-corrected multispectral images of the object, respectively. It should be noted that dual-wavelength images require optical fluence compensation with the corresponding coefficients. Then the linear unmixing algorithm was applied, expressed as:

$$\begin{bmatrix} C_{HbO} \\ C_{Hb} \end{bmatrix} = [\, Ab_{HbO} \quad Ab_{Hb} \,]^{-1} PA_0' \tag{8}$$

Where $Ab_{HbO}$, $Ab_{Hb}$, $C_{HbO}$, and $C_{Hb}$ denote the absorption spectrum and the relative concentration of the oxygenated and deoxygenated hemoglobin, respectively. The formula was solved using the least square method with nonnegativity restrictions. Then the estimated maps were obtained as follows:

$$sO_2 = \frac{C_{HbO}}{C_{HbO} + C_{Hb}} \tag{9}$$

$$HBT = C_{HbO} + C_{Hb} \tag{10}$$

It is important to note that, although we have corrected the optical fluence on the object's surface, the deep-tissue optical fluence is still affected by scattering and absorption at different wavelengths and cannot be completely corrected. Therefore, the sO$_2$ and HBT values in this study should be considered as effective references rather than strictly accurate quantitative values.

### Temporal information extraction

In 25 Hz liver imaging, we first removed the DC component from the time-series raw signal corresponding to each voxel. Then the Fourier transform was performed on the signal. The heartbeat frequency was determined and a finite-impulse-response high-pass filter was applied to the signal. Then the heartbeat waveform could be extracted while keeping the phase constant. The arterial map was obtained by traversing all the voxels and extracting the amplitude near the heartbeat frequency. In mapping the phase of hepatic pulse waves, we calculated the phase angle $\varphi$ corresponding to the 3.9 Hz frequency component of the sinusoidal waveform for the time sequence of each voxel, with the phase angle at the proximal vascular location designated as the zero phase angle. The conversion from phase angle $\varphi$ to temporal phase delay $t_\varphi$ can be formulated as:

$$t_\varphi = \frac{\varphi}{2\pi f} \tag{11}$$

where $f = 3.9$ Hz. The calculated results were overlaid on the anatomical structure using a pseudocolor encoding.

In 10 Hz imaging of whole-brain neurovascular coupling response, we first imaged the brain over 3 min and averaged them to obtain the anatomy background and the reference sO$_2$ map, which was performed after the electrodes were inserted to avoid signal changes caused by the pain of needling. Then the CHRF curve was created by convoluting the pulse train with the HRF curve and was normalized. After the experiment, the image sequence was reconstructed and the intensity-based automatic image registration algorithm (in Image Processing Toolbox of MATLAB 2018) was applied onto the image sequence with the anatomy background as a reference to eliminate voxel shifts caused by minor movements of the animal. Then the sO$_2$

map of the whole brain was calculated frame by frame. Based on the $sO_2$ map sequence, the correlation coefficient was solved between CHRF and the $sO_2$ sequence voxel by voxel. The voxels with a result greater than 0.95 were recognized as the response region evoked by the stimuli. Within the response region, the percentage change in $sO_2$ for each frame was calculated relative to the baseline, and a moving average filter was then applied to the curve for further denoising.

In whole-body probe tracking, a total of 120 time-point three-dimensional (3D) image sequences were acquired with an interval of 5 s, and each time-point image was reconstructed using the STINT method. In Fig. 6h, the 3D image sequence was first subjected to frame-by-frame rigid registration. Taking the image before administration as $Img_{ref}$, the 3D mapping of relative enhancement $RE(t)$ of the image acquired at time $t$ $Img(t)$ can be calculated by processing the sequence of each voxel:

$$RE(t) = \frac{Img(t) - Img_{ref}}{Img_{ref}} \times 100\% \qquad (12)$$

And voxels with RE value below 10% were considered to be affected by noise and were set to zero. In Fig. 6e–g, fitting of the raw data of enhancement was performed using two empirical models to intuitively analyze the probe metabolism dynamics across different organs. For changes that initially rose rapidly and then declined gradually (the iliac artery, the heart and the bladder), the fitted relative enhancement $FRE(t)$ was generated using the following model[82]:

$$FRE(t) = \begin{cases} 0 & 0 \le t \le t_0 \\ A\left[1 - e^{-\alpha(t-t_0)}\right]^q \cdot e^{-\beta(t-t_0)} & t_0 \le t \end{cases} \qquad (13)$$

The change of enhancement from an initial rise to a steady state within the time window (the liver, the mesenteric artery, the intestine and the bladder) was modeled using an alternative model[83]:

$$FRE(t) = \begin{cases} 0 & 0 \le t \le t_0 \\ \frac{A}{1 + B \cdot e^{-\alpha(t-t_0)}} & t_0 \le t \end{cases} \qquad (14)$$

For each organ, the parameters $C_m$ and $t_p$ can be determined by $C_m = \max(FRE)$ and $FRE(t_p) = C_m$. The fitting procedure was performed using the built-in fitting toolbox in MATLAB 2018.

## Animal preparation

A total of 11 mice were enrolled for all studies (3 for whole-body imaging studies, 3 for liver imaging studies, 2 for brain imaging studies, and 3 for the probe tracking studies). No sample size calculation was performed. The number of animals used was determined to sufficiently demonstrate the system performance in 3D imaging and technology applications. All animal experiments were performed using the principles outlined in the Guide for the Care and Use of Laboratory Animals of Hangzhou Medical College. All animal studies and experimental protocols were approved by Institutional Animal Care and Use Committee (IACUC) at Hangzhou Medical College (Protocol No. SYXK2019-0011). All mice were housed under standardized conditions with a 12-h light/dark cycle (07:00–19:00 light phase, <5 lux intensity during active period), ambient temperature maintained at $22 \pm 1\,°C$ via automated heating ventilation air conditioning system, and relative humidity controlled at $50 \pm 5\%$ with continuous monitoring. Environmental parameters were stabilized for 7 days prior to each experiments.

For trunk imaging, adult, 10-week-old mice (NU/NU, nude mouse, Charles River Co.) were used. To avoid the influence of unwanted signals, the mouse underwent a 1-day fasting period before the imaging test, and the digestive residue was cleared. A small amount of

depilatory cream was applied to remove the fuzz on the skin. 1.5% vaporized isoflurane was ventilated as an anesthetic during the imaging.

For whole-brain imaging, 12-week-old mice (NU/NU, nude mouse, Charles River Co.) were used. To achieve superior deep-brain imaging results, a skull-removed operation was performed on all mice, which reduced the distortion and attenuation of the PA signal. We removed a portion of the parietal bone with a diameter of ~ 8 mm and sutured the scalp (detailed operation steps were introduced in ref. 39). To avoid overpressure, we carefully removed only the skull without causing any damage to the dura mater. We also employed low-dose mannitol to further alleviate brain swelling that may be induced by static over-pressure. During the recovery period from surgery to imaging testing, mannitol was administered every 24 h at a dose of 0.25 g/kg body weight. The injection volume of the solution was 0.1 mL/g body weight, and the injection rate was controlled at 5 mL/min. The postoperative mice were reared normally and observed for 10 days, during which time the wounds were completely healed and the animals behaved in the course of nature while gaining weight. When imaging the functional dynamics of SNP administration, SNP (Hangzhou Xiaoyou Biotechnology Co. Ltd) was dissolved in saline and reserved before experiments (concentration: $300\,\mu g\,mL^{-1}$, stored in dark place at $4\,°C$). A retention syringe was adopted for SNP administration via the caudal vein over 180 s (dosage: $25\,mg\,kg^{-1}$ bodyweight) while the animal was anesthetized with 1.2% vaporized isoflurane on the holder and able to be imaged simultaneously. Then the whole mouse brain was constantly imaged for 20 min to record the changes. When monitoring the stimulation-evoked brain region response, the mouse was kept mildly anesthetized using 0.8% vaporized isoflurane for a stronger signal change. Two silver electrodes (positive and negative) were led out from an electric pulse stimulator (YLS-9A, Yiyan Tech Co. Ltd) and embedded under the skin of the forelimbs. The pulse train was applied continuously for 10 s (period: 250 ms, width: 1 ms, voltage: +25 V, current limit: 1 mA) and the monitoring started 2 s ahead of the pulse output and lasted for 30 s, counting from the pulse output moment. These procedures were carried out first on the left limb and then the right limb, with an intermission of 5 min. The mouse was unscathed and behaved normally after the imaging tests.

For whole-body probe tracking, 12-week-old mice (NU/NU, nude mouse, Charles River Co.) were used. A1094 (JHE0001, Suzhou Zhuoxinya Technology Co., Ltd., China) was selected as the small-molecule tracer. To prepare an A1094 solution, 0.3 mg of solid A1094 was first dissolved in $20\,\mu L$ of dimethyl sulfoxide (DMSO). Subsequently, 7 mL of distilled water was added, and the mixture was subjected to ultrasonication for 5 min to ensure complete dissolution. The injection dose of A1094 was 1 μmol per kilogram of body weight, and calculations indicated that approximately 0.4 mL of the prepared solution was required for injection. During the experiment, the animals were maintained under light anesthesia with 1.2% isoflurane, and a 29 G indwelling tail vein catheter was connected to a syringe. The injection was completed within 90 seconds using a precision syringe pump. The mice behaved normally after the probe tracking tests.

## Reporting summary

Further information on research design is available in the Nature Portfolio Reporting Summary linked to this article.

## Data availability

The main data supporting the results of this study are available within the paper and its Supplementary Information. Imaging data for in-vivo experiments and the plot source files are available in a publicly accessible repository: https://figshare.com/articles/dataset/3D-PanoPACT_Data_Code/28443374. Some of the data are too large to

be publicly shared, yet they are available for research purposes from the corresponding authors on request.

## Code availability

The image reconstruction and processing algorithms are described in detail in Methods. The necessary pseudocode has been provided in the Supplementary Methods. Image reconstruction and processing programs are available in the repository: https://figshare.com/articles/dataset/3D-PanoPACT_Data_Code/28443374.

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

## Acknowledgements

This study was sponsored by Key R&D Program of Zhejiang (2024SSYS0014), Zhejiang Provincial Key Research and Development Program (2021C03052) and Key Research Project of Zhejiang Lab (2020MC0AD02), all awarded to J. S. This work was also partially supported by National Institutes of Health grants U01 EB029823 (BRAIN3 Initiative), R35 CA220436 (Outstanding Investigator Award), and R01 EB028277 (PATH), all awarded to L. V. W. Additionally, we are grateful to Dr. Yiming Bi for the technical support and expertise in conducting the whole-body small-molecule probe tracking experiments, which significantly enhanced the quality of our data.

## Author contributions

L. V. W. and J. S. conceived and designed the study. X. W., K. W. and D. R. constructed the hardware system. Y. W. and S. W. developed the control program. C. L. and R. C. designed the in-vivo functional and molecular imaging schemes. X. W., M. S. and X. G. performed all the experiments. Y. Y. made the phantoms for imaging test. R. W. developed the reconstruction algorithm. Y. M. and X. W. analyzed the data. J. S. supervised the study. All authors discussed the results and contributed to writing the manuscript.

## Competing interests

L. V. W. has a financial interest in Microphotoacoustics, Inc., CalPACT, LLC and Union Photoacoustic Technologies, Ltd., which, however, did not support this work. The other authors declare no conflict of interest.
