## [Transparent Peer Review file · Nature Communications]

Cross-regional real-time visualization of systemic physiology and dynamics with 3D panoramic photoacoustic computed tomography (3D-PanoPACT)

Corresponding Author: Professor Junhui Shi

Version 0:

Reviewer comments:

Reviewer #1

(Remarks to the Author)

Wang X. and colleagues present an impressive 3D, video-rate photoacoustic tomography (PAT) system, achieving whole-organ and whole-body imaging in vivo with a spatial resolution of $\sim 200 \mu\text{m}$. The integration of a hemispheric piezoelectric probe with 1024 elements marks a significant leap forward in high-speed, high-resolution imaging for large-scale biological applications. This is a remarkable technical achievement that deserves to be highlighted.

While the technical innovations behind this system are substantial, many critical details are relegated to the supplementary materials, where their importance may be overlooked. We believe that incorporating more of these technical insights into the main text would greatly enhance the manuscript's clarity and impact, helping readers fully appreciate the advancements and performance improvements introduced in this work. In our opinion, the current version of the manuscript mainly highlights the performances of the imaging system, and one must dig quite deep to find out how this was achieved, which will not be so easy for a non-specialist audience.

Key Innovations and Performance Enhancements:

* Several technical contributions stand out but warrant further emphasis. The calibration of element positions and fluence, for instance, is briefly mentioned, yet this process likely plays a crucial role in ensuring the high image quality achieved. Better highlighting this calibration in the main text and referencing related prior works would provide readers with valuable context.

Suggested references:

Jiang, Yuting et al. "Spatially distributed sensor array calibration for photoacoustic imaging." (2021).

Oeri, Milan et al. "Analytical calibration of linear transducer arrays for photoacoustic tomography." European Conference on Biomedical Optics (2015).

Sahlström, Teemu et al. "Modeling of Errors Due to Uncertainties in Ultrasound Sensor Locations in Photoacoustic Tomography." IEEE Transactions on Medical Imaging 39 (2020): 2140-2150.

* Similarly, the introduction of the STINT method—slight rotation along the hemisphere axis to synthetically increase element density and reduce spatial aliasing—is an important innovation, but it is somewhat underexplained. Please clarify that this has already been used in ref 23 (Lin et al 2021), as the work is cited but the acronym is not used in it, or explain to which extent the method used here differs. Providing quantitative metrics of its effectiveness would also strengthen its presentation.

* The system's capacity for trunk imaging via XY axis translation reflects careful attention to motion correction which significantly enhance image quality while maintaining a high temporal resolution. These details, along with impulse response correction, should be emphasized further in the main text. Performance-wise, the ability to image entire organs or whole bodies at high speed and resolution is a significant achievement. However, the factors contributing to the system's high frame rates, superior signal-to-noise ratio (SNR), and impressive imaging depth should be more clearly highlighted in the main text, particularly in comparison to other systems in Supplementary Table 1.

* Is there any improvement due to the choice of sensor element, in terms of sensitivity? How does it compare with piezoelectric elements used in other works? Did the authors characterize and if so attempted to minimize the dispersion of

sensitivities and bandwidths across the elements?

* I.106-107: What is the role of the phase plate connecting the preamplifier and DAQ system?

* The two laser sources have a 10 Hz repetition rate. Please explain how a 25Hz single color frame rate is achieved.

* Please explain how the laser pulses are delayed by 20us in the the dual color approach.

Liver imaging:

* I.146-147: 'positioning the ROI'

Please clarify whether this means centering with respect to the hemispheric sensor.

* Heartbeat is usually not a sine, but shows several segments that corresponds to the successive opening, closing and contractions of the heart sub-structures. The sine wave extraction has a limited physiological relevance in itself, although the phase delay is interesting.

Did the authors to extract more detailed information (for instance related to distole and systole), using a more complex filter?

* Motion artifacts may be influencing the phase delay shown in Fig.2d, how did the authors ruled this out?

* As this phase can be extracted for each voxel of the reconstruction volume, a phase map similar to Fig.2e could be generated and might reveal a phase gradient related to the blood pressure wave mentioned in I.160.

Brain Imaging:

* The term "cranial window" is used ambiguously, referring either to a pure craniotomy or to procedures where the skull is replaced with an ultrasound-transparent cover, as seen in Ref 37. To avoid confusion, it is crucial to clarify that the skull was removed without replacement in this study. Additionally, specifying in the main text which part of the skull was removed would improve understanding, especially when comparing this system's results to others like the FONT 2 system by Razansky's group (ref 4 in Supp Inf, 63 in main text), which works with an intact scalp and skull. Supplementary Table 1 should be adjusted to reflect this.

* A major concern with such a large craniotomy is the static overpressure inducing brain swelling, which can impact imaging quality and strongly limits physiological relevance of the acquired images. This physiological effect is not sufficiently addressed in the paper, please provide details on how the authors managed this challenge.

* To which extent does the reduced frame rate induce some smearing of the final image due to motion? This can be quantified by imaging at 25Hz with one single excitation wavelength.

Other comments:

* Figures could be made more readable with minor adjustments, such as clearly marking the ROI in Figure 1b, adding a grayscale colormap in Figure 2e, and labeling color-coded brain regions in Figure 3a. Scale bars in Fig.3g would be beneficial to better grasp the depth of the circle of Willis.

* In Supplementary Video 5: it seems left and right images (gray scale and depth color coded) are desynchronized at some point. Please correct or explain why this happens if relevant.

* I. 501-502: images were reconstructed for each set of 1024 signals at a given angle and them summed ('superimposed' in the text). Why not reconstructing directly from the 10×1024 signals from all virtual elements?

* sup inf, I.89: 'after fabrication' Did the authors mean 'after calibration'?

In summary, Wang et al. have made a groundbreaking contribution to the field of photoacoustic tomography, particularly in terms of speed, resolution and FOV. By emphasizing key technical details in the main text and clarifying certain physiological and methodological aspects, the manuscript would further enhance its impact and provide readers with a more comprehensive understanding of the system's capabilities. We commend the authors for their innovative work and recommend for publication once the above comments are addressed.

Reviewer #2

(Remarks to the Author)

The authors demonstrate a new 3D panoramic photoacoustic computed tomography (PACT) system for real-time, cross regional physiological dynamic imaging of the system. This study achieves high-quality images with 3D photoacoustic tomography, especially in the detailed visualization of vascular and organ structures, which is technically commendable. Although this study demonstrates the advantages of 3D PanoPACT in high spatiotemporal resolution and wide field of view (FOV), there are also some noteworthy issues and comments:

1. Despite the high imaging quality, the paper should focus more on specific applications of the technology. Readers are looking not just for clear images but also for how this technology can solve real biological or physiological questions. Enhancing the focus on practical applications would increase the relevance and value of this work.

2. The authors claim that 3D-PanoPACT enables “cross-regional real-time visualization of systemic physiology and dynamics from organ to whole-body level” and emphasizes its “dynamic imaging with unprecedented quality of the whole trunk” to provide insights into multi-organ physiology (Abstract). To fully support these claims, it’s necessary to include an experiment tracking a small-molecule probe’s metabolic pathway in vivo. Without this, it’s challenging to see how this approach offers an advantage over established methods like PET, MRI, or fluorescence imaging for functional, whole-body tracking. This application would highlight the unique strengths of 3D photoacoustic imaging and its relevance to biological research.

3. The paper demonstrates both whole-body and brain functional imaging of the mouse, but the rationale for combining these isn’t entirely clear. Is this meant to showcase versatility, or is there a specific scientific question being addressed by both examples? A brief explanation of the purpose behind this combination would help readers better understand the study’s overall objective.

4. The limitations of this system are not discussed in the paper. Adding some discussion about the imaging depth limits, signal-to-noise considerations, or any areas where resolution might be affected would provide a balanced perspective and inform future improvements.

5. In line 455, the manuscript describes the application of depth compensation during image post-processing: “In the image post-processing, a depth compensation was applied to enhance the PA amplitude from the deep tissue ($e^{7.5 \times \text{depth}}$ mm for trunk imaging; $e^{6.5 \times \text{depth}}$ mm for brain imaging).” Could the authors clarify how the gain coefficients ($e^{7.5}$ and $e^{6.5}$) were determined for depth compensation? Specifically, it would be helpful to understand the methodology used to derive these values and whether they are empirically or theoretically based.

6. How long does it take to acquire a whole-body image? I think the slow imaging speed may constrain the real scenarios.

7. In Figure 5d, sections 2, 3, and 4 don’t seem to match those in Figure 5e (1, 2, 3). Could the authors check for a labeling error?

Reviewer #3

(Remarks to the Author)

Version 1:

Reviewer comments:

Reviewer #1

(Remarks to the Author)

Overall, Wang and colleagues addressed our concerns, adding a lot of new and useful material. They clearly put a lot of effort and rigour into it, and they should be praised for that.

We would like though to highlight a few points:

- The so-called STINT method actually encompasses 3 different techniques that are used in quite different experimental contexts.

Upon our request, the authors now describe these three techniques in detail, making it much clearer for the readers. It appears though that this could be enough material for separate papers.

While the manuscript is now much clearer, we are uncertain whether it still aligns with the format of Nature Communications. As this is primarily a matter of editorial scope rather than scientific content, we defer the decision to the editor.

- One of the STINT implementation refers to respiratory gating. Similar works have been published already, and proper references should be cited.

- Similarly, electrical impulse deconvolution has already been performed for PA imaging, and proper references should be cited as well.

In conclusion, we do recommend for publication in Nature Communications, provided it fits the length standards.

(Remarks on code availability)

Reviewer #2

(Remarks to the Author)

1. While 3D-PanoPACT demonstrates excellent temporal resolution and generally satisfactory tracking outcomes for small-molecule probes, its imaging resolution has significantly decreased overall, rather than the resolution degradation being limited to the edges as described in the results section. Although this resolution compromise might be an acceptable trade-off for rapid monitoring, its potential impact on result validity warrants further investigation. To strengthen this methodology,

we recommend acquiring high-resolution data from specific metabolic organs at critical timepoints using smaller FOVs. If these targeted measurements align with the metabolic trends observed through rapid whole-body monitoring, it would compellingly demonstrate that controlled resolution reduction does not compromise monitoring accuracy. Additionally, providing a supplemental video showing the dynamic metabolic progression of molecular probes would significantly enhance data visualization and methodological transparency.

2. To better highlight this technique's breakthroughs in whole-body metabolic monitoring and disease research, citing relevant prior studies in photoacoustics is recommended:

Radiology, 2021, 300(1): 89-97.

(Remarks on code availability)

Reviewer #3

(Remarks to the Author)

(Remarks on code availability)

Version 2:

Reviewer comments:

Reviewer #2

(Remarks to the Author)

I am satisfied with the revisions. OK to publish.

(Remarks on code availability)

----- Author Response Letter -----

We sincerely appreciate the constructive comments from all the reviewers. We believe that all raised concerns have been addressed through the performance of additional experiments, the incorporation of new data and figures, and the implementation of improvements based on the comments. Our point-by-point responses to the reviewers' comments are provided below. The original reviewers' comments are indicated in black, our responses are provided in blue, and the revisions in the manuscript are marked in red. A highlighted version of the manuscript with revisions marked with a yellow background is also provided in the submission.

Reviewer #1:

Wang X. and colleagues present an impressive 3D, video-rate photoacoustic tomography (PAT) system, achieving whole-organ and whole-body imaging in vivo with a spatial resolution of ~200 μm . The integration of a hemispheric piezoelectric probe with 1024 elements marks a significant leap forward in high-speed, high-resolution imaging for large-scale biological applications. This is a remarkable technical achievement that deserves to be highlighted.

While the technical innovations behind this system are substantial, many critical details are relegated to the supplementary materials, where their importance may be overlooked. We believe that incorporating more of these technical insights into the main text would greatly enhance the manuscript's clarity and impact, helping readers fully appreciate the advancements and performance improvements introduced in this work. In our opinion, the current version of the manuscript mainly highlights the performances of the imaging system, and one must dig quite deep to find out how this was achieved, which will not be so easy for a non-specialist audience.

Reply:

Thank you for your kind and positive remarks. We appreciate your support.

In response to the insightful suggestions from the other reviewers, particularly the request to further illustrate the practical value of 3D-PanoPACT in biological applications, we have added a part titled "High-spatiotemporal-resolution tracking of small molecule metabolic pathways at whole-body scale" and Fig. 6 in the Results section, demonstrating the system's ability to track the metabolic pathways of the small molecule A1094 with high spatiotemporal resolution across the whole body. This addition directly addresses the feedback received, showcasing the system's capacity for detailed, whole-body imaging. By using an engineered diffuser with a large divergence angle and a high-energy laser, we achieve uniform illumination and a high signal-to-noise ratio (SNR), enabling whole-body imaging using the spatiotemporal-integration method without array translations. The array rotates 15° within 5 s to capture 50 frames, allowing 12 imaging sessions per minute, equivalent to a frame rate of 0.2 Hz. This approach minimizes vibrations and maintains high spatial resolution, even at the periphery of the imaging field of view (FOV). Quantitative analysis of A1094's distribution reveals its metabolic route through key organs, demonstrating the system's potential for whole-body biological studies, significantly enhancing our understanding of small molecule dynamics in vivo. The dynamic process can be more intuitively observed through Movie S6. The addition of this content will help readers better understand the effectiveness of the STINT method and the significant practical value of 3D-PanoPACT in studying biological problems. Additionally, we have also supplemented technical details in the main text to enhance the readability of the paper.

Key Innovations and Performance Enhancements:

* Several technical contributions stand out but warrant further emphasis. The calibration of element positions and fluence, for instance, is briefly mentioned, yet this process likely plays a crucial role in ensuring the high image quality achieved. Better highlighting this calibration in the main text and referencing related prior works would provide readers with valuable context.

Suggested references:

Jiang, Yuting et al. "Spatially distributed sensor array calibration for photoacoustic imaging." (2021).

Oeri, Milan et al. "Analytical calibration of linear transducer arrays for photoacoustic tomography." European Conference on Biomedical Optics (2015).

Sahlström, Teemu et al. "Modeling of Errors Due to Uncertainties in Ultrasound Sensor Locations in Photoacoustic Tomography." *IEEE Transactions on Medical Imaging* 39 (2020): 2140-2150.

Reply:

Thanks for the insightful comments. In the main text, we have emphasized the technical details of element position calibration and optical fluence compensation. In the first part of the Results section, we have supplemented the principles, procedures, and comparative effects of element position calibration, and revised Fig. S3 accordingly. We have included the suggested references that explored similar calibration techniques, which provided readers with valuable context and acknowledged the contributions of previous research in this area. We have also described the experimental setup, testing principles, and results for calculating the optical fluence compensation coefficients, and included Fig. S5. Given the addition of technical details concerning systematic calibration, we have renamed the subsection of this part to "Development and calibration of 3D-PanoPACT". Furthermore, we have provided comprehensive and formulaic descriptions of these technical implementations in the Methods section to offer readers a more intuitive presentation of the details. With these revisions, the manuscript highlighted the technical advancements of 3D-PanoPACT. We believe readers will be able to quickly grasp the technical contributions of our work.

Revisions:

Results section:

Lines 111:

"Development and calibration of 3D-PanoPACT (title)"

Lines 157-177:

"In 3D-PanoPACT, the deviation of the detection coordinates is influenced by both the element position deviations of the hemispherical array itself and the deviations of the rotation axis, which can affect the quality of image reconstruction^{25, 26}. To calibrate the spatial position of each element, we used a point absorber (50 μm diameter black polystyrene microspheres) embedded in an agarose phantom. The array housing, machined with high precision for an exact 20 cm diameter and ideal Fibonacci grid distribution, still introduced radial deviations due to individual element insertion, necessitating calibration. The calibration procedure involved three steps (see Methods for details). First, we centered the FOV at the origin and adjusted the point absorber's position while repeatedly imaging it at 690 nm until it was centered in the field with PA raw data acquired (Fig. S3a). Second, we measured the water temperature with a high-precision thermocouple and calculated the speed of sound in the medium using the classic temperature-velocity formula²⁷. Third, we positioned the signals from the transducer surface and the point absorber in the raw data, identified the sample points where the PA signals peaked (Fig. S3b), and calculated the radial deviation for each element using the travel time and acoustic velocity²⁸ (see Methods for details). The corrected positions of the elements were then determined in Cartesian coordinate with the radial deviations kept within

± 0.5 mm (Fig. S3c). We then calibrated the system rotation axis due to assembly errors. To correct this, we positioned a point absorber 15 mm from the origin and imaged it while rotating the array by 70° . By co-reconstructing 15 evenly-spaced frames, we formed a 3D image where the center of the reconstructed points clearly defined a circle (Fig. S3d). The normal through the center was identified as the calibrated rotation axis of the system, allowing us to correct the misalignment and improve image quality. The calibration revealed that the actual rotation axis was parallel to the ideal axis but shifted by 0.07 mm along the X-axis and 0.29 mm along the Y-axis (Fig. S3d). Applying these calibrations, we observed improvements in resolution and contrast, particularly in regions away from the center of the FOV (Fig. S3e, f).”

Lines 187-199:

“In PA imaging, accurate optical fluence compensation is crucial for enhancing image fidelity and ensuring reliable quantitative analysis²⁹, especially for multi-wavelength unmixing. We employed empirical testing to estimate optical attenuation in freshly excised tissue for subsequent compensation. A simple setup was first constructed using two quartz plates, a collimated laser beam, a 1:9 beam splitter, and a pair of optical energy meters. One optical energy meter was used to detect the light passing through the quartz plates, while the other was used to monitor the energy fluctuations of the split beam as a normalization reference (Fig. S5). By comparing these measurements with and without a sample, we calculated the optical attenuation coefficient (μ_{eff}) for brain and liver tissues at different wavelengths based on the Lambert-Beer law³⁰, which relates optical attenuation to the properties of the material (see Methods for details). For brain tissue, μ_{eff} was 0.065 mm^{-1} at 1064 nm and 0.077 mm^{-1} at 800 nm, which were utilized for imaging whole-brain anatomies and dynamic functions. For liver tissue, μ_{eff} was 0.075 mm^{-1} at 1064 nm and 0.133 mm^{-1} at 690 nm, which were applied in dynamic liver imaging and whole-body imaging. These coefficients were then used in image processing to apply exponential depth compensation, enhancing the PA amplitude from deep tissues.”

Methods section:

Lines 857-890:

“Optical fluence compensation (title)

The gain coefficients used for compensating optical fluence attenuation were estimated through straightforward experimental tests. First, we constructed a simple testing setup. We utilized two quartz plates ($50 \text{ mm} \times 50 \text{ mm}^2$, 2.5 mm thick), with the lower plate horizontally secured to a support stand and the upper plate horizontally affixed to a precision elevation stage, allowing precise control of the distance h between them. The space between the quartz plates was used for placing the sample to be measured. The collimated laser output with a diameter of ~ 10 mm was split into two beams through a 1:9 beam splitter (Fig. S5). The 90% portion of the energy traveled vertically downward through the quartz plates and reached optical energy meter 1, while the remaining 10% continued to travel horizontally and reached optical energy meter 2, and the laser output was synchronized with data acquisition from the two optical energy meters (Fig. S5). The energy meters had a detection area of ~ 30 mm in diameter, which was sufficient to capture all the energy within the laser beam (Fig. S5). Prior to testing, without placing the sample, h was adjusted to 4 mm, and the average readings of 50 laser pulses on optical energy meters 1 and 2 (denoted as I_1 and I_2 , respectively) were recorded, which can be expressed as:

$$I_1 = \alpha \cdot I_2 \cdot \tau \quad (4)$$

where α represents the total light intensity conversion coefficient, and τ represents the light attenuation of the quartz glass.

Subsequently, we rapidly excised fresh organs from the experimental animals (selected from the same strain and age as the animals imaged in this study) for use as test samples. The upper quartz plate was elevated, the

sample was inserted, and then the distance h was readjusted to 4 mm, ensuring that the sample was compressed and fully occupied the interspace between the quartz plates. The procedure was repeated to obtain the readings from the two optical energy meters (denoted as I_1 and I_2 , respectively), which can be expressed according to Lambert-Beer law³⁰:

$$I_1 = \alpha \cdot I_2 \cdot \tau \cdot e^{-\mu_{\text{eff}} \cdot h} \quad (5)$$

By solving the two equations above, the optical attenuation coefficient of the sample can be calculated as:

$$\mu_{\text{eff}} = \frac{1}{h} \ln\left(\frac{I_2 I_1}{I_1 I_2}\right) \quad (6)$$

Using this method, we estimated μ_{eff} for the liver and brain. For each organ, measurements were taken from three animals and averaged to obtain the results. For the brain tissue, the estimated μ_{eff} for 1064 nm and 800 nm laser are 0.065 mm^{-1} and 0.077 mm^{-1} , respectively. These parameters were utilized for optical fluence compensation in imaging whole-brain anatomies and dynamic functions. For liver tissue, the μ_{eff} at 1064 nm and 690 nm were estimated to be 0.075 mm^{-1} and 0.133 mm^{-1} , respectively. The coefficients were applied for compensation in dynamic liver imaging and whole-body imaging. In the image processing, depth compensation was applied as $e^{\mu_{\text{eff}} \cdot \text{depth}(\text{mm})}$ for different wavelengths to enhance the PA amplitude from deep tissues, with depth = 0 at the surface. It should also be noted that the empirical testing here serves merely as a rough estimate for optical fluence compensation.”

Supplementary Materials:

Fig. S3, Fig. S5 (please refer to the end of this file to prevent confusion of figure numbers)

References:

Ref. 25-30:

25. Sahlström, T., Pulkkinen, A., Tick, J., Leskinen, J. & Tarvainen, T. Modeling of errors due to uncertainties in ultrasound sensor locations in photoacoustic tomography. *IEEE Transactions on Medical Imaging* 39, 2140-2150 (2020).
26. Oeri, M., Bost, W. & Fournelle, M. in *European Conference on Biomedical Optics* 95390P (Optica Publishing Group, 2015).
27. Bilaniuk, N. & Wong, G.S. Speed of sound in pure water as a function of temperature. *The Journal of the Acoustical Society of America* 93, 1609-1612 (1993).
28. Jiang, Y. et al. in *Photons Plus Ultrasound: Imaging and Sensing* 2021, Vol. 11642 95-101 (SPIE, 2021).
29. Bu, S. et al. Model-based reconstruction integrated with fluence compensation for photoacoustic tomography. *IEEE transactions on biomedical engineering* 59, 1354-1363 (2012).
30. Oshina, I. & Spigulis, J. Beer–Lambert law for optical tissue diagnostics: current state of the art and the main limitations. *Journal of biomedical optics* 26, 100901-100901 (2021).

* Similarly, the introduction of the STINT method—slight rotation along the hemisphere axis to synthetically increase element density and reduce spatial aliasing—is an important innovation, but it is somewhat underexplained. Please clarify that this has already been used in ref 23 (Lin et al 2021), as the work is cited but the acronym is not used in it, or explain to which extent the method used here differs. Providing quantitative metrics of its effectiveness would also strengthen its presentation.

Reply:

Thank you for your insightful comments on our manuscript. We have addressed the concerns as follows:

1. Clarification on the STINT method. The STINT method is an important innovation and a key step in achieving large-FOV and high-spatiotemporal-resolution imaging, but the detailed implementation of

this method was not included in the main text, which may cause confusion to readers. Therefore, we have added a part with details titled “The STINT Method” in the last part of the Results section and included Fig. R1 (Fig. 7 in the revised manuscript). The content encompasses the implementation logic of the STINT method, its timing design, data processing strategies tailored to diverse imaging requirements, and validation of its efficacy. In the revised main text, in addition to providing rich technical details, we also emphasized how specific imaging results relate to the implementation methods. For instance, Fig. 3, 4, and 6 correspond to the first implementation of the STINT method, while Fig. 5 corresponds to the second implementation approach. We also added necessary Supplementary Methods in the Supplementary Materials for clarification. In this way, readers will not only be able to understand the performance of 3D-PanoPACT but also clearly correspond to the key technical details.

2. Comparison with 3D-PACT (Lin et al 2021). 3D-PACT employs four 1/4 arc arrays to receive photoacoustic (PA) signals, achieving 3D imaging by rotating 90° around the central axis. In contrast, the 3D-PanoPACT detection array consists of 1024 elements arranged according to a Fibonacci grid, completing 3D imaging by rotating a very small angle, which demonstrates several advantages:

- 1) Benefiting from the angular isotropic design of the Fibonacci grid, 3D-PanoPACT can achieve high-quality 3D imaging within a certain FOV without any additional motion (Fig. 2). The imaging frame rate is determined by the laser repetition rate, while 3D-PACT must rely on rotation to achieve 3D imaging. This is because its array distribution does not possess the characteristic of 3D spatial isotropy. This difference limits the ability of 3D-PACT to capture high-speed biological processes, while 3D-PanoPACT is capable of this task.

- 2) 3D-PACT requires a minimum rotation of 90° to cover the 2π solid angle in spatial frequency, which corresponds to a large angular velocity of 0.785 rad/s when performing rapid imaging (such as the mentioned 0.5 Hz). With such a large mechanical load, the high-speed rotation inevitably induces vibrations, causing random deviations of the element positions, thereby degrading spatial resolution. Moreover, this phenomenon becomes increasingly pronounced with the increase of the transducer's center frequency. In 3D-PanoPACT, a small rotation angle suffices to achieve uniform spatial upsampling, markedly reducing random errors and ensuring spatial resolution. Assuming a rotation of 15° in 3D-PanoPACT, the angular velocity is 6 times slower than that of 3D-PACT for the same imaging frame rate, which can significantly reduce vibrations.

- 3) In 3D-PanoPACT, the essence of using the STINT method is the spatial upsampling process, which involves trading off temporal resolution for the imaging FOV. In other words, even without the STINT method, 3D-PanoPACT already possesses an isotropic 3D-FOV (Fig. 2), and, the STINT method further achieves spatially scalable imaging (Fig. 3-6) by integrating multiple temporal data. In contrast, the array rotation in 3D-PACT is more akin to a scanning method (e.g., using linear array translation for 3D image reconstruction¹), and the system itself does not possess an isotropic 3D FOV. This leads to some imaging that can be achieved in 3D-PanoPACT being unattainable in 3D-PACT (Fig. 2 and 5).

Therefore, even though both involve the rotation of the array, the implementation logic of 3D-PanoPACT is significantly different from that of 3D-PACT, and the STINT method is more suitable for describing the imaging approach of 3D-PanoPACT. We have provided explanations in the Discussion section for clarification.

Fig. R1 (Fig. 7 in the revised manuscript). The spatiotemporal-integration (STINT) method.

3. The effectiveness of the STINT method. To demonstrate this, we imaged a complete leaf vein at 690 nm with an FOV diameter of 60 mm. In the single-pulse imaging results, the outline of the leaf and the larger vein branches can be observed, indicating that 3D-PanoPACT has near-isotropic 3D PA imaging capabilities without motion, as evidenced by the maximum intensity projections (MIP) in the X-Z and

X-Y planes (Fig. R2a). By integrating 20 sets of data within 2 s using the STINT method, 3D-PanoPACT reconstructs images that reveal rich details, with significantly reduced artifacts and markedly enhanced imaging fidelity (Fig. R2b). We first provide a feature correlation coefficient (FCC) at the same image location before and after using the STINT method as a metric to quantify the difference in image features. It can be observed that in the main vein, the FCC value is ~ 0.97 in the X-Y plane (Fig. R2c) and ~ 0.98 (Fig. R2d) in the X-Z plane, indicating that the STINT method does not introduce additional image blurring or artifacts. At the edge branches of the leaf veins, the FCC value is 0.39, indicating that the STINT method can indeed reveal detailed features that are almost impossible to reconstruct in single-pulse imaging (Fig. R2e). We further calculate the standard deviation (STD) as a metric of artifact suppression. We find that the STD value at the edge of FOV is reduced by 8 times after using the STINT method, indicating that artifacts are significantly weakened, and this effect could be further enhanced by integrating more data for reconstruction (Fig. R2f). We believe these metrics clearly demonstrate the effectiveness of the STINT method. Accordingly, we have incorporated these contents into the Results section and supplemented Fig. S9.

Fig. R2 (Fig. S9 in the revised manuscript). Quantification of the STINT method's effectiveness. **a** The single-pulse 3D image results of a complete leaf vein. **b** The results with the STINT method. **c-f** Profile comparison, where the single-pulse image is represented by dashed lines, and the results obtained using the STINT method are indicated by solid lines. Colors correspond to the arrow positions in a. FCC, feature correlation coefficient. STD, the standard deviation.

Revisions:

Results section:

Lines 269-271:

“To attain high spatiotemporal resolution within whole-brain FOV, we employed a method termed STINT, which involves the continuous and rapid rotation of the transducer array at small angular increments to synthesize an equivalent high-density detection array (refer to the final part of the Results section for details).”

Lines 355-357:

“To preserve high resolution while achieving a broad FOV, the STINT method was applied through small-angle rotation of the hemispherical array combined with a two-dimensional translation of the animal holder (Fig.5a) within approximately 15 s (refer to the final part of the Results section for details).”

Lines 398-401:

“In each whole-body imaging session, the transducer array rotated 15° around the central axis within 5 s, acquiring 50 frames of raw PA data, which were then reconstructed into a 3D image using the STINT method (refer to the final part of the Results section for details).”

Lines 470-603:

“The spatiotemporal-integration (STINT) method (title)

Theoretically, the transducer array composed of 1024 elements is still not the optimal approach for providing a large enough FOV radius (denoted by r), which is determined by the following equation⁵⁸:

$$r = \frac{v}{4f_{uc}} \sqrt{\frac{2N}{\pi}} \quad (1)$$

where v denotes the speed of sound in the medium and N denotes the total number of elements. f_{uc} denotes the upper cutoff frequency of the transducer where the frequency-dependent SNR decreased to one above the central frequency. With calculation, $r = 1.5$ mm when $N = 1024$ and, if r grows to 20 mm, N will reach a number of approximately 175000 (v is set to 1500 m/s, and f_{uc} is calculated as 6.26 MHz by definition)⁵⁸. The STINT method serves to physically augment detection density and broaden the imaging FOV (especially in the deep tissue) with an appropriate reduction in temporal resolution. Here, the arrangement of transducer elements in a Fibonacci grid constitutes the essential hardware foundation for implementing the STINT method. The Fibonacci grid provides an elegant description of “uniform” angular sampling on the surface of a sphere, while notably lacking any central symmetry. This unique property allows us to rotate the array by small angles to generate an equivalent high-density detection array with near-uniform element distribution. In the phantom validations, rotating the original array by 1° for 10 consecutive times yielded a detection array comprising 10240 transducers (1024×10), and the reconstruction of leaf vein details using the equivalent high-density array was significantly superior to that achieved with the original array within the same FOV (Fig. 7a and b). In this study, the STINT method was further implemented in two different modes to accommodate various imaging targets and desired outcomes. This feature makes it suitable for imaging non-ultrafast physiological processes (for example, the hemodynamics in the Circle of Willis and the whole-body tracking of small molecules). For some exceptionally large FOV (for example, whole-trunk imaging),

the STINT method can also reconstruct 3D image sequences, achieving the imaging of dynamics.

We employed the first implementation of the STINT method in brain imaging and the whole-body probe tracking applications, achieving large-FOV, high-spatiotemporal-resolution 3D reconstruction solely through rapid, small-angle rotations of the transducer array during each imaging cycle (Fig. 3, 4 and 6). In brain imaging, we utilized the spherical array to integrate into an equivalent dense array with 10240 elements by rotating it 10 times at an equidistant angular increment of 1.5° , ensuring the uniformity of the array element density after integration according to this method (Fig. 7c). During each imaging cycle, the array was synchronized with a 10Hz dual-wavelength laser for triggering. After each group of laser pulses and raw signal acquisition, the array rotated clockwise (CW) by 1.5° (Fig. 7d, one black arrow represents a group of temporal closely-spaced dual-wavelength laser outputs). The 10 rotations took a total of 1.25 s (including the time for motor start-up and stabilization), followed by a counterclockwise (CCW) rotation of 15° to the initial position within 0.75 s (Fig. 7d), ready for the next imaging cycle. Owing to the rigid fixation of the skull by the animal holder, data acquired within each imaging cycle can be utilized for spatiotemporal-integrated reconstruction without the need for respiratory gating. In whole-body probe tracking, to achieve a large FOV, the array rotated CW by 15° within 5 s, acquiring 50 frames of raw data with a 10 Hz laser pulse, with each frame corresponding to an angular increment of 0.3° of the detection array (Fig. 7e). The array then rotated CCW by 15° over the following 5 s to return to its initial position, during which a similar data acquisition procedure was executed, with the sole difference being the reversed order of array positions. To elaborate, the reciprocating small-angle rotation of the array enabled continuous data acquisition during both CW and CCW rotations, thereby maximizing imaging temporal resolution (Fig. 7e). Due to the influence of thoracic respiratory motion, the 50 frames of raw data require respiratory-gating screening. The cross-correlation matrix was then obtained by calculating the correlation coefficient between frames. It could be seen that some of the frames were obviously different from other frames, containing the respiratory movement information (Fig. 7f). Due to symmetry, averaging the cross-correlation matrix along any dimension yielded the phase curve (Fig. 7g). By combining the derivative of the phase curve with threshold segmentation, frames with high correlation coefficients and low derivative values were identified as static frames, which were then used for spatiotemporal-integrated reconstruction (Fig. 7g). In STINT, the raw data corresponding to each array angular position was reconstructed according to the dual-speed-of-sound universal back-projection algorithms, in which each data frame was normalized using the detector surface signal to eliminate the influence of laser energy fluctuations. The detection coordinate used for each data frame was obtained by rotating the corresponding angle of the original coordinates around the rotation axis that has been calibrated. The images from all angular positions were then superimposed to obtain the reconstructed images with the STINT method (Supplementary Methods). It should be noted that reconstructing images directly from all-angle data is equivalent to reconstructing separately and then summing the results, as the universal back-projection⁵⁹ is a linear reconstruction process. However, the former approach requires higher GPU memory, which can be addressed with a more advanced GPU that supports larger memory and parallel computations.

We then employed the second implementation of the STINT method in reconstructing the high-resolution dynamics of the whole trunk of a living animal (Fig. 5), allowing a large 3D FOV to be obtained in a short period of time and reducing artifacts caused by spatial undersampling. This implementation utilized a more complex synthesis of array rotation and translation, achieving spatial resolution comparable to that at the field center across the entire FOV, while alleviating the resolution degradation in peripheral regions caused by spatial impulse response (SIR)^{60, 61}. To apply the STINT method in this case, the motorized translation stages drove the animal to scan six positions in the X-Y plane, rotating the array at a small angle around Z-

axis at each fixed position, and collecting all the raw data for image reconstruction. In each X-Y scanning position, part of the animal's trunk was inside the FOV of the original array (Fig. 7h). The output of a 10Hz dual-wavelength laser pulse was triggered synchronously with the rotating motion of the hemispherical array around the calibrated axis, and each group of laser pulses triggered the array to rotate CW by an angular spacing of 1.5° (Fig. 7i, one black arrow represents a group of temporal closely-spaced dual-wavelength laser outputs). In this process, 20 frames of dual-wavelength raw data sequence were acquired, after which the array was rotated CCW back to the initial position, taking a total of about 2.5 s (note that only single-wavelength raw data was shown here for clarity since the time interval of the dual-wavelength laser pulse was almost negligible relative to the laser repetition period, and, the processing methods of the dual-wavelength data in this paragraph were the same). The cross-correlation matrix as well as the phase curve were then obtained using the approach analogous to that described previously (Fig. 7j and k). Throughout the imaging procedure for the whole trunk, there existed relative scanning between the animal and the FOV. We conducted scans at six positions in the X-Y plane, with intervals of 12 mm along the X-axis and 20 mm along the Y-axis, which enables the creation of a comprehensive expanded FOV that encompasses the whole trunk (Fig. 7l). The coordinate system of the translation stages had been meticulously aligned with that of the hemispherical array, ensuring they shared the same Cartesian coordinate system to avoid coordinate mismatch. Corresponding raw data sequences and phase curves were generated at each scanning position and, by arranging the phase curves together, it became apparent that each contained at least one complete respiratory cycle with a frequency of approximately 1Hz (Fig. 7m). We selected the complete respiratory cycle composed of 11 frames from position 1 as a reference, and performed sliding correlation on the phase curves from other positions to complete phase alignment, generating a new sequence with 11 frames (Fig. 7n, the frame rate was still 10Hz). Within these curves, identical frame points represented the in-phase states within one respiratory cycle, albeit corresponding to different array detection angles (Fig. 7n). For each scanning position, 11 frames of raw data were individually reconstructed using the array coordinates of the corresponding angle of rotation with an FOV of $35 \times 35 \times 20 \text{ mm}^3$ (the voxel size $0.1 \times 0.1 \times 0.1 \text{ mm}^3$). According to the scanning intervals, the FOV for the whole trunk was set to $75 \times 47 \times 20 \text{ mm}^3$ (Fig. 7o). It was noted that different FOVs overlapped, which inhibited artifacts to a certain extent and improved the imaging SNR. By reconstructing the raw data corresponding to identical frame points from different scanning positions and integrating them according to the scanning intervals (orange dashed line in Fig. 7n), a single-frame 3D image of the whole trunk was obtained (Fig. 7n and p). It should be noted that the reconstruction results from each position needed Gaussian blur (filter size = 13, $\sigma = 13$) along the edge planes perpendicular to the X-Y plane to prevent discontinuities during the superposition process. After traversing all frame points in the aligned phase curves, the 10Hz whole-trunk dynamics was reconstructed with a total imaging time of approximately 15 s (Fig. 7p). In the whole process, while the STINT method formed a large FOV, an imaging region was reconstructed by spatial sampling from different positions and angles, which weakened artifacts and improved image fidelity.

To evaluate the effectiveness of the STINT method, we imaged a complete leaf vein at 690 nm with an FOV diameter of 60 mm. In the single-pulse imaging results, the outline of the leaf and the larger vein branches can be observed, indicating that 3D-PanoPACT has near-isotropic 3D PA imaging capabilities without motion, as evidenced by the maximum intensity projections (MIP) in the X-Z and X-Y planes (Fig. S9a). By integrating 20 sets of data within 2 s using the STINT method, 3D-PanoPACT reconstructs images that reveal rich details, with significantly reduced artifacts and markedly enhanced imaging fidelity (Fig. S9b). We first provide feature correlation coefficient (FCC) at the same image location before and after using the STINT method as a metric to quantify the difference in image features. It can be observed that in the

main vein, the FCC value is ~ 0.97 in the X-Y plane (Fig. S9c) and ~ 0.98 (Fig. S9d) in the X-Z plane, indicating that the STINT method does not introduce additional image blurring or artifacts. At the edge branches of the leaf veins, the FCC value is 0.39, indicating that the STINT method can indeed reveal detailed features that are almost impossible to reconstruct in single-pulse imaging (Fig. S9e). We further calculate the standard deviation (STD) as a metric of artifact suppression. It is observed that the STD value at the edge of FOV is reduced by 8 times after using the STINT method, indicating that artifacts are significantly weakened, and this effect could be further enhanced by integrating more data for reconstruction (Fig. S9f). We subsequently conducted a comparative analysis between the efficacy of the STINT method and the outcomes of numerical spatial interpolation reconstruction. In the interpolation, each transducer served as a node to construct a Delaunay triangulation mesh. Virtual transducers were placed at the centroids of each triangular element, and their corresponding data were generated by averaging the data from the three vertex transducers. A single spatial interpolation increased the total number of transducers by approximately threefold, thus, the STINT method performed a lateral comparison by synthesizing physical transducer arrays at three angular positions (Fig. S10b, c). By comparing the imaging results of the leaf skeleton, it is evident that spatial interpolation causes significant blurring of the details and degraded image quality (Fig. S10a, b), whereas the STINT method preserves the details, suppresses artifacts, and thereby improves image fidelity (Fig. S10a, c). We further utilized whole-body imaging to evaluate the impact of respiratory motion on the STINT method (the first implementation). Apparently, the reconstruction results using the STINT method, with or without respiratory gating, are superior to those of single-pulse imaging, as the 1024 elements are severely spatially-undersampled for the whole-body FOV (Fig. S11a), which is consistent with the leaf vein results. The benefits of respiratory gating are evident in reconstructing the details, particularly in the small vessels of the liver and the iliac artery, where it significantly improves the visibility and definition of these structures (Fig. S11b). To quantify this, the image sharpness is evaluated using the Sobel operator to detect gradients, and the standard deviation of the gradient magnitudes is calculated as the sharpness metric. Higher values indicate clearer images, while lower values suggest blurriness. Calculations show that the STINT method with respiratory gating improves image sharpness compared with the others (1.375 and 1.27 times for liver vessels, respectively; 1.25 and 1.07 times for the iliac artery, respectively), avoiding image blurring caused by motion (Fig. S11b). Additionally, this method hardly alters the existing image features in single-pulse results, as evidenced by the surface vessels, with a correlation coefficient for the cross-section of approximately 0.989 (Fig. S11c). In summary, we have verified the outstanding performance of the STINT method from multiple aspects.”

Discussion section:

Lines 663-686:

“We specifically explore the uniqueness of the STINT method in 3D-PanoPACT, particularly in comparison with the 3D-PACT (Lin et al. 2021), an advanced imaging system based on rotational scanning²¹⁻²³. From the perspective of array design, the angular isotropic nature of the Fibonacci grid enables 3D-PanoPACT to achieve high-quality 3D imaging within a certain FOV without requiring any additional motion (Fig. 2). The imaging frame rate is determined by the laser repetition rate in 3D-PanoPACT, while 3D-PACT must rely on rotation to achieve 3D imaging. This is because its array distribution does not possess the characteristic of 3D spatial isotropy, which limits the ability of 3D-PACT to capture high-speed biological processes, while 3D-PanoPACT is capable of this task. Moreover, 3D-PACT requires a minimum rotation of 90° to cover the 2π solid angle in spatial frequency, which corresponds to a large angular velocity of 0.785 rad/s when performing rapid imaging. With large mechanical load, the high-speed rotation inevitably induces vibrations, causing random deviations of the element positions and thereby degrading spatial resolution, with this

phenomenon becoming increasingly pronounced as the transducer's center frequency increases. For comparison, in 3D-PanoPACT, a small rotation angle suffices to achieve uniform spatial upsampling, markedly reducing random errors caused by vibrations and ensuring spatial resolution. Assuming a rotation of 15° in 3D-PanoPACT, the angular velocity is 6 times slower than that of 3D-PACT for the same imaging frame rate, which significantly reduces vibrations. In 3D-PanoPACT, the essence of using the STINT method is the spatial upsampling process, which involves trading off temporal resolution for the imaging FOV. In other words, even without the STINT method, 3D-PanoPACT already possesses an isotropic 3D-FOV (Fig. 2), and, the STINT method further achieves spatially scalable imaging (Fig. 3-6) by integrating multiple temporal data. In contrast, the array rotation in 3D-PACT is more akin to a scanning method (e.g., using linear array translation for 3D image reconstruction⁶²), and the system itself does not possess an isotropic 3D FOV. This leads to some imaging that can be achieved in 3D-PanoPACT being unattainable in 3D-PACT (Fig. 2 and 5). Therefore, even though both involve the rotation of the array, the implementation logic of 3D-PanoPACT is significantly different from that of 3D-PACT, and the STINT method is more suitable for describing the imaging approach of 3D-PanoPACT.”

Figures:

Fig. 7 (please refer to the end of this file to prevent confusion of figure numbers)

Supplementary Materials:

Fig. S9 (please refer to the end of this file to prevent confusion of figure numbers)

Reference:

Ref. 62

62. Huang, C. et al. Dual-scan photoacoustic tomography for the imaging of vascular structure on foot. *IEEE Transactions on Ultrasonics, Ferroelectrics, and Frequency Control* 70, 1703-1713 (2023).

*** The system's capacity for trunk imaging via XY axis translation reflects careful attention to motion correction which significantly enhance image quality while maintaining a high temporal resolution. These details, along with impulse response correction, should be emphasized further in the main text. Performance-wise, the ability to image entire organs or whole bodies at high speed and resolution is a significant achievement. However, the factors contributing to the system's high frame rates, superior signal-to-noise ratio (SNR), and impressive imaging depth should be more clearly highlighted in the main text, particularly in comparison to other systems in Supplementary Table 1.**

Reply:

Thank you for the valuable comments. In trunk imaging, we reconstructed the whole-body dynamics using spatiotemporal integration combined with phase alignment, which is the second implementation of the STINT method. In whole-body probe tracking, to eliminate the effects of respiratory motion, we extracted static frames for reconstruction using the method of cross-correlation gating, which is the first implementation form of the STINT method. We have added a detailed description titled “The STINT Method” in the last part of the Results section, and included Fig. 7. To further highlight the correction of the electrical impulse response (EIR), we have added explanations in the first part of the Results section and emphasized it again in the brain imaging section.

The high frame rate of 3D-PanoPACT is attributed to the Fibonacci grid arrangement of the array and the implementation of the STINT method. The transducer array used in our system can directly form an isotropic 3D FOV, supporting high-quality, high-frame-rate imaging, and the frame rate is solely dependent on the laser pulse repetition rate. The STINT method, through small rotation of the transducer array, enables image

reconstruction with minimal sacrifice of temporal resolution. This approach is particularly suited for non-ultrafast wide-field dynamic imaging applications, where maintaining high temporal resolution is crucial. The high SNR is mainly due to the larger transducer element size, as there is a positive correlation between element size and piezoelectric signal intensity within a certain range. The incorporation of preamplifiers, suppression of channel crosstalk, and shielding of noise are also important factors in improving SNR. For example, in 3D-PanoPACT, the array interface, preamplifier circuit, and data acquisition module are all plug-ins, which greatly avoids electromagnetic interference introduced by transmission cables. Moreover, from the perspective of data acquisition, a high SNR fundamentally enhances the imaging depth. On the other hand, the array geometric calibration and the dual-speed-of-sound universal back-projection (UBP) method enable a good reconstruction of deep tissue signals. Otherwise, even if there are deep tissue signals, it would be difficult to reconstruct image features due to coordinate deviations, acoustic speed heterogeneity, and other issues. We have added relevant discussions in the Discussion section and included the above-mentioned factors in Table S1 to facilitate horizontal comparisons between different PACT systems.

Revisions:

Results section:

Lines 178-186:

“To enhance the quantitative accuracy of the PA images, we addressed the issue of limited bandwidth in the transducers and subsequent circuits, which affects the fidelity of the acquired signals (see Methods). We employed the edge-emission method to measure the system’s electrical impulse response (EIR), using a 10 μm thick polyimide film cut into a 2×15 mm rectangle as the PA signal source (Fig. S4a and b). This method produced a pair of negative and positive unipolar PA pulses at the edges of the film, which were used to determine the system EIR (Fig. S4c). To demonstrate the effectiveness of EIR deconvolution, we imaged three square polyimide films with side lengths of 3 mm, 6 mm, and 9 mm. The deconvolution process, using a Wiener filter, significantly improved the reconstruction of low-frequency components (Fig. S4d and e), which are crucial for accurate sO_2 unmixing and visualizing neurovascular coupling responses in the cortex.”

Lines 294-296:

“Given that low-frequency information often accompanies functional responses such as cerebral blood oxygenation changes, we adopted the precisely measured EIR and applied deconvolution to the signals, enabling more accurate reconstruction of PA image amplitude (see Methods).”

Lines 492-566:

“We employed the first implementation of the STINT method in brain imaging and the whole-body probe tracking applications, achieving large-FOV, high-spatiotemporal-resolution 3D reconstruction solely through rapid, small-angle rotations of the transducer array during each imaging cycle (Fig. 3, 4 and 6). In brain imaging, we utilized the spherical array to integrate into an equivalent dense array with 10240 elements by rotating it 10 times at an equidistant angular increment of 1.5° , ensuring the uniformity of the array element density after integration according to this method (Fig. 7c). During each imaging cycle, the array was synchronized with a 10Hz dual-wavelength laser for triggering. After each group of laser pulses and raw signal acquisition, the array rotated clockwise (CW) by 1.5° (Fig. 7d, one black arrow represents a group of temporal closely-spaced dual-wavelength laser outputs). The 10 rotations took a total of 1.25 s (including the time for motor start-up and stabilization), followed by a counterclockwise (CCW) rotation of 15° to the initial position within 0.75 s (Fig. 7d), ready for the next imaging cycle. Owing to the rigid fixation of the skull by the animal holder, data acquired within each imaging cycle can be utilized for spatiotemporal-integrated reconstruction without the need for respiratory gating. In whole-body probe tracking, to achieve a large FOV, the array rotated CW by 15° within 5 s, acquiring 50 frames of raw data with a 10 Hz laser

pulse, with each frame corresponding to an angular increment of 0.3° of the detection array (Fig. 7e). The array then rotated CCW by 15° over the following 5 s to return to its initial position, during which a similar data acquisition procedure was executed, with the sole difference being the reversed order of array positions. To elaborate, the reciprocating small-angle rotation of the array enabled continuous data acquisition during both CW and CCW rotations, thereby maximizing imaging temporal resolution (Fig. 7e). Due to the influence of thoracic respiratory motion, the 50 frames of raw data require respiratory-gating screening. The cross-correlation matrix was then obtained by calculating the correlation coefficient between frames. It could be seen that some of the frames were obviously different from other frames, containing the respiratory movement information (Fig. 7f). Due to symmetry, averaging the cross-correlation matrix along any dimension yielded the phase curve (Fig. 7g). By combining the derivative of the phase curve with threshold segmentation, frames with high correlation coefficients and low derivative values were identified as static frames, which were then used for spatiotemporal-integrated reconstruction (Fig. 7g). In STINT, the raw data corresponding to each array angular position was reconstructed according to the dual-speed-of-sound universal back-projection algorithms, in which each data frame was normalized using the detector surface signal to eliminate the influence of laser energy fluctuations. The detection coordinate used for each data frame was obtained by rotating the corresponding angle of the original coordinates around the rotation axis that has been calibrated. The images from all angular positions were then superimposed to obtain the reconstructed images with the STINT method (Supplementary Methods). It should be noted that reconstructing images directly from all-angle data is equivalent to reconstructing separately and then summing the results, as the universal back-projection⁵⁹ is a linear reconstruction process. However, the former approach requires higher GPU memory, which can be addressed with a more advanced GPU that supports larger memory and parallel computations.

We then employed the second implementation of the STINT method in reconstructing the high-resolution dynamics of the whole trunk of a living animal (Fig. 5), allowing a large 3D FOV to be obtained in a short period of time and reducing artifacts caused by spatial undersampling. This implementation utilized a more complex synthesis of array rotation and translation, achieving spatial resolution comparable to that at the field center across the entire FOV, while alleviating the resolution degradation in peripheral regions caused by spatial impulse response (SIR)^{60, 61}. To apply the STINT method in this case, the motorized translation stages drove the animal to scan six positions in the X-Y plane, rotating the array at a small angle around Z-axis at each fixed position, and collecting all the raw data for image reconstruction. In each X-Y scanning position, part of the animal's trunk was inside the FOV of the original array (Fig. 7h). The output of a 10Hz dual-wavelength laser pulse was triggered synchronously with the rotating motion of the hemispherical array around the calibrated axis, and each group of laser pulses triggered the array to rotate CW by an angular spacing of 1.5° (Fig. 7i, one black arrow represents a group of temporal closely-spaced dual-wavelength laser outputs). In this process, 20 frames of dual-wavelength raw data sequence were acquired, after which the array was rotated CCW back to the initial position, taking a total of about 2.5 s (note that only single-wavelength raw data was shown here for clarity since the time interval of the dual-wavelength laser pulse was almost negligible relative to the laser repetition period, and, the processing methods of the dual-wavelength data in this paragraph were the same). The cross-correlation matrix as well as the phase curve were then obtained using the approach analogous to that described previously (Fig. 7j and k). Throughout the imaging procedure for the whole trunk, there existed relative scanning between the animal and the FOV. We conducted scans at six positions in the X-Y plane, with intervals of 12 mm along the X-axis and 20 mm along the Y-axis, which enables the creation of a comprehensive expanded FOV that encompasses the whole trunk (Fig. 7l). The coordinate system of the translation stages had been meticulously aligned with that of

the hemispherical array, ensuring they shared the same Cartesian coordinate system to avoid coordinate mismatch. Corresponding raw data sequences and phase curves were generated at each scanning position and, by arranging the phase curves together, it became apparent that each contained at least one complete respiratory cycle with a frequency of approximately 1Hz (Fig. 7m). We selected the complete respiratory cycle composed of 11 frames from position 1 as a reference, and performed sliding correlation on the phase curves from other positions to complete phase alignment, generating a new sequence with 11 frames (Fig. 7n, the frame rate was still 10Hz). Within these curves, identical frame points represented the in-phase states within one respiratory cycle, albeit corresponding to different array detection angles (Fig. 7n). For each scanning position, 11 frames of raw data were individually reconstructed using the array coordinates of the corresponding angle of rotation with an FOV of $35 \times 35 \times 20 \text{ mm}^3$ (the voxel size $0.1 \times 0.1 \times 0.1 \text{ mm}^3$). According to the scanning intervals, the FOV for the whole trunk was set to $75 \times 47 \times 20 \text{ mm}^3$ (Fig. 7o). It was noted that different FOVs overlapped, which inhibited artifacts to a certain extent and improved the imaging SNR. By reconstructing the raw data corresponding to identical frame points from different scanning positions and integrating them according to the scanning intervals (orange dashed line in Fig. 7n), a single-frame 3D image of the whole trunk was obtained (Fig. 7n and p). It should be noted that the reconstruction results from each position needed Gaussian blur (filter size = 13, $\sigma = 13$) along the edge planes perpendicular to the X-Y plane to prevent discontinuities during the superposition process. After traversing all frame points in the aligned phase curves, the 10Hz whole-trunk dynamics was reconstructed with a total imaging time of approximately 15 s (Fig. 7p). In the whole process, while the STINT method formed a large FOV, an imaging region was reconstructed by spatial sampling from different positions and angles, which weakened artifacts and improved image fidelity.”

Discussion section:

Lines 605-634:

“In this study, we introduced the design philosophy, parameter calibration, and imaging strategies of a novel PA imaging device—3D-PanoPACT—and demonstrated its broad applicability and significant potential for practical biological research. The 3D-PanoPACT system has been designed to non-invasively image anatomical structures and functional dynamics, offering superior spatiotemporal resolution, high fidelity, and a broad 3D FOV. This system encompasses several advanced features: (1) The design of array element arrangement using a high-density Fibonacci grid provides single-pulse 3D imaging ability. The hemispherical ultrasonic transducer array utilized in 3D-PanoPACT covers all receiving angles in k-space, with a uniform distribution of 1024 elements ensuring spatial sampling density. The array directly forms an isotropic 3D FOV without any scanning, supporting high-quality, real-time imaging, and the frame rate is solely dependent on the laser pulse repetition rate. The single-pulse imaging capability enables 3D visualization of dynamic processes at video frame rates, enhancing its attractiveness (Fig. 2). By synchronously controlling two or more lasers, real-time multi-wavelength imaging can be achieved, which is capable of capturing dynamic functional information that undergoes rapid changes (Fig. 4f-k). (2) The STINT method ensures a wide-field 3D FOV with high spatiotemporal resolution. Based on the Fibonacci grid array, the STINT method only requires a small rotation angle of the array to achieve equivalent spatial upsampling, enabling FOV expansion with minimal sacrifice of temporal resolution. With this method, 3D-PanoPACT achieves pioneering progress by distinctly imaging the anatomy of the important cerebrovascular network—the Circle of Willis—and monitoring its functional dynamics (Fig. 3 and 4a-e). We also use the STINT method to obtain high-fidelity 3D dynamic imaging of the whole trunk (Fig. 5) and effectively track the metabolic pathways of the small-molecule probe A1094 across multiple organs at the whole-body scale (Fig. 6), thereby highlighting the practical utility of our approach. (3) The meticulous engineering

implementation. The transducer array design was validated, with sensitivity variation among the 1024 elements kept within 4% and bandwidth variation within 1.5% during fabrication. The incorporation of preamplifiers, suppression of channel crosstalk, and shielding of noise are important factors in improving the SNR of the raw data, thereby enhancing imaging depth. The application of dual-speed-of-sound reconstruction allows the accurate depiction of deep-seated features and also improves imaging depth. The calibration of system array element positions and the rotation axis further optimizes the spatial resolution. The application of EIR deconvolution processing and optical fluence compensation enhances the quantitative capability of images and improves the accuracy of functional parameter estimation in biological applications. All the implementation methods and technical details mentioned above collectively determine the outstanding imaging performance of 3D-PanoPACT (Table S1).”

Supplementary Materials:

Table S1 (please refer to the end of this file)

*** Is there any improvement due to the choice of sensor element, in terms of sensitivity? How does it compare with piezoelectric elements used in other works? Did the authors characterize and if so attempt to minimize the dispersion of sensitivities and bandwidths across the elements?**

Reply:

Thank you for the insightful comments. After determining the detection array structure, we have specifically designed the element size. We first used the k-Wave toolbox to investigate the relationship between detection sensitivity and transducer element size. We placed an ideal PA point source at the center of FOV with the transducer set to the center frequency and bandwidth used and observed the relationship between the detected PA amplitude and the radius of the transducer element (Fig. R3a). The simulation results indicate that the PA amplitude exhibits a quadratic relationship with the radius (i.e., is linearly proportional to the area) when the radius is small. However, when the radius is large, the simulated values fall below the ideal values, and the quadratic relationship is no longer maintained (Fig. R3b). The critical point where these two relationships intersect is precisely around Radius = 2.5 mm. In other words, designing the transducer element radius to be 2.5 mm can maximize the detection sensitivity at the highest cost-effectiveness ratio. We further used Field II to simulate the receiving aperture angles for different element radii. The simulations show that while a transducer with a radius of 0.5 mm has a large aperture angle (Fig. R3c), its sensitivity is 25 times lower than that of a transducer with a radius of 2.5 mm (Fig. R3b), which is not conducive to deep tissue imaging. For comparison, the transducer with a radius of 1.5 mm has an aperture angle comparable to that of the transducer with a radius of 2.5 mm, but its sensitivity is 2.78 times weaker. After evaluating both detection sensitivity and receiving aperture angle, we determine that a transducer element radius of 2.5 mm (i.e., 5mm diameter) offers the optimal balance for our imaging system. We have added relevant explanations in the Results and Discussion section and included Fig. S2 to provide further clarification.

By comparing with other imaging systems, we found that 3D-PanoPACT uses a larger transducer element size (3D-PanoPACT: 5 mm; 3D-PACT: 0.7mm; PAI-04: 1.5mm; FONT: 1-2.5mm) to optimize SNR. We also calculated the ratio of the transducer element size to the central wavelength (ROSW) to qualitatively characterize the aperture angles of different transducers. The 3D-PACT system, with a ROSW value close to 1, forms an ideal receiving aperture but has limited sensitivity due to its small transducer size, and is more affected by the quantization noise of data acquisition. Other systems have comparable ROSW values (3D-PanoPACT: 10.53; PAI-04: 4; FONT: 3.33-8.33) indicating similar aperture angles, yet 3D-PanoPACT achieves higher sensitivity. The optimization of transducers using materials with wider bandwidth or higher sensitivity (such as PVDF or CMUT)

may potentially enhance imaging performance, a possibility that warrants further investigation in future studies. We have added relevant content in the Discussion section and a comparison of transducers across different systems in Table S1.

The transducer array used in 3D-PanoPACT was rigorously characterized during the manufacturing process. The manufacturer tested the bandwidth and sensitivity dispersion of the transducers, ensuring that the sensitivity dispersion is less than 4% and the bandwidth dispersion is less than 1.5%. These values are already sufficiently good and do not significantly impact the imaging performance. Additionally, these characteristics are non-time-varying and do not compromise the accuracy of longitudinal imaging or analysis. Therefore, we omitted the calibration steps for fluctuations in sensitivity and bandwidth.

Fig. R3 (Fig. S2 in the revised manuscript). Design of transducer element size.

a The simulation setup. **b** Simulation results of the relation between element radius and detected PA amplitude. **c-e** Simulation of transducer element sensitivity distribution.

Revisions:

Results section:

Lines 140-156:

“Considering that acoustic detection is crucial for imaging quality, the determination of the array element size has undergone special design and validation. We first used the k-Wave toolbox to investigate the relationship between detection sensitivity and transducer element size. We placed an ideal PA point source at the center of FOV with the transducer set to the center frequency and bandwidth used and observed the relationship between the detected PA amplitude and the radius of the transducer element (Fig. S2a). The

simulation results indicate that the PA amplitude exhibits a quadratic relationship with the radius (i.e., is linearly proportional to the area) when the radius is small. However, when the radius is large, the simulated values fall below the ideal values, and the quadratic relationship is no longer maintained (Fig. S2b). The critical point where these two relationships intersect is around Radius = 2.5 mm. In other words, designing the transducer element radius to be 2.5 mm can maximize the detection sensitivity at the highest cost-effectiveness ratio. We further used Field II to simulate the receiving aperture angles for different element radii. The simulations show that while a transducer with a radius of 0.5 mm has a large aperture angle (Fig. S2c), its sensitivity is 25 times lower than that of a transducer with a 2.5 mm radius (Fig. S2b), which is not conducive to deep tissue imaging. For comparison, the transducer with a radius of 1.5 mm has an aperture angle comparable to that of the transducer with a 2.5 mm radius, but its sensitivity is 2.78 times weaker. After evaluating both detection sensitivity and receiving aperture angle, we determine that a transducer element radius of 2.5 mm (i.e., 5 mm diameter) offers the optimal balance for 3D-PanoPACT.”

Discussion section:

Lines 637-646:

“In terms of ultrasound transducers, 3D-PACT features a larger element size with optimized SNR (3D-PanoPACT: 5 mm; 3D-PACT: 0.7mm; PAI-04: 1.5mm; FONT: 1-2.5mm). The ratio of element size to the central wavelength (ROSW) is calculated to qualitatively characterize the aperture angles of different transducers. The 3D-PACT system, with a ROSW value close to 1, forms an ideal receiving aperture but has limited sensitivity due to its small transducer size and is more affected by the quantization noise of data acquisition. Other systems have comparable ROSW values (3D-PanoPACT: 10.53; PAI-04: 4; FONT: 3.33-8.33) indicating similar aperture angles, yet 3D-PanoPACT achieves higher sensitivity. The optimization of transducers using materials with wider bandwidth or higher sensitivity may potentially enhance imaging performance, a possibility that warrants further investigation.”

Supplementary Materials:

Fig. S2 (please refer to the end of this file to prevent confusion of figure numbers)

*** I.106-107: What is the role of the phase plate connecting the preamplifier and DAQ system?**

Reply:

The role of the phase plate is to correspond the output of the preamplifier to the input channel number of the DAQ. Because the two have different interface forms, the phase plate acts as an adapter plate. This direct plug-in architecture is conducive to minimizing coupling noise. To avoid confusion, we have replaced the term “phase plate” with “adapter plate” and added corresponding descriptions.

Revisions:

Results section:

Lines 119-122:

“Each module contains 256-channel pre-amplification circuits and 256-channel data acquisition (DAQ) circuits interlinked with an adapter plate for facilitating interface correspondence. This direct plug-in architecture is conducive to minimizing coupling noise.”

*** The two laser sources have a 10 Hz repetition rate. Please explain how a 25Hz single color frame rate is achieved.**

Reply:

The 3D-PanoPACT offers two illumination modes. 1064 nm output reaches a maximum laser pulse repetition rate of 25 Hz when single-wavelength imaging mode is applied (Fig. R4), and a lower repetition rate (e.g., 10 Hz) is also achievable in this mode. The tunable and 1064 nm outputs work in pairs to provide a 10 Hz dual-wavelength imaging mode, whose pulse repetition rate is limited by the damage threshold of the crystal inside OPO (Fig. R4). We have emphasized this point in the first part of the Results section.

Fig. R4 (Fig. 1c in the revised manuscript)

Revisions:

Results section:

Lines 124-127:

“The 3D-PanoPACT offers two illumination modes. 1064 nm output reaches a maximum laser pulse repetition rate of 25 Hz when single-wavelength imaging mode is applied (Fig. 1c). The tunable and 1064 nm outputs work in pairs to provide a 10 Hz dual-wavelength imaging mode, whose pulse repetition rate is limited by the damage threshold of the crystal inside OPO (Fig. 1c).”

*** Please explain how the laser pulses are delayed by 20us in the dual color approach.**

Reply:

The timing of the entire system, including the triggering of the two lasers, is controlled by an external delay pulse generator. The Q-switch delay for laser 1 (tunable output) is 160 μs, and for laser 2 (1064nm output), it is 230 μs. Therefore, to achieve a 20 μs delay in the output of laser 2 relative to laser 1, we delay the flash signal of laser 1 by 50 μs relative to that of laser 2. This results in a laser output interval of $160 + 50 - 230 = -20 \mu\text{s}$, which corresponds to a 20 μs delay. We have added explanations in the Results section.

Revisions:

Results section:

Lines 129-131:

“To achieve a 20 μs delay in the output of laser 2 (1064nm output) relative to laser 1 (tunable output), we delayed the flash signal of laser 1 by 50 μs relative to that of laser 2, resulting in a laser output interval of $160 + 50 - 230 = -20 \mu\text{s}$, corresponding to a 20 μs delay.”

Liver imaging:

*** I.146-147: ‘positioning the ROI’: Please clarify whether this means centering with respect to the hemispheric sensor.**

Reply:

When we refer to “positioning the ROI”, we mean centering the specific area of interest, such as the left lateral lobe of the liver, within the imaging FOV of the hemispherical array. This ensures that the target region is optimally positioned for high-resolution imaging. To avoid any misunderstanding, we have revised the corresponding description to: “By placing the left lateral lobe, which is the largest lobe of the liver, at the center of the imaging FOV of the hemispherical array, obvious displacement can be observed during the expansion and contraction of the thoracic cavity.”

Revisions:

Results section:

Lines 224-226:

“By placing the left lateral lobe, which is the largest lobe of the liver, at the center of the imaging FOV of the hemispherical array, obvious displacement can be observed during the expansion and contraction of the thoracic cavity (Fig. 2a).”

*** Heartbeat is usually not a sine, but shows several segments that corresponds to the successive opening, closing and contractions of the heart sub-structures. The sine wave extraction has a limited physiological relevance in itself, although the phase delay is interesting. Did the authors to extract more detailed information (for instance related to distole and systole), using a more complex filter?**

Reply:

Thank you for your insightful comments regarding the extraction of detailed information from heartbeat signals. As correctly pointed out, heartbeat signals are complex and consist of several segments. In our study, the 3D-PanoPACT’s 25 Hz real-time 3D imaging capability allows for accurate reconstruction of the heartbeat frequency, which is around 4 Hz. The 25 Hz frame rate, exceeding four times the typical 4 Hz heartbeat frequency, meets the practical requirement for accurate signal reconstruction, which often demands at least four times the signal frequency despite the Nyquist theorem’s minimum of two times. The pulse wave signals in the hepatic artery are correlated with the heartbeat frequency, enabling us to analyze the amplitude and phase near the heartbeat frequency. This approach provides detailed insights into cardiac activity and its impact on peripheral blood flow. Similar analysis methods have also been reported in SIP-PACT in imaging 2D sections². In Fig. 2b and d, the sine wave is used merely to represent the heartbeat frequency and to characterize the phase information, rather than to depict the heartbeat signal itself.

The dynamics of the cardiac substructures, which have key physiological significance, manifest as higher-frequency harmonic information in the spectrum, often necessitating a higher sampling frame rate. For reference in small animal echocardiography imaging, a cardiac cycle typically requires 100 sampling frames³. If the heartbeat frequency is around 4 Hz, the ideal imaging frame rate will reach approximately 400 Hz. Therefore, we believe that an imaging frame rate of 25 Hz is insufficient to capture valuable substructure dynamics of the heart, and even with more complex filters, obtaining convincing results remains challenging. In fact, the solution to this limitation is to equip the 3D-PanoPACT with a high-energy pulsed laser with a higher repetition rate (e.g., 100 Hz). This necessity arises from the system’s capability for real-time 3D imaging, where the imaging frame rate is fundamentally determined by the laser repetition rate. Therefore, we believe that, after optimizing the laser

configuration, 3D-PanoPACT will have great potential for studying ultrafast dynamics in living organisms. We have incorporated the analysis of this system limitation into the Discussion section.

Revisions:

Discussion section:

Lines 693-701:

“(2) Optimization of temporal resolution. Despite its single-pulse 3D imaging capability, the low laser pulse repetition rate limits 3D-PanoPACT’s ability to capture ultrafast physiological processes in this study. For instance, in imaging liver dynamics, we successfully extracted the heartbeat frequency from the temporal sequence but the detailed dynamics of cardiac substructures remained elusive. This limitation arises because these dynamics typically manifest as high-frequency harmonics of the heartbeat frequency, and the current imaging frame rate of 25 Hz fails to meet the Nyquist sampling criterion, thereby making it difficult to resolve this information. For reference in echocardiography imaging³¹, an imaging frame rate exceeding 100 Hz may significantly enhance the capability to capture rapid physiological phenomena but also impose higher demands on high-energy, high-repetition-rate pulsed laser equipment.”

References:

Ref. 31:

31. Zacchigna, S. et al. Towards standardization of echocardiography for the evaluation of left ventricular function in adult rodents: a position paper of the ESC Working Group on Myocardial Function. *Cardiovascular research* 117, 43-59 (2021).

*** Motion artifacts may be influencing the phase delay shown in Fig.2d, how did the authors ruled this out?**

Reply:

The 3D-PanoPACT system possesses considerable single-pulse 3D imaging capabilities within a certain FOV, even without the STINT method. Its imaging frame rate depends on the laser repetition rate when single-pulse imaging is applied, enabling real-time 3D imaging. The liver imaging results and data analysis shown in Fig. 2 are all based on 25 Hz real-time 3D imaging and thus are not affected by motion artifacts. Movie S1 further illustrates this imaging characteristic more intuitively. We emphasized this point in the Results section.

Revisions:

Results section:

Lines 218-222:

“The 3D-PanoPACT possesses considerable single-pulse 3D imaging capabilities within a certain FOV. Its imaging frame rate depends on the laser repetition rate when single-pulse imaging is applied, enabling real-time 3D imaging. Here, we demonstrate that 3D-PanoPACT was capable of achieving real-time 3D imaging of the liver non-invasively, enabling the study of rapid hemodynamic processes at the organ level, without exogenous contrast agents (Movie S1).”

*** As this phase can be extracted for each voxel of the reconstruction volume, a phase map similar to Fig.2e could be generated and might reveal a phase gradient related to the blood pressure wave mentioned in l.160.**

Reply:

Thank you for your insightful comments and suggestions. In response, we have added details on mapping the 3D phase gradient of hepatic vasculature using the cardiac frequency component (Fig. R5f). We calculated the phase angle for the heartbeat frequency component (3.9 Hz) of each voxel’s time sequence, with the proximal vascular

location set as the zero phase angle, and the temporal phase delay was then derived. The conversion from phase angle φ to temporal phase delay t_φ can be formulated as:

$$t_\varphi = \frac{\varphi}{2\pi f}$$

where $f=3.9$ Hz. The calculated results were overlaid on the anatomical structure using a pseudocolor encoding. This mapping, enabled by 3D-PanoPACT's high-quality video-rate imaging, may relate to the spatial gradient of blood pressure⁴ and serve as a reference for assessing hepatic hemodynamics^{5, 6}. We have added Fig. 2f, included relevant explanations in the Results section, and supplemented the implementation methods in the Methods section.

Fig. R5 (Fig. 2 in the revised manuscript). Real-time visualization of liver dynamics in 3D-PanoPACT (Movie S1).

Revisions:

Results section:

Lines 247-252:

“Further, by extracting the phase of each voxel’s time sequences at the cardiac frequency component, we are able to map the 3D phase gradient of hepatic vasculature (Fig. 2f) (see Methods), which may be related to the spatial gradient of blood pressure³². This unique mapping relies on the high-quality video-rate 3D imaging enabled by 3D-PanoPACT, which may serve as a reference for assessing overall hepatic hemodynamics or as an indicator for certain liver diseases, such as acute liver injury^{33, 34}.”

Methods section:

Lines 946-951:

“In mapping the phase of hepatic pulse waves, we calculated the phase angle φ corresponding to the 3.9 Hz frequency component of the sinusoidal waveform for the time sequence of each voxel, with the phase angle at the proximal vascular location designated as the zero phase angle. The conversion from phase angle φ to temporal phase delay t_φ can be formulated as:

$$t_\varphi = \frac{\varphi}{2\pi f} \quad (11)$$

where $f = 3.9$ Hz. The calculated results were overlaid on the anatomical structure using a pseudocolor encoding.”

Figures:

Fig. 2f (please refer to the end of this file to prevent confusion of figure numbers)

References:

Ref. 32-34:

32. Hirata, K., Kawakami, M. & O'Rourke, M.F. Pulse wave analysis and pulse wave velocity a review of blood pressure interpretation 100 years after Korotkov. *Circulation journal* 70, 1231-1239 (2006).
33. Uyeda, J.W., LeBedis, C.A., Penn, D.R., Soto, J.A. & Anderson, S.W. Active hemorrhage and vascular injuries in splenic trauma: utility of the arterial phase in multidetector CT. *Radiology* 270, 99-106 (2014).
34. Chen, C. et al. Peripheral microvessel area better predicts the severity of coronary stenosis of acute myocardial infarction patients over pulse wave velocity. *Scientific Reports* 14, 28584 (2024).

Brain Imaging:

* The term “cranial window” is used ambiguously, referring either to a pure craniotomy or to procedures where the skull is replaced with an ultrasound-transparent cover, as seen in Ref 37. To avoid confusion, it is crucial to clarify that the skull was removed without replacement in this study. Additionally, specifying in the main text which part of the skull was removed would improve understanding, especially when comparing this system’s results to others like the FONT 2 system by Razansky’s group (ref 4 in Supp Inf, 63 in main text), which works with an intact scalp and skull. Supplementary Table 1 should be adjusted to reflect this.

Reply:

Thank you for the comments. In the brain imaging using 3D-PanoPACT, we removed a portion of the parietal bone with a diameter of ~ 8 mm and sutured the scalp without adding replacement. This approach ensures that signals from the basal brain vessels, such as the Circle of Willis, can propagate outward with weakened distortion. To avoid the ambiguity associated with the term “cranial window” and enhance readability, we have clarified this point in the Results section.

3D-PACT imaged the brain with skull thinning to enhance signal transmission, whereas the FONT system by

Razansky's group performed imaging with the intact scalp and skull. In comparison, 3D-PanoPACT focuses more on the imaging depth of the whole brain to capture physiological activities from the brain base. We have adjusted Table S1 to reflect these differences and made clarifications, ensuring that readers can accurately compare the methodologies and results of these systems.

Revisions:

Results section:

Lines 261-263:

“To minimize acoustic loss and wavefront distortion, we removed a portion of the parietal bone with a diameter of ~ 8 mm and sutured the scalp without adding replacement.”

Discussion section:

Lines 650-652:

“Despite 3D-PACT and FONT systems not partially removing the skull, 3D-PanoPACT focuses on whole-brain imaging depth to capture physiological activities from the brain base (Table S1).”

Methods section:

Lines 988-991:

“To achieve superior deep-brain imaging results, a skull-removed operation was performed on all mice, which reduced the distortion and attenuation of the PA signal. We removed a portion of the parietal bone with a diameter of ~ 8 mm and sutured the scalp (detailed operation steps were introduced in reference 39).”

Supplementary Materials:

Table S1 (please refer to the end of this file to prevent confusion of figure numbers)

*** A major concern with such a large craniotomy is the static overpressure inducing brain swelling, which can impact imaging quality and strongly limits physiological relevance of the acquired images. This physiological effect is not sufficiently addressed in the paper, please provide details on how the authors managed this challenge.**

Reply:

Thanks for the valuable comments. To image the basal brain vascular network, we removed a portion of the parietal bone and sutured the scalp. During the modeling process, to avoid overpressure, we carefully removed only the skull without causing any damage to the dura mater. We also employed low-dose mannitol to further alleviate brain swelling that may be induced by static overpressure^{7, 8}. During the 10-day recovery period from surgery to imaging testing, mannitol was administered every 24 hours at a dose of 0.25 g/kg body weight. The injection volume of the solution was 0.1 mL/g body weight, and the injection rate was controlled at 5 mL/min. During the postoperative period, the wounds healed completely, and the animals exhibited normal behavior, gaining weight as expected.

To address the concern regarding the physiological relevance and reliability of our cranial window model, we have previously published a paper in 2021 that specifically investigated the value of this model⁹. In that study, we compared the brains of modeled animals with those of control groups using detailed structural imaging (via small-animal MRI) and pathological analysis (via H&E staining). Our results confirmed the effectiveness of this modeling method for short-term PA brain imaging, and the model has been used in other deep-brain photoacoustic imaging studies¹⁰. Using an acoustically transparent cranial window may physically prevent static overpressure to support long-term brain imaging studies, but this does not affect the imaging capabilities of 3D-PanoPACT which we aim to highlight in this work. We have added necessary explanations in the Results section and included additional details about animal modeling in the Methods section.

Revisions:**Results section:**

Lines 263-267:

“During the modeling process, to avoid overpressure, we carefully removed only the skull without causing any damage to the dura mater and employed low-dose mannitol to further alleviate brain swelling that may be induced by static overpressure (see Methods)^{36,37}. The reliability of this modeling method for short-term brain imaging studies has been confirmed through structural imaging and pathological sections of the control groups³⁸.”

Methods section:

Lines 991-997:

“To avoid overpressure, we carefully removed only the skull without causing any damage to the dura mater. We also employed low-dose mannitol to further alleviate brain swelling that may be induced by static overpressure. During the recovery period from surgery to imaging testing, mannitol was administered every 24 hours at a dose of 0.25 g/kg body weight. The injection volume of the solution was 0.1 mL/g body weight, and the injection rate was controlled at 5 mL/min. The postoperative mice were reared normally and observed for 10 days, during which time the wounds were completely healed and the animals behaved in the course of nature while gaining weight.”

References:

Ref. 36-38:

36. Paczynski, R., He, Y., Diring, M. & Hsu, C. Multiple-dose mannitol reduces brain water content in a rat model of cortical infarction. *Stroke* 28, 1437-1444 (1997).

37. Zeng, H.-K. et al. A comparative study on the efficacy of 10% hypertonic saline and equal volume of 20% mannitol in the treatment of experimentally induced cerebral edema in adult rats. *BMC neuroscience* 11, 1-10 (2010).

38. Wang, X. et al. A skull-removed chronic cranial window for ultrasound and photoacoustic imaging of the rodent brain. *Frontiers in Neuroscience* 15, 673740 (2021).

*** To which extent does the reduced frame rate induce some smearing of the final image due to motion? This can be quantified by imaging at 25Hz with one single excitation wavelength.**

Reply:

Thanks for the comments. During brain imaging, the holder rigidly fixes the skull, so the head does not shift with respiratory motion. The STINT method reconstructs a whole-brain image by synthesizing 10 frames of data collected at different angular positions within 1.25 seconds. Since there are no head motions, there is no image blurring. For reference, the imaging results of the leaf vein in Fig. R2 illustrate the comparison between single-pulse 3D images and the images reconstructed using the STINT method. Both the leaf vein phantom and the brain can be considered as objects without motions, making them suitable for this type of comparison. As previously analyzed, no image blurring occurs in this situation.

To further illustrate the impact of respiratory motion on image quality and the methods to address these issues, we have used whole-body imaging (used in probe tracking) as an example. Even when animals are fixed in the imaging holder, respiratory motion in the thoracic cavity can still cause regular movement in the trunk. Therefore, when using the STINT method, we incorporated respiratory gating to extract static frames (i.e., frames without respiratory motion) from the multi-angle data for reconstruction. At each imaging time point, we retained about 30 static frames from the 50 acquired data frames using respiratory gating for reconstruction. Fig. R6 clearly

compares the single-pulse 3D images, the reconstruction using the STINT method with respiratory gating, and the results using the STINT method directly (Fig. R6a). Overall, the reconstruction results using the STINT method, with or without respiratory gating, are superior to those of single-pulse imaging, as the 1024 transducer elements are severely spatially-undersampled for the whole-body FOV, which is consistent with the leaf vein tests (Fig. R2). Respiratory motion appears to have no macroscopic impact on image quality, but the effect of respiratory gating is evident in the details. In the small vessels of the liver and the iliac artery, respiratory gating significantly improves the visibility and definition of these structures (Fig. R6b). To quantify this, the image sharpness is evaluated using the Sobel operator to detect gradients, and the standard deviation of the gradient magnitudes is calculated as the sharpness metric. Higher values indicate clearer images, while lower values suggest blurriness. Calculations show that the STINT method with respiratory gating improves image sharpness compared with the others (1.375 and 1.27 times for liver vessels, respectively; 1.25 and 1.07 times for the iliac artery, respectively), avoiding image blurring caused by motion (Fig. R6b). Additionally, this method hardly alters the existing image features in single-pulse results, as evidenced by the surface vessels, with a correlation coefficient for the cross section of approximately 0.989 (Fig. R6c). In summary, we have quantitatively assessed the image blurring caused by respiratory motion and addressed this issue effectively. We incorporated these analysis into the final part of the Results section and supplemented it with Fig. S11.

Fig. R6 (Fig. S11 in the revised manuscript). Quantification of respiratory motion effects in the

STINT method.

a Comparison of whole-body imaging using single-pulse and STINT methods with and without respiratory gating. **b, c** Comparison of imaging details, with different regions corresponding to the boxes in **a**. RG, respiratory gating. SP, image sharpness.

Revisions:

Results section:

Lines 590-603:

“We further utilized whole-body imaging to evaluate the impact of respiratory motion on the STINT method (the first implementation). Apparently, the reconstruction results using the STINT method, with or without respiratory gating, are superior to those of single-pulse imaging, as the 1024 elements are severely spatially-undersampled for the whole-body FOV (Fig. S11a), which is consistent with the leaf vein results. The benefits of respiratory gating are evident in reconstructing the details, particularly in the small vessels of the liver and the iliac artery, where it significantly improves the visibility and definition of these structures (Fig. S11b). To quantify this, the image sharpness is evaluated using the Sobel operator to detect gradients, and the standard deviation of the gradient magnitudes is calculated as the sharpness metric. Higher values indicate clearer images, while lower values suggest blurriness. Calculations show that the STINT method with respiratory gating improves image sharpness compared with the others (1.375 and 1.27 times for liver vessels, respectively; 1.25 and 1.07 times for the iliac artery, respectively), avoiding image blurring caused by motion (Fig. S11b). Additionally, this method hardly alters the existing image features in single-pulse results, as evidenced by the surface vessels, with a correlation coefficient for the cross-section of approximately 0.989 (Fig. S11c). In summary, we have verified the outstanding performance of the STINT method from multiple aspects.”

Supplementary Materials:

Fig. S11 (please refer to the end of this file to prevent confusion of figure numbers)

Other comments:

* **Figures could be made more readable with minor adjustments, such as clearly marking the ROI in Figure 1b, adding a grayscale colormap in Figure 2e, and labeling color-coded brain regions in Figure 3a. Scale bars in Fig.3g would be beneficial to better grasp the depth of the circle of Willis.**

Reply:

Thanks for your suggestions. We have revised the figures in accordance with your suggestions.

Revisions:

Figures:

Fig. 1b, Fig. 2e, Fig. 2f, Fig. 3a, and Fig. 3g (please refer to the end of this file to prevent confusion of figure numbers)

* **In Supplementary Video 5: it seems left and right images (gray scale and depth color coded) are desynchronized at some point. Please correct or explain why this happens if relevant.**

Reply:

Thank you for pointing this out. The desynchronization in Movie S5 was due to an operational error during production. We have recreated the movie to correct the issue.

Revisions:

Supplementary Materials:

Movie S5 (please refer to the video files in this submission)

* **l. 501-502:** images were reconstructed for each set of 1024 signals at a given angle and then summed ('superimposed' in the text). Why not reconstructing directly from the 10*1024 signals from all virtual elements?

Reply:

Reconstructing 1024 elements in 10 steps and then summing the results is equivalent to reconstructing directly from the full 10×1024 elements because back-projection¹¹ is a linear operation. The reason we do not directly use the full 10×1024 elements in one step is due to the GPU memory limitations, which would cause overflow. Upgrading to a more advanced GPU with higher memory capacity would allow for direct reconstruction from the full set of elements, supporting larger parallel computations. We have added relevant explanations in the text.

Revisions:

Results section:

Lines 522-525:

“It should be noted that reconstructing images directly from all-angle data is equivalent to reconstructing separately and then summing the results, as the universal back-projection⁵⁹ is a linear reconstruction process. However, the former approach requires higher GPU memory, which can be addressed with a more advanced GPU that supports larger memory and parallel computations.”

* **supp inf, l.89:** ‘after fabrication’ Did the authors mean ‘after calibration’?

Reply:

Yes, thank you for pointing out the spelling errors. We have made the corrections and double-checked the entire manuscript.

Revisions:

Supplementary Materials:

Fig. S3 (please refer to the end of this file to prevent confusion of figure numbers)

In summary, Wang et al. have made a groundbreaking contribution to the field of photoacoustic tomography, particularly in terms of speed, resolution and FOV. By emphasizing key technical details in the main text and clarifying certain physiological and methodological aspects, the manuscript would further enhance its impact and provide readers with a more comprehensive understanding of the system’s capabilities. We commend the authors for their innovative work and recommend for publication once the above comments are addressed.

Reply:

Thank you for your positive feedback and suggestions. We have addressed all comments and believe the revisions have strengthened the manuscript. We appreciate your recommendation for publication.

Reviewer #2:

The authors demonstrate a new 3D panoramic photoacoustic computed tomography (PACT) system for real-time, cross-regional physiological dynamic imaging of the system. This study achieves high-quality images with 3D photoacoustic tomography, especially in the detailed visualization of vascular and organ structures, which is technically commendable. Although this study demonstrates the advantages of 3D PanoPACT in high spatiotemporal resolution and wide field of view (FOV), there are also some noteworthy issues and comments:

1. Despite the high imaging quality, the paper should focus more on specific applications of the technology. Readers are looking not just for clear images but also for how this technology can solve real biological or physiological questions. Enhancing the focus on practical applications would increase the relevance and value of this work.

Reply:

Thanks for the valuable comments. We appreciate your perspective that imaging systems ultimately serve specific biomedical applications. However, the focus of this study is to introduce the design philosophy, parameter calibration, and imaging strategies of a novel photoacoustic imaging device—3D-PanoPACT—and to demonstrate its broad applicability and significant potential for practical biological research. Obtaining clear, large field-of-view (FOV), high-spatiotemporal-resolution images is a goal for biomedical imaging researchers and also provides pathways to address cutting-edge biological questions. To this end, we present diverse biomedical application examples to highlight the versatility of 3D-PanoPACT. By performing real-time dynamic imaging of the hepatic vascular network, we extracted information related to the pulse wave and constructed 3D arterial and pulse phase gradient maps, which may serve as a pathway for assessing overall hepatic hemodynamics or as an indicator for certain liver diseases (Fig. 2). By imaging the neurovascular coupling response under electrical stimulation, we confirmed 3D-PanoPACT's ability to rapidly capture whole-brain physiological changes, thereby offering a potential solution for studying deep-brain neural activities (Fig. 4f-k). In particular, in the case of monitoring pharmacological responses in the whole-brain vasculature, the excellent deep-tissue imaging capability of 3D-PanoPACT enabled real-time functional imaging of the Circle of Willis and the accompanied changes in other cerebral vessels, highlighting its significance in noninvasively elucidating the hemodynamic mechanisms of the whole-brain vasculature and in the study of cerebrovascular diseases (Fig. 3 and 4a-e). The 3D-PanoPACT system demonstrates its unique advantages in cross-regional and multi-organ studies through whole-body 3D dynamic visualization (Fig. 5) and real-time tracking of small-molecule probes at the whole-body scale (Fig. 6), which remains challenging for other PACT systems. The significant practical value of 3D-PanoPACT is not limited to pharmacokinetic studies but also provides new pathways for a range of whole-body investigations, such as studies of systemic blood and lymphatic circulation, as well as tumor diffusion and metastasis throughout the body. We have incorporated the above explanations into the Discussion section.

Revisions:

Discussion section:

Lines 605-607:

“In this study, we introduced the design philosophy, parameter calibration, and imaging strategies of a novel PA imaging device—3D-PanoPACT—and demonstrated its broad applicability and significant potential for practical biological research.”

Lines 717-725:

“To highlight the versatility of 3D-PanoPACT, we have presented several key applications. Real-time imaging of the hepatic vascular network allowed for pulse wave extraction and 3D arterial mapping, serving

as indicators for hepatic hemodynamics and liver disease. Imaging neurovascular coupling confirmed 3D-PanoPACT's capability to rapidly capture whole-brain physiological changes, facilitating deep-brain neural activity studies. Monitoring pharmacological responses in the whole-brain vasculature enabled functional imaging of the Circle of Willis and other cerebral vessels, underscoring its significance in cerebrovascular research. Additionally, 3D-PanoPACT demonstrated unique advantages in cross-regional and multi-organ studies through whole-body 3D dynamic visualization and real-time small-molecule probe tracking, opening new pathways for investigating systemic pharmacokinetics, circulation, and tumor metastasis."

2. The authors claim that 3D-PanoPACT enables “cross-regional real-time visualization of systemic physiology and dynamics from organ to whole-body level” and emphasizes its “dynamic imaging with unprecedented quality of the whole trunk” to provide insights into multi-organ physiology (Abstract). To fully support these claims, it's necessary to include an experiment tracking a small-molecule probe's metabolic pathway in vivo. Without this, it's challenging to see how this approach offers an advantage over established methods like PET, MRI, or fluorescence imaging for functional, whole-body tracking. This application would highlight the unique strengths of 3D photoacoustic imaging and its relevance to biological research.

Reply:

Thanks for the valuable comments. We appreciate your suggestion regarding the inclusion of an experiment tracking a small-molecule probe's metabolic pathway in vivo to fully support our claims about 3D-PanoPACT's capabilities. In response, we have included a new experiment in our manuscript that tracks the metabolic pathway of a small-molecule probe at whole-body scale. This experiment showcases 3D-PanoPACT's ability to provide real-time, high-resolution imaging of the probe's distribution and metabolism across multiple organs, thereby offering a distinct advantage over other imaging modalities. We believe this addition strengthens our case for 3D-PanoPACT as a powerful tool for functional, whole-body tracking in biological research. The added content is as follows:

The newly added content in the Results section:

Title: High-spatiotemporal-resolution tracking of small molecule metabolic pathways at whole-body scale

To further illustrate the practical value of 3D-PanoPACT in biological applications, we tracked a small molecule probe, A1094, through its metabolic pathways in the whole body, allowing us to capture the intricate changes and developments occurring across different organs over an extended period. Tracking probes at the whole-body scale necessitates that the imaging system concurrently exhibits high spatiotemporal resolution and an extensive imaging FOV. From the perspective of signal excitation, an engineered diffuser with a divergence angle of 60° was installed at the bottom of the array. This setup enables uniform illumination within an approximate 12cm diameter range at the center of the array, thereby allowing a single laser pulse to simultaneously excite the entire mouse body and collect PA signals. Considering the inverse square relationship between energy density and spot diameter, the laser operated at its maximum output energy (single-pulse energy of 500 mJ @1064 nm) to ensure a high SNR in the raw data, which also limited it to operate at a repetition rate of 10 Hz to prevent damage. For acoustic detection, we adopted a strategy similar to that described in the aforementioned brain functional imaging. High-quality whole-body images can be obtained by simply rotating the array through a small angle, without the need for additional translational movement (Fig. R7a). In each whole-body imaging session, the transducer array

rotated 15° around the central axis within 5 s, acquiring 50 frames of raw PA data, which were then reconstructed into a 3D image using the STINT method (refer to the final part of the Results section for details). The array subsequently rotates in the opposite direction within 5 s to complete the next imaging session. Utilizing small-angle reciprocating rotation of the array and the STINT method, 3D-PanoPACT achieved high temporal resolution by completing 12 whole-body imaging sessions within 1 minute (equivalent to a frame rate of 0.2 Hz), thus fulfilling a broad spectrum of biological application requirements. It should be emphasized that to minimize the stochastic vibrations induced by mechanical rotation, the rotational velocity of the array was deliberately reduced. Consequently, this approach alleviated the spatial resolution degradation at the peripheral regions of the extensive FOV during STINT reconstruction. The exceptional quality of the whole-body 3D imaging with clear vascular networks and organ morphologies corroborated the efficacy of this method (Fig. R7b). Upon comparison, it was observed that the imaging results achieved high spatial resolution and were largely consistent with those in Fig. 5b across most of the trunk region, with resolution degradation only occurring at the peripheral edges (such as the limbs, neck, and tail), which was attributed to the spatial impulse response of the ultrasound transducer (Fig. R7c). Overall, by utilizing wide-field illumination and the Fibonacci grid arrangement of the array elements, 3D-PanoPACT achieves high spatiotemporal resolution imaging at whole-body scale with only a small angular rotation of the detection array. Concurrently, the streamlined scanning approach and reconstruction methods enable long-duration continuous imaging and batch data processing, making it highly suitable for biological applications involving the tracking of small molecule probes.

In this study, we selected a small molecule probe, A1094, as the tracer to visualize its metabolic pathways throughout the whole body. A1094 is a stable and rapidly cleared NIR-II imaging agent that has shown potential for clinical translation due to its sufficient blood solubility, good interference resistance, and pH/oxidative/metabolic stability¹². In addition, its absorption peak is close to the fundamental output wavelength of the Nd:YAG laser (i.e., 1064 nm), making it suitable for deep-tissue PA imaging (Fig. R7c). Therefore, the imaging wavelength was selected to be 1064 nm in this application (Fig. R7d). During the experimental procedure, a syringe pump was utilized to administer 0.4 ml prepared A1094 solution into the animal via a tail vein indwelling needle within 90 s (refer to Methods section for details). The onset of injection was designated as the zero point of tracking time, and continuous whole-body tracking consisting of 120 frames was completed within the following 10 minutes. Fig. R7h illustrates the whole-body tracking results at time points 0 s, 10 s, 60 s, and 500 s (refer to Methods section for details), thereby evidencing the capacity of 3D-PanoPACT to record the metabolic trajectory of the A1094 probe with high spatiotemporal fidelity across the whole body.

To quantify the interregional variations of A1094, registrations were performed on the reconstructed image sequence, and the percentage changes in voxel amplitude relative to the initial state were calculated for different organs, thereby generating corresponding time courses. Employing empirical models to fit the time courses enables the extraction of variation of A1094 from the fitted curves (refer to Methods section for details). From the fitted curves, we can extract several valuable parameters, such as the earliest time to reach the peak t_p , and the corresponding maximum percentage enhancement C_m . In the circulatory system, A1094 first reached the heart after injection, leading to the earliest signal changes, with $t_p \sim 88$ s. The iliac artery, being a branch of the aorta supplying the lower trunk, exhibited slightly slower signal changes, with $t_p \sim 145$ s (Fig. R7e). The C_m value of the heart ($\sim 14\%$) was weaker than that in the iliac artery ($\sim 51\%$), possibly due to the heart's relatively stable coronary arterial blood supply system, which made it less susceptible to changes in blood composition compared to peripheral vessels¹³ (Fig. R7e). The relatively slow signal decay may be attributed to the extended administration time coupled with increased blood flow associated with abdominal agent metabolism. The sustained enhancement in liver, with $t_p \sim 505$ s and $C_m \sim 75\%$, further underscored the liver's pivotal role in the metabolism of A1094 (Fig. R7f). Additionally, the observed ~ 55 s delay of intestine signal change compared to

the liver suggested A1094's metabolic route through the liver and then via the biliary tract to the intestine¹⁴. The $t_p \sim 300$ s and $C_m \sim 41\%$ indicated that the intestine, although involved in A1094 metabolism, did not play a dominant role (Fig. R7f). We also found that the mesenteric artery, with $t_p \sim 434$ s and $C_m \sim 61\%$, facilitated the hepatoenteral circulation mechanism, serving as a crucial link in maintaining the metabolic activities between the liver and the intestine^{14, 15} (Fig. R7f). Given that the ureter was directly connected to the kidneys, the observed signal changes in the ureter ($t_p \sim 263$ s and $C_m \sim 26\%$) indicated that the kidneys were also involved in the metabolism of A1094 (Fig. R7g). This metabolic pathway was distinct from that of another commonly used probe, ICG¹⁶. In addition to the ureter signal, which initially increased and subsequently decreased, the bladder signal exhibited a sustained increase, with $C_m \sim 49\%$, approximately double that of the ureter C_m value ($\sim 26\%$). This observation may be attributed to the bilateral ureters collectively channeling urine containing metabolites into the bladder, which, to some extent, underscores the efficacy of this analytical approach (Fig. R7g). In Movie S6, the dynamic cross-regional tracking of A094 across the whole body was visually demonstrated using 3D-PanoPACT. Compared to PET, MRI, and fluorescence imaging, 3D-PanoPACT offers unique advantages for small-molecule probe tracking. PET involves ionizing radiation, limiting its use for repeated imaging and long-term tracking, and has poor spatial resolution. MRI has lower sensitivity for certain molecular targets and often suffers from low temporal resolution and high cost. Fluorescence imaging is limited by shallow penetration and lower spatial resolution due to light scattering and can suffer from photobleaching and phototoxicity. In contrast, 3D-PanoPACT provides non-invasive, real-time, high-resolution imaging at deeper tissue depths with abundant and highly specific optical absorption contrast, making it a powerful tool for tracking small-molecule probes and studying systemic pharmacokinetics. These results not only highlight the high spatiotemporal resolution imaging capabilities of 3D-PanoPACT for large-scale 3D FOV but also further underscore its utility and potential for studying whole-body biological processes in practical applications.

The newly added prospects in the Discussion section:

By further increasing the density of array elements or incorporating deep learning methodologies¹⁷, the single-pulse FOV could potentially be expanded to a whole-body scale. This advancement will not only enable the high-speed tracking of dynamic changes in probe concentration throughout the entire body but, when combined with data analysis techniques such as optical flow methods¹⁸, could also facilitate the generation of 3D vector maps depicting the migration of probes¹⁹. This potential of 3D-PanoPACT may offer novel perspectives for research in fields such as metabolomics and oncology, thereby significantly enhancing our understanding of complex biological processes.

The newly added technical details in the Methods section:

In whole-body probe tracking, a total of 120 time-point three-dimensional (3D) image sequences were acquired with an interval of 5 s, and each time-point image was reconstructed using the STINT method. In Fig. R7h, the 3D image sequence was first subjected to frame-by-frame rigid registration. Taking the image before administration as a reference Img_{ref} , the 3D mapping of relative enhancement $RE(t)$ of the image acquired at time t $Img(t)$ can be calculated by processing the sequence of each voxel:

$$RE(t) = \frac{Img(t) - Img_{ref}}{Img_{ref}} \times 100\%$$

And voxels with RE value below 10% were considered to be affected by noise and were set to zero. In Fig. R7e-g, fitting of the raw data of enhancement was performed using two empirical models to intuitively analyze the

probe metabolism dynamics across different organs. For changes that initially rose rapidly and then declined gradually (the iliac artery, the heart, and the bladder), the fitted relative enhancement at time t $FRE(t)$ was generated using the following model²⁰:

$$FRE(t) = \begin{cases} 0 & 0 \leq t \leq t_0 \\ A \cdot [1 - e^{-\alpha(t-t_0)}]^q \cdot e^{-\beta(t-t_0)} & t_0 \leq t \end{cases}$$

The change of enhancement from an initial rise to a steady state within the time window (the liver, the mesenteric artery, the intestine, and the bladder) was modeled using an alternative model²¹:

$$FRE(t) = \begin{cases} 0 & 0 \leq t \leq t_0 \\ \frac{A}{1 + B \cdot e^{-\alpha(t-t_0)}} & t_0 \leq t \end{cases}$$

For each organ, the parameters C_m and t_p can be determined by $C_m = \max(FRE)$ and $FRE(t_p) = C_m$. The fitting procedure was performed using the built-in fitting toolbox in MATLAB 2018.

For whole-body probe tracking, 12-week-old mice (NU/NU, nude mouse, Charles River Co.) were used. A1094 (JHE0001, Suzhou Zhuoxinya Technology Co., Ltd., China) was selected as the small-molecule tracer. To prepare an A1094 solution, 0.3 mg of solid A1094 was first dissolved in 20 μ L of dimethyl sulfoxide (DMSO). Subsequently, 7 mL of distilled water was added, and the mixture was subjected to ultrasonication for 5 min to ensure complete dissolution. The injection dose of A1094 was 1 μ mol per kilogram of body weight, and calculations indicated that approximately 0.4 mL of the prepared solution was required for injection. During the experiment, the animals were maintained under light anesthesia with 1.2% isoflurane, and a 29G indwelling tail vein catheter was connected to a syringe. The injection was completed within 90 s using a precision syringe pump. The mouse behaved normally after the probe tracking tests.

(Due to space constraints, the figure is placed on the next page.)

Fig. R7 (Fig. 6 in the revised manuscript) High-spatiotemporal-resolution tracking of small molecules at whole-body scale using 3D-PanoPACT (Movie S6). **a** A schematic of the experimental setup and imaging strategy. **b** High-quality whole-body 3D imaging acquired within 5 s under this imaging strategy. **c** Comparison of the absorption spectra of A1094 with haemoglobin (HbO₂) and deoxygenated haemoglobin (Hb). Imaging wavelength: 1064 nm. **d-f** Relative enhancement in signal intensity across different organs following A1094 administration. Raw data are shown in grayscale, while fitted curves are depicted in colors. IA, iliac artery; HT, heart; LV, liver; MA, mesenteric artery; IN, intestines; UT, ureter; BD, bladder. **g** Mapping of whole-body cross-regional enhancement at 0 s, 10 s, 60 s, and 500 s post-A1094 administration, revealing molecular metabolic pathways.

Revisions:**Abstract:**

Lines 38-40:

“We tracked the metabolic pathways of a small-molecule probe across multiple organs using 3D-PanoPACT, highlighting its value in whole-body, cross-regional dynamic studies.”

Introduction section:

Lines 64-67:

“The unique combination of high-velocity and multiwavelength imaging abilities in two-dimensional (2D) PACT previously extended its performance to the tracking of metabolism dynamics¹⁶, monitoring physiological parameters, and recording exogenous probes for specialized applications¹⁷⁻¹⁹.”

Lines 83-85:

“Overall, the inherent trade-off between the FOV and spatiotemporal resolution limits the application of PACT in biomedical research, particularly in studying the dynamics of multiple organs and cross-regional systems throughout the body.”

Lines 99-101:

“We further tracked the metabolic pathways of a small-molecule probe across multiple organs with high spatiotemporal resolution using 3D-PanoPACT, thereby demonstrating its significant practical value in cross-regional dynamic studies at the whole-body scale.”

Results section:

Lines 383-469 (given the repetition of content and space limitations, please refer to the response in blue)

Discussion section:

Lines 735-742 (given the repetition of content and space limitations, please refer to the response in blue)

Methods section:

Lines 764-767, 963-979, 1010-1018 (given the repetition of content and space limitations, please refer to the response in blue)

Figures:

Fig. 6 (please refer to the end of this file to prevent confusion of figure numbers)

Supplementary Materials:

Fig. S11, Movie S6 (please refer to the end of this file and the video files in this submission)

Reference:

Ref. 16-19, 54-57, 74-78:

16. Lv, J. et al. Dynamic synthetic-scanning photoacoustic tracking monitors hepatic and renal clearance pathway of exogeneous probes in vivo. *Light: Science & Applications* 13, 304 (2024).

17. Li, L. et al. Single-impulse panoramic photoacoustic computed tomography of small-animal whole-body dynamics at high spatiotemporal resolution. *Nature biomedical engineering* 1, 0071 (2017).

18. Merčep, E., Herraiz, J.L., Deán-Ben, X.L. & Razansky, D. Transmission–reflection optoacoustic ultrasound (TROPUS) computed tomography of small animals. *Light: Science & Applications* 8, 18 (2019).

19. Olefir, I. et al. Spatial and spectral mapping and decomposition of neural dynamics and organization of the mouse brain with multispectral optoacoustic tomography. *Cell Reports* 26, 2833-2846. e2833 (2019).

54. Huang, Y. et al. Small-Molecule Absorber A1094 as a Stable and Fast-Clearing NIR-II Imaging Agent. *ChemMedChem* 16, 2497-2503 (2021).

55. Bernanke, D.H. & Velkey, J.M. Development of the coronary blood supply: changing concepts and current ideas. *The Anatomical Record: An Official Publication of the American Association of Anatomists*

269, 198-208 (2002).

56. von Richter, O. et al. Determination of in vivo absorption, metabolism, and transport of drugs by the human intestinal wall and liver with a novel perfusion technique. *Clinical Pharmacology & Therapeutics* 70, 217-227 (2001).

57. Desmettre, T., Devoisselle, J. & Mordon, S. Fluorescence properties and metabolic features of indocyanine green (ICG) as related to angiography. *Survey of ophthalmology* 45, 15-27 (2000).

74. Deng, H., Qiao, H., Dai, Q. & Ma, C. Deep learning in photoacoustic imaging: a review. *Journal of Biomedical Optics* 26, 040901-040901 (2021).

75. Van den Berg, P., Daoudi, K. & Steenbergen, W. Review of photoacoustic flow imaging: its current state and its promises. *Photoacoustics* 3, 89-99 (2015).

76. Zhang, Y., Olick-Gibson, J., Khadria, A. & Wang, L.V. Photoacoustic vector tomography for deep haemodynamic imaging. *Nature biomedical engineering* 8, 701-711 (2024).

77. Saito, S. et al. Assessment of liver function in thioacetamide-induced rat acute liver injury using an empirical mathematical model and dynamic contrast-enhanced MRI with Gd-EOB-DTPA. *Journal of Magnetic Resonance Imaging* 36, 1483-1489 (2012).

78. Wang, H. et al. Local and global function model of the liver. *International Journal of Radiation Oncology* Biology* Physics* 94, 181-188 (2016).

3. The paper demonstrates both whole-body and brain functional imaging of the mouse, but the rationale for combining these isn't entirely clear. Is this meant to showcase versatility, or is there a specific scientific question being addressed by both examples? A brief explanation of the purpose behind this combination would help readers better understand the study's overall objective.

Reply:

Thank you for the insightful comments. In our study, we illustrated the comprehensive imaging performance of 3D-PanoPACT by extending from single-organ to whole-body FOV and from structural to functional imaging. Given that high-speed 3D imaging of the whole brain and whole body are significant research areas, presenting the whole-brain and whole-body imaging results separately helps to highlight the broad applicability of 3D-PanoPACT in addressing diverse imaging needs and its potential for solving key biological questions. Therefore, the emphasis on the anatomical and functional dynamics of the Circle of Willis, whole-body 3D dynamics, and whole-body probe tracking is not directed towards a single scientific question, but rather highlights the versatility of 3D-PanoPACT to address a diverse range of cutting-edge biological problems. Relevant explanations have been incorporated into the Introduction section to facilitate a better understanding of the study's overall objectives.

Revisions:

Introduction section:

Lines 101-106:

“Therefore, we illustrated the comprehensive imaging performance of 3D-PanoPACT by extending from single-organ to whole-body FOV and from structural to functional imaging. Given that high-speed 3D imaging of the whole brain and whole body are significant research areas, presenting the whole-brain and whole-body imaging results separately helps to highlight the broad applicability of 3D-PanoPACT in addressing diverse imaging needs and its potential for solving key biological questions.”

4. The limitations of this system are not discussed in the paper. Adding some discussion about the imaging depth limits, signal-to-noise considerations, or any areas where resolution might be affected would provide a balanced perspective and inform future improvements.

Reply:

Thank you for the valuable comments. We discuss several rational prospects as limitations for further improvement:

(1) Improvement of ultrasound transducer materials.

The current system employs traditional piezoelectric ceramic composite materials for its transducers, which, despite their high robustness, have limited receiving bandwidth. Since the transducers in PA imaging are used solely for receiving acoustic waves and not for transmission, replacing them with a Polyvinylidene Fluoride (PVDF) array that has a wider receiving bandwidth may further enhance the system's spatial resolution and imaging quality.

(2) Optimization of temporal resolution.

Despite its single-pulse 3D imaging capability, the low laser pulse repetition rate limits 3D-PanoPACT's ability to capture ultrafast physiological processes in this study. For instance, in imaging liver dynamics, we successfully extracted the heartbeat frequency from the temporal sequence but the detailed dynamics of cardiac substructures remained elusive. This limitation arises because these dynamics typically manifest as high-frequency harmonics of the heartbeat frequency, and the current imaging frame rate of 25 Hz fails to meet the Nyquist sampling criterion, thereby making it difficult to resolve this information. For reference in echocardiography imaging³, an imaging frame rate exceeding 100 Hz may significantly enhance the capability to capture rapid physiological phenomena but also imposes higher demands on high-energy, high-repetition-rate pulsed laser equipment. With a higher laser repetition rate, increasing the rotation speed of the array without introducing additional vibrations could further optimize the temporal resolution of the STINT method. The implementation involves reducing the rotational stage load by changing the connection mode between the array and the preamplifier and data acquisition modules from direct plug-ins to cabled connections, presenting higher engineering requirements on the electromagnetic shielding. In this way, the rotational stage only supports the weight of the transducer array without incorporating additional housings.

(3) SIR correction for large FOV optimizes spatial resolution.

The SIR arises from the non-ideal nature of the transducers. In whole-body probe tracking, the edge of FOV is not impacted by undersampling after the application of the STINT method, making the effects of SIR more pronounced, specifically that the resolution decreases closer to the edge of FOV. Accurate calibration of the SIR coupled with spatial deconvolution, or the implementation of model-based image reconstruction incorporating the SIR, may potentially mitigate the resolution degradation, thereby further enhancing the imaging quality^{22, 23}.

(4) Further exploration of imaging depth.

Employing omnidirectional laser illumination can significantly enhance the raw data's SNR from deep tissues. On this basis, the development of advanced segment-based iterative solvers^{24, 25}, or other methods for correcting acoustic heterogeneity for 3D reconstruction may substantially improve the imaging SNR of deep-seated features. We have added this content in the Discussion section.

Revisions:

Discussion section:

Lines 687-716:

“While 3D-PACT exhibits superior imaging performance, it still has some limitations and areas for further improvement. We briefly discuss several rational prospects: (1) Improvement of ultrasound transducer materials. The current system employs traditional piezoelectric ceramic composite materials for its

transducers, which, despite their high robustness, have limited receiving bandwidth. Since the transducers in PA imaging are used solely for receiving acoustic waves and not for transmission, replacing them with a Polyvinylidene Fluoride (PVDF) array featuring a wider receiving bandwidth may further enhance the system's spatial resolution and imaging quality. (2) Optimization of temporal resolution. Despite its single-pulse 3D imaging capability, the low laser pulse repetition rate limits 3D-PanoPACT's ability to capture ultrafast physiological processes in this study. For instance, in imaging liver dynamics, we successfully extracted the heartbeat frequency from the temporal sequence but the detailed dynamics of cardiac substructures remained elusive. This limitation arises because these dynamics typically manifest as high-frequency harmonics of the heartbeat frequency, and the current imaging frame rate of 25 Hz fails to meet the Nyquist sampling criterion, thereby making it difficult to resolve this information. For reference in echocardiography imaging³¹, an imaging frame rate exceeding 100 Hz may significantly enhance the capability to capture rapid physiological phenomena but also impose higher demands on high-energy, high-repetition-rate pulsed laser equipment. With a higher laser repetition rate, increasing the rotation speed of the array without introducing additional vibrations could further optimize the temporal resolution of the STINT method. The implementation involves reducing the rotational stage load by changing the connection mode between the array and the preamplifier and data acquisition modules from direct plug-ins to cabled connections, presenting higher engineering requirements on the electromagnetic shielding. In this way, the rotational stage only supports the weight of the transducer array without incorporating additional housings. (3) SIR correction for large FOV optimizes spatial resolution. The SIR arises from the non-ideal nature of the transducers. In whole-body probe tracking, the edge of FOV is not impacted by undersampling after the application of the STINT method, making the effects of SIR more pronounced, specifically that the resolution decreases closer to the edge of FOV. Accurate calibration of the SIR coupled with spatial deconvolution, or the implementation of model-based image reconstruction incorporating the SIR, may potentially mitigate the resolution degradation, thereby further enhancing the imaging quality^{60, 63}. (4) Further exploration of imaging depth. Employing omnidirectional laser illumination can significantly enhance the raw data's SNR from deep tissues. On this basis, the development of advanced segment-based iterative solvers^{64, 65}, or other methods for correcting acoustic heterogeneity for 3D reconstruction may substantially improve the imaging SNR of deep-seated features.”

Reference:

Ref. 31, 60, 63-65:

31. Zacchigna, S. et al. Towards standardization of echocardiography for the evaluation of left ventricular function in adult rodents: a position paper of the ESC Working Group on Myocardial Function. *Cardiovascular research* 117, 43-59 (2021).
60. Lu, Q.-B. et al. Probing the spatial impulse response of ultrahigh-frequency ultrasonic transducers with photoacoustic waves. *Physical Review Applied* 14, 034026 (2020).
63. Mitsuhashi, K., Wang, K. & Anastasio, M.A. Investigation of the far-field approximation for modeling a transducer's spatial impulse response in photoacoustic computed tomography. *Photoacoustics* 2, 21-32 (2014).
64. Cai, C. et al. Feature coupling photoacoustic computed tomography for joint reconstruction of initial pressure and sound speed in vivo. *Biomedical optics express* 10, 3447-3462 (2019).
65. Deng, K. et al. Multi-segmented feature coupling for jointly reconstructing initial pressure and speed of sound in photoacoustic computed tomography. *Journal of Biomedical Optics* 27, 076001-076001 (2022).

5. In line 455, the manuscript describes the application of depth compensation during image post-processing: “In the image post-processing, a depth compensation was applied to enhance the PA amplitude from the deep tissue (e7.5×depth mm for trunk imaging; e6.5×depth mm for brain imaging).” Could the authors clarify how the gain coefficients (e7.5 and e6.5) were determined for depth compensation? Specifically, it would be helpful to understand the methodology used to derive these values and whether they are empirically or theoretically based.

Reply:

Thank you for your insightful comments regarding the determination of the gain coefficients for depth compensation in our image processing. The gain coefficients used for compensating optical fluence attenuation were estimated through straightforward experimental tests, and we provide additional details here. First, we constructed a simple testing setup. We utilized two quartz plates (50 mm × 50 mm², 2.5 mm thick), with the lower plate horizontally secured to a support stand and the upper plate horizontally affixed to a precision elevation stage, allowing precise control of the distance h between them. The space between the quartz plates was used for placing the sample to be measured. The collimated laser output with a diameter of ~ 10 mm was split into two beams through a 1:9 beam splitter. The 90% portion of the energy traveled vertically downward through the quartz plates and reached optical energy meter 1, while the remaining 10% continued to travel horizontally and reached optical energy meter 2, and the laser output was synchronized with data acquisition from the two optical energy meters. The energy meters had a detection area of ~ 30 mm in diameter, which was sufficient to capture all the energy within the laser beam (Fig. R8). Prior to testing, without placing the sample, h was adjusted to 4 mm, and the average readings of 50 laser pulses on optical energy meters 1 and 2 (denoted as I_1 and I_2 , respectively) were recorded, which can be expressed as:

$$I_1 = \alpha \cdot I_2 \cdot \tau$$

where α represents the total light intensity conversion coefficient, and τ represents the light attenuation of the quartz glass.

Subsequently, we rapidly excised fresh organs from the experimental animals (selected from the same strain and age as the animals imaged in this study) for use as test samples. The upper quartz plate was elevated, the sample was inserted, and then the distance h was readjusted to 4 mm, ensuring that the sample was compressed and fully occupied the interspace between the quartz plates. The procedure was repeated to obtain the readings from the two optical energy meters (denoted as I'_1 and I'_2 , respectively), which can be expressed according to Lambert-Beer law²⁶:

$$I'_1 = \alpha \cdot I'_2 \cdot \tau \cdot e^{-\mu_{\text{eff}} \cdot h}$$

By solving the two equations above, the optical attenuation coefficient of the sample can be calculated as:

$$\mu_{\text{eff}} = \frac{1}{h} \ln\left(\frac{I'_2 I_1}{I_1 I'_2}\right)$$

Using this method, we estimated μ_{eff} for the liver and brain. For each organ, measurements were taken from three animals and averaged to obtain the results. For the brain tissue, the estimated μ_{eff} for 1064 nm and 800 nm laser are 0.065 mm⁻¹ and 0.077 mm⁻¹, respectively. These parameters were utilized for optical fluence compensation in imaging whole-brain anatomies and dynamic functions. For liver tissue, the μ_{eff} at 1064 nm and 690 nm were estimated to be 0.075 mm⁻¹ and 0.133 mm⁻¹, respectively. The coefficients were applied for compensation in dynamic liver imaging and whole-body imaging. In the image processing, depth compensation was applied as $e^{\mu_{\text{eff}} \cdot \text{depth (mm)}}$ for different wavelengths to enhance the PA amplitude from deep tissues, with depth = 0 at the surface. It should also be noted that the empirical testing here serves merely as a rough estimate for optical fluence compensation.

In the previous manuscript, we made a unit error (mistakenly writing coefficients 0.065 and 0.075 as 6.5 and 7.5),

which has now been corrected. We have supplemented the revised manuscript with the above technical details in the Results and Methods sections and have added Fig. S5. We believe these contents would be helpful to understand the methodologies.

Fig. R8 (Fig. S5 in the revised manuscript). The setup for optical attenuation coefficient estimation in fresh tissue.

Revisions:

Results section:

Lines 187-199:

“In PA imaging, accurate optical fluence compensation is crucial for enhancing image fidelity and ensuring reliable quantitative analysis²⁹, especially for multi-wavelength unmixing. We employed empirical testing to estimate optical attenuation in freshly excised tissue for subsequent compensation. A simple setup was first constructed using two quartz plates, a collimated laser beam, a 1:9 beam splitter, and a pair of optical energy meters. One optical energy meter was used to detect the light passing through the quartz plates, while the other was used to monitor the energy fluctuations of the split beam as a normalization reference (Fig. S5). By comparing these measurements with and without a sample, we calculated the optical attenuation coefficient (μ_{eff}) for brain and liver tissues at different wavelengths based on the Lambert-Beer law³⁰, which relates optical attenuation to the properties of the material (see Methods for details). For brain tissue, μ_{eff} was 0.065 mm^{-1} at 1064 nm and 0.077 mm^{-1} at 800 nm , which were utilized for imaging whole-brain anatomies and dynamic functions. For liver tissue, μ_{eff} was 0.075 mm^{-1} at 1064 nm and 0.133 mm^{-1} at 690 nm , which were applied in dynamic liver imaging and whole-body imaging. These coefficients were then used in image processing to apply exponential depth compensation, enhancing the PA amplitude from deep tissues.”

Methods section:

Lines 857-890:

“Optical fluence compensation (title)

The gain coefficients used for compensating optical fluence attenuation were estimated through straightforward experimental tests. First, we constructed a simple testing setup. We utilized two quartz plates ($50 \text{ mm} \times 50 \text{ mm}^2$, 2.5 mm thick), with the lower plate horizontally secured to a support stand and the upper

plate horizontally affixed to a precision elevation stage, allowing precise control of the distance h between them. The space between the quartz plates was used for placing the sample to be measured. The collimated laser output with a diameter of ~ 10 mm was split into two beams through a 1:9 beam splitter (Fig. S5). The 90% portion of the energy traveled vertically downward through the quartz plates and reached optical energy meter 1, while the remaining 10% continued to travel horizontally and reached optical energy meter 2, and the laser output was synchronized with data acquisition from the two optical energy meters (Fig. S5). The energy meters had a detection area of ~ 30 mm in diameter, which was sufficient to capture all the energy within the laser beam (Fig. S5). Prior to testing, without placing the sample, h was adjusted to 4 mm, and the average readings of 50 laser pulses on optical energy meters 1 and 2 (denoted as I_1 and I_2 , respectively) were recorded, which can be expressed as:

$$I_1 = \alpha \cdot I_2 \cdot \tau \quad (4)$$

where α represents the total light intensity conversion coefficient, and τ represents the light attenuation of the quartz glass.

Subsequently, we rapidly excised fresh organs from the experimental animals (selected from the same strain and age as the animals imaged in this study) for use as test samples. The upper quartz plate was elevated, the sample was inserted, and then the distance h was readjusted to 4 mm, ensuring that the sample was compressed and fully occupied the interspace between the quartz plates. The procedure was repeated to obtain the readings from the two optical energy meters (denoted as I'_1 and I'_2 , respectively), which can be expressed according to Lambert-Beer law³⁰:

$$I'_1 = \alpha \cdot I'_2 \cdot \tau \cdot e^{-\mu_{\text{eff}} \cdot h} \quad (5)$$

By solving the two equations above, the optical attenuation coefficient of the sample can be calculated as:

$$\mu_{\text{eff}} = \frac{1}{h} \ln\left(\frac{I'_2 I_1}{I_1 I'_2}\right) \quad (6)$$

Using this method, we estimated μ_{eff} for the liver and brain. For each organ, measurements were taken from three animals and averaged to obtain the results. For the brain tissue, the estimated μ_{eff} for 1064 nm and 800 nm laser are 0.065 mm^{-1} and 0.077 mm^{-1} , respectively. These parameters were utilized for optical fluence compensation in imaging whole-brain anatomies and dynamic functions. For liver tissue, the μ_{eff} at 1064 nm and 690 nm were estimated to be 0.075 mm^{-1} and 0.133 mm^{-1} , respectively. The coefficients were applied for compensation in dynamic liver imaging and whole-body imaging. In the image processing, depth compensation was applied as $e^{\mu_{\text{eff}} \cdot \text{depth}(\text{mm})}$ for different wavelengths to enhance the PA amplitude from deep tissues, with $\text{depth} = 0$ at the surface. It should also be noted that the empirical testing here serves merely as a rough estimate for optical fluence compensation.”

Lines 907-908:

“In the image post-processing, depth compensations were applied to enhance the PA amplitude from the deep tissue (as described previously).”

Lines 928-929:

“It should be noted that dual-wavelength images require optical fluence compensation according to the corresponding coefficients (as described previously).”

Supplementary Materials:

Fig. S5 (please refer to the end of this file to prevent confusion of figure numbers)

Reference:

Ref. 29, 30:

29. Bu, S. et al. Model-based reconstruction integrated with fluence compensation for photoacoustic tomography. *IEEE transactions on biomedical engineering* 59, 1354-1363 (2012).

30. Oshina, I. & Spigulis, J. Beer–Lambert law for optical tissue diagnostics: current state of the art and the main limitations. *Journal of biomedical optics* 26, 100901-100901 (2021).

6. How long does it take to acquire a whole-body image? I think the slow imaging speed may constrain the real scenarios.

Reply:

In the small-molecule probe tracking experiment, 3D-PanoPACT acquired a whole-body image in 5 s with a combination of the STINT method and the wide-field laser illumination (Fig. R7a-c). This enabled the system to achieve high temporal resolution by completing 12 whole-body imaging sessions within 1 minute, equivalent to a frame rate of 0.2 Hz. In this scenario, 3D-PanoPACT distinctly revealed the metabolic pathways of small molecules across regions and multiple organs (Fig. R7e-h), indicating that the current imaging frame rate is suitable for pharmacokinetic applications.

It is worth noting that 3D-PanoPACT is capable of acquiring whole-body imaging from a single laser pulse (Fig. R6a), and its imaging frame rate is determined by the laser repetition rate. Although this whole-body imaging achieves video-rate speed, spatial undersampling leads to degraded image quality, thereby limiting its application scope in terms of spatial resolution. Therefore, the application of the STINT method balances spatial and temporal resolution, enabling high-definition whole-body real-time monitoring for non-ultrafast physiological dynamics. Moreover, for scenarios where a whole-body FOV is not requisite, 3D-PanoPACT can achieve high-fidelity video-rate imaging at the organ scale, such as imaging the liver dynamics in our study (Fig. R5).

Due to the small-angle array rotation inherent in the STINT method, achieving real-time high-quality video-rate 3D imaging at whole-body scale remains challenging, although further improvements in imaging speed while preserving image quality may be possible. Using a pulsed laser with a higher repetition rate coupled with a reduced rotational stage load (e.g., by changing the array connectors from plug-ins to cabled connection), the array can increase its rotation speed while operating smoothly, thereby reducing the time required for the same dataset and improving temporal resolution. We have added the necessary explanations in the Discussion section.

Revisions:

Discussion section:

Lines 693-706:

“(2) Optimization of temporal resolution. Despite its single-pulse 3D imaging capability, the low laser pulse repetition rate limits 3D-PanoPACT’s ability to capture ultrafast physiological processes in this study. For instance, in imaging liver dynamics, we successfully extracted the heartbeat frequency from the temporal sequence but the detailed dynamics of cardiac substructures remained elusive. This limitation arises because these dynamics typically manifest as high-frequency harmonics of the heartbeat frequency, and the current imaging frame rate of 25 Hz fails to meet the Nyquist sampling criterion, thereby making it difficult to resolve this information. For reference in echocardiography imaging³¹, an imaging frame rate exceeding 100 Hz may significantly enhance the capability to capture rapid physiological phenomena but also impose higher demands on high-energy, high-repetition-rate pulsed laser equipment. With a higher laser repetition rate, increasing the rotation speed of the array without introducing additional vibrations could further optimize the temporal resolution of the STINT method. The implementation involves reducing the rotational stage load by changing the connection mode between the array and the preamplifier and data acquisition modules from direct plug-ins to cabled connections, presenting higher engineering requirements on the electromagnetic shielding. In this way, the rotational stage only supports the weight of the transducer array without incorporating additional housings.”

7. In Figure 5d, sections 2, 3, and 4 don't seem to match those in Figure 5e (1, 2, 3). Could the authors check for a labeling error?

Reply:

Thanks for pointing this out. This is a labeling error in our figure preparation. We have corrected labels “2, 3, and 4” to “1, 2, and 3” in Fig. 5d.

Revisions:

Figures:

Fig. 5 (please refer to the end of this file to prevent confusion of figure numbers)

Reviewer #3:

Reply:

Thank you for your remarks and for the opportunity to participate in this review process.

Reference

1. Huang, C. et al. Dual-scan photoacoustic tomography for the imaging of vascular structure on foot. *IEEE Transactions on Ultrasonics, Ferroelectrics, and Frequency Control* **70**, 1703-1713 (2023).
2. Li, L. et al. Single-impulse panoramic photoacoustic computed tomography of small-animal whole-body dynamics at high spatiotemporal resolution. *Nature biomedical engineering* **1**, 0071 (2017).
3. Zacchigna, S. et al. Towards standardization of echocardiography for the evaluation of left ventricular function in adult rodents: a position paper of the ESC Working Group on Myocardial Function. *Cardiovascular research* **117**, 43-59 (2021).
4. Hirata, K., Kawakami, M. & O'Rourke, M.F. Pulse wave analysis and pulse wave velocity a review of blood pressure interpretation 100 years after Korotkov. *Circulation journal* **70**, 1231-1239 (2006).
5. Uyeda, J.W., LeBedis, C.A., Penn, D.R., Soto, J.A. & Anderson, S.W. Active hemorrhage and vascular injuries in splenic trauma: utility of the arterial phase in multidetector CT. *Radiology* **270**, 99-106 (2014).
6. Chen, C. et al. Peripheral microvessel area better predicts the severity of coronary stenosis of acute myocardial infarction patients over pulse wave velocity. *Scientific Reports* **14**, 28584 (2024).
7. Zeng, H.-K. et al. A comparative study on the efficacy of 10% hypertonic saline and equal volume of 20% mannitol in the treatment of experimentally induced cerebral edema in adult rats. *BMC neuroscience* **11**, 1-10 (2010).
8. Paczynski, R., He, Y., Diring, M. & Hsu, C. Multiple-dose mannitol reduces brain water content in a rat model of cortical infarction. *Stroke* **28**, 1437-1444 (1997).
9. Wang, X. et al. A skull-removed chronic cranial window for ultrasound and photoacoustic imaging of the rodent brain. *Frontiers in Neuroscience* **15**, 673740 (2021).
10. Chen, Y. et al. Photoacoustic mouse brain imaging using an optical Fabry-Pérot interferometric ultrasound sensor. *Frontiers in Neuroscience* **15**, 672788 (2021).
11. Xu, M. & Wang, L.V. Universal back-projection algorithm for photoacoustic computed tomography. *Physical Review E* **71**, 016706 (2005).
12. Huang, Y. et al. Small-Molecule Absorber A1094 as a Stable and Fast-Clearing NIR-II Imaging Agent. *ChemMedChem* **16**, 2497-2503 (2021).
13. Bernanke, D.H. & Velkey, J.M. Development of the coronary blood supply: changing concepts and current ideas. *The Anatomical Record: An Official Publication of the American Association of Anatomists* **269**, 198-208 (2002).
14. Lv, J. et al. Dynamic synthetic-scanning photoacoustic tracking monitors hepatic and renal clearance pathway of exogeneous probes in vivo. *Light: Science & Applications* **13**, 304 (2024).
15. von Richter, O. et al. Determination of in vivo absorption, metabolism, and transport of drugs by the human intestinal wall and liver with a novel perfusion technique. *Clinical Pharmacology & Therapeutics* **70**, 217-227 (2001).
16. Desmettre, T., Devoisselle, J. & Mordon, S. Fluorescence properties and metabolic features of indocyanine green (ICG) as related to angiography. *Survey of ophthalmology* **45**, 15-27 (2000).
17. Deng, H., Qiao, H., Dai, Q. & Ma, C. Deep learning in photoacoustic imaging: a review. *Journal of Biomedical Optics* **26**, 040901-040901 (2021).
18. Van den Berg, P., Daoudi, K. & Steenbergen, W. Review of photoacoustic flow imaging: its current state and its promises. *Photoacoustics* **3**, 89-99 (2015).
19. Zhang, Y., Olick-Gibson, J., Khadria, A. & Wang, L.V. Photoacoustic vector tomography for deep haemodynamic imaging. *Nature biomedical engineering* **8**, 701-711 (2024).
20. Saito, S. et al. Assessment of liver function in thioacetamide-induced rat acute liver injury using an

- empirical mathematical model and dynamic contrast-enhanced MRI with Gd-EOB-DTPA. *Journal of Magnetic Resonance Imaging* **36**, 1483-1489 (2012).
21. Wang, H. et al. Local and global function model of the liver. *International Journal of Radiation Oncology* Biology* Physics* **94**, 181-188 (2016).
 22. Lu, Q.-B. et al. Probing the spatial impulse response of ultrahigh-frequency ultrasonic transducers with photoacoustic waves. *Physical Review Applied* **14**, 034026 (2020).
 23. Mitsuhashi, K., Wang, K. & Anastasio, M.A. Investigation of the far-field approximation for modeling a transducer's spatial impulse response in photoacoustic computed tomography. *Photoacoustics* **2**, 21-32 (2014).
 24. Cai, C. et al. Feature coupling photoacoustic computed tomography for joint reconstruction of initial pressure and sound speed in vivo. *Biomedical optics express* **10**, 3447-3462 (2019).
 25. Deng, K. et al. Multi-segmented feature coupling for jointly reconstructing initial pressure and speed of sound in photoacoustic computed tomography. *Journal of Biomedical Optics* **27**, 076001-076001 (2022).
 26. Oshina, I. & Spigulis, J. Beer–Lambert law for optical tissue diagnostics: current state of the art and the main limitations. *Journal of biomedical optics* **26**, 100901-100901 (2021).

Figure Revisions

Figure 1. System design and imaging examples of 3D-PanoPACT.

a The schematic diagram of the 3D-PanoPACT system. HUTA, hemispherical ultrasonic transducer array; AMP, amplification circuits; DAQ, data acquisition module; RS, rotation stage; BC, beam combiner; FSM, front-silvered mirror; ED, engineered diffuser; TS, translation stage. **b** Real-time 3D imaging of the whole liver anatomy produced by the 3D-PanoPACT with separate displays of the different liver lobes attached. The image was acquired with a single laser pulse. MLL, middle liver lobe; LMLL, left middle liver lobe; LLLL, left lateral liver lobe. **c** The sequence diagram of 25 Hz single-wavelength mode and 10 Hz dual-wavelength mode. **d** The schematic diagrams of *in-vivo* trunk imaging in 3D-PanoPACT. **e** The schematic diagrams of *in-vivo* brain imaging in 3D-PanoPACT.

Figure 2. Real-time visualization of liver dynamics in 3D-PanoPACT (Movie S1).

a The 1064 nm real-time 3D imaging of the liver at 25 Hz. The images are taken at different points in a respiratory cycle, which was labelled in **b** with gray lines. **b** The raw signal of PA amplitude change (dark blue solid line) of the blood vessel (blue arrow in **a**) and the extracted heartbeat signal (light blue dashed line). **c** The Fourier transform of the raw data showing the respiratory frequency and heartbeat frequency, respectively. **d** The comparison of heartbeat signals extracted from blood vessels at different locations (the blue and orange arrows in **a**) with the phase delay clearly visible. The corresponding time windows are indicated by shaded areas in **b**. **e** Heartbeat-encoded arterial network mapping overlaid on the anatomy background in gray scale. **f** Mapping of pulse wave phase delay overlaid on the gray-scale anatomy background. SV, superficial blood vessels; HPV, hepatic portal vein; HA, hepatic artery; MA, mesenteric artery.

Figure 3. Label-free whole-brain anatomy in 3D-PanoPACT (Movie S4).

a Schematic diagrams illustrating regional segmentation of whole-brain imaging results from various observation perspectives. **b** The cut-open view of the deep brain with the cutting position and view orientation labelled in **a** (orange arrow). The image was depth-encoded along the Y-axis with the cutting position set to zero for better visualization. **c** The image of the cortical region depth-encoded along the Z-axis (yellow region in **a**). **d** The image of the deep-brain region depth-encoded along the Z-axis (blue region in **a**). **e** The 3D view of the Circle of Willis (displayed in red for emphasis) along with its constituent vascular structures. **f** The *in-situ* location of Circle of Willis from the perspective of the base of the brain. **g** Six coronal sections from the image (labelled in **a**) showing the imaging depth in 3D-PanoPACT. Each cross-section was obtained by averaging a 1mm thick 3D image in the thickness direction. OA, olfactory artery; AACAA, azygos of the anterior cerebral artery; LHV, longitudinal hippocampal vein; ISS, inferior sagittal sinus; DTV, dorsal thalamic vein; ACA, anterior cerebral artery; IC, internal carotid; RRV, rostral rhinal vein; SV, supraorbital vein; SSS, superior sagittal sinus; SS, sigmoid sinus; TS, transverse sinus; CRV, caudal rhinal vein; MCA, middle cerebral artery; RE, right eye; THA, transverse hippocampal arteries; PCA, posterior cerebral artery; ACoA, anterior communicating artery; PCoA, posterior communicating artery.

Figure 5. Label-free visualization of whole-trunk dynamics in 3D-PanoPACT (Movie S5).

a The steps to reconstruct dynamics of the whole trunk at 10 Hz. **b** The MIP and depth-encoded images from the frontal view of the whole trunk. RL, right lung; LL, left lung; MLL, middle liver lobe; LMLL, left middle liver lobe; LLLL, left lateral liver lobe; MA, mesenteric artery; SI, small intestine; ST, stomach; CE, cecum; SV, seminal vesicle; SC, spermatic cord; TE, testicle; PE, penis; AC, arteria cruralis. **c** The cross-sections from the frontal view of the whole-trunk image (orange arrows in **b**) at different time points, indicating that the imaging depth extends to a minimum of 20 mm. Each cross-section was obtained by averaging a 1mm thick 3D image in the thickness direction. **d** The MIP and depth-encoded images from the backside view of the whole trunk. IBAT, interscapular brown adipose tissue; SP, spleen; LK, left kidney; RK, right kidney. **e** The cross-sections from the backside view of the whole-trunk image (cyan arrows in **d**) at different time points.

Figure 6. High-spatiotemporal-resolution tracking of small molecules at whole-body scale using 3D-PanoPACT (Movie S6).

a A schematic of the experimental setup and imaging strategy. **b** High-quality whole-body 3D imaging acquired within 5 s under this imaging strategy. **c** Comparison of the absorption spectra of A1094 with haemoglobin (HbO₂) and deoxygenated haemoglobin (Hb). Imaging wavelength: 1064 nm. **d-f** Relative enhancement in signal intensity across different organs following A1094 administration. Raw data are shown in grayscale, while fitted curves are depicted in colors. IA, iliac artery; HT, heart; LV, liver; MA, mesenteric artery; IN, intestines; UT, ureter; BD, bladder. **g** Mapping of whole-body cross-regional enhancement at 0 s, 10 s, 60 s, and 500 s post-A1094 administration, revealing molecular metabolic pathways.

Figure 7. The spatiotemporal-integration (STINT) method.

a The reconstruction of the original hemispherical array with 1024 elements. **b** The schematic diagram and reconstruction results of the STINT method. The leaf skeleton was used as a phantom for a more intuitive illustration. **c-g** The first implementation of STINT. **c** The array motion timing in brain imaging. The color bar

indicates the order of array rotation. **d** The overall timing diagram of array rotations and the synchronization of the laser pulse with the array motion in each cycle in brain imaging. **e** The overall timing diagram of array reciprocating rotations and the synchronization of the laser pulse with the motions in whole-body probe tracking. **f** The frame-to-frame cross-correlation matrix. **g** The phase curve generated from the correlation matrix. Red star symbols indicate the static frames used for reconstruction. **h-p** The second implementation of STINT. **h** The relative position of the animal to the FOV at a certain scanning position in the X-Y plane. The red point indicates the position of the animal. **i** Synchronization sequence diagram of dual-wavelength laser pulse outputs, array rotation motion, and data acquisition at a certain scanning position (only single-wavelength raw data is shown here). **j-k** The frame-to-frame cross-correlation matrix and the generated phase curve. **l** Schematic diagram of the scanning process in the X-Y plane (intervals of 12 mm along the X-axis and 20 mm along the Y-axis) and the expanded FOV (the red point represents the position of the animal). **m** The phase curves at different scanning positions. The complete respiratory cycle is enclosed by black dashed rectangles, with the range of array rotation angles annotated. **n** The aligned phase curves from different positions. The orange dashed line indicates the raw data used for single-frame image reconstruction. **o** The process of integrating FOVs from different scanning positions into a large FOV in single-frame whole-trunk image reconstruction. The red dashed box represents the FOV at a certain scanning position. **p** 10 Hz imaging results of the whole-trunk dynamics using the STINT method.

Supplementary Materials Revisions:

Supplementary Figures:

Figure S2. Design of transducer element size.

a The simulation setup. **b** Simulation results of the relation between element radius and detected PA amplitude. **c** Simulation of transducer element sensitivity distribution.

Figure S3. Geometric calibrations in 3D-PanoPACT.

a-c Calibrations of the element position. **d** Calibrations of the system rotation axis. **a** The raw data from a point absorber without position correction. **b** The corrected raw data from a point absorber after calibration. **c** The radial deviation of the 1024 transducer elements. **d** The reconstructed image of the chain of absorbers is shown in different views. The ideal rotation axis and tracks are plotted in red, while the actual ones are in green. Through the calculation of analytical geometry, the actual axis is parallel to the ideal axis, which is only shifted 0.07 mm along the X-axis and 0.29 mm along the Y-axis, as labelled. **e** The reconstructed image of the leaf skeleton without calibrations. **f** The reconstructed image after the geometric calibrations. Two representative parts are zoomed in to observe the improvements.

Figure S5. The setup for optical attenuation coefficient estimation in fresh tissue.

Figure S9. Quantification of the STINT method's effectiveness.

a The single-pulse 3D image results of a complete leaf vein. **b** The results with the STINT method. **c-f** Profile comparison, where the single-pulse image is represented by dashed lines, and the results obtained using the STINT method are indicated by solid lines. Colors correspond to the arrow positions in **a**. FCC, feature correlation coefficient. STD, the standard deviation.

Figure S11. Quantification of respiratory motion effects in the STINT method.

a Comparison of whole-body imaging using single-pulse and STINT methods with and without respiratory gating. **b, c** Comparison of imaging details, with different regions corresponding to the boxes in **a**. RG, respiratory gating. SP, image sharpness.

Supplementary Tables

System name	Our 3D-PanoPACT	Caltech 3D-PACT ^{1,2}	Canon PAI-04 ³	Zurich FONT ⁴
Array type	$f_c = 3.16$ MHz Spherical array with 1024 elements (2π solid angle)	$f_c = 2.25$ MHz four arc-shaped arrays with 256×4 elements (2π solid angle)	$f_c = 4$ MHz Spherical array with ~ 500 elements (2π solid angle)	$f_c = 5$ MHz Spherical array with 512 elements ($\frac{7\pi}{9}$ solid angle)
Piezoelectric conversion mechanism	Piezocomposite materials	Not demonstrated	CMUT	Piezocomposite materials
Element Size	5 mm in diameter (19.6 mm^2)	0.6 mm \times 0.7 mm (0.42 mm^2)	1.5 mm in diameter (1.77 mm^2)	1-2.5 mm in diameter for different versions (0.79 - 4.9 mm^2)
Ratio of element size to the central wavelength (ROSW).	10.53	1.05	4	3.33-8.33
Array motion	Small-angle rotation with optional translation	Large-angle rotation	Spiral translational scanning	Not demonstrated
Geometric calibration	Applied	Applied	Not demonstrated	Not demonstrated
Image reconstruction	Dual-speed-of-sound UBP algorithm	Dual-speed-of-sound UBP algorithm	Dual-speed-of-sound UBP algorithm	UBP algorithm
Spatial resolution	250 μm (nearly isotropic)	370 μm (nearly isotropic)	270 μm (nearly isotropic)	150 μm (nearly isotropic)
In-vivo imaging depth	~ 20 mm	10 mm in rat brains 15 mm in rat liver 40 mm in human breasts	~ 10 mm in human breasts	7 mm in mouse brain
Imaging speed and well-resolved FOV	1) 25 Hz single-wavelength imaging, FOV = 6 cm^3 (depending on the laser rate) 2) 10 Hz dual-wavelength whole-brain imaging, FOV = 3.2 cm^3 (depending on the laser rate) 3) 0.5 Hz brain base functional imaging, FOV =	1) 0.5 Hz functional imaging, FOV = 2.2 cm^3 2) 10 s for human breast imaging, FOV = 50.2 cm^3	2 minutes for human breast imaging, FOV = 46.2 cm^3	25 Hz for mouse brain functional imaging, FOV = 0.033 cm^3

	3.2 cm ³ 4) 10Hz whole-trunk imaging, FOV = 70.5 cm ³			
Acquisition time per voxel per laser pulse	~11 ns	~ 4 μs	Not involved	~ 0.5 μs
Functional imaging	1) Hepatic artery mapping with a single wavelength 2) Hemodynamics during electrical stimulations on limbs 3) Hemodynamics during SNP administration in the Circle of Willis (first proposed) 4) Whole-trunk dynamics (first proposed) 5) High-spatiotemporal-resolution tracking of small molecule metabolic pathways at whole-body scale (first proposed)	Hemodynamics during 1) Hypoxic challenges 2) Deep to light anesthesia 3) Resting state 4) Electrical stimulations on limbs	S-factor (i.e., estimated oxygen saturation)	1) Hemodynamics 2) GCaMP6 responses during electrical stimulations on limbs
Small animal whole-brain imaging	 With a portion of the parietal bone removed (~ 8 mm in diameter) and the sutured scalp	 With thinned skull	Not demonstrated	 With intact scalp and skull
Small animal whole-trunk imaging	Structural imaging: 		Not demonstrated	Not demonstrated

	Functional imaging: 			
Human breast imaging	Not demonstrated		Not demonstrated	

Table S1. Performance comparison of 3D-PanoPACT with other state-of-the-art three-dimensional PACT systems based on spherical array configuration. Note that in the comparison of the metric “Acquisition time per voxel per laser pulse” (row 6), the voxel size of different systems is calculated using a quarter of the central wavelength corresponding to the transducer array, for a fair comparison.

----- Author Response Letter -----

We sincerely appreciate the constructive comments from all the reviewers. We believe that all raised concerns have been addressed through the performance of additional experiments, the incorporation of new data and figures, and the implementation of improvements based on the comments. Our point-by-point responses to the reviewers' comments are provided below. The original reviewers' comments are indicated in black, our responses are provided in blue, and the revisions are marked in red. A highlighted version of the manuscript with revisions marked with a yellow background is also provided in the submission.

Reviewer #1 (Remarks to the Author):

Overall, Wang and colleagues addressed our concerns, adding a lot of new and useful material. They clearly put a lot of effort and rigour into it, and they should be praised for that.

We would like though to highlight a few points:

- The so-called STINT method actually encompasses 3 different techniques that are used in quite different experimental contexts.

Upon our request, the authors now describe these three techniques in detail, making it much clearer for the readers. It appears though that this could be enough material for separate papers.

While the manuscript is now much clearer, we are uncertain whether it still aligns with the format of Nature Communications. As this is primarily a matter of editorial scope rather than scientific content, we defer the decision to the editor.

Reply:

We sincerely appreciate the reviewers' constructive feedback and kind recognition of our efforts in strengthening the manuscript.

In 3D-PanoPACT, the STINT method enabled large-field-of-view, high-spatiotemporal-resolution photoacoustic imaging through multiple tailored implementations addressing distinct experimental demands. Crucially, the STINT method is intrinsically coupled with our hardware design (for example, the transducer array configuration and the system control), with their synergistic integration conferring 3D-PanoPACT with exceptional imaging capability. We therefore maintain that the comprehensive description of the STINT method is essential to represent the systematic research paradigm underlying this work. Furthermore, the technical details provided may serve as a valuable implementation guide for practitioners in this field. Should any additional adjustments be required to better align with the formatting guidelines of *Nature Communications*, we are fully prepared to collaborate with the editorial team to refine the presentation accordingly.

- One of the STINT implementations refers to respiratory gating. Similar works have been published already, and proper references should be cited.

- Similarly, electrical impulse deconvolution has already been performed for PA imaging, and proper references should be cited as well.

Reply:

Thanks for the constructive suggestions. References corresponding to EIR testing and respiratory gating have been added to the revised manuscript.

Revisions:**References related to respiratory gating:**

62. Ron, A., Davoudi, N., Deán-Ben, X.L. & Razansky, D. Self-gated respiratory motion rejection for optoacoustic tomography. *Applied Sciences* 9, 2737 (2019).

63. Wei, J., Wang, Q., Song, X., Luo, Q. & Yang, X. Prospective respiration-gated photoacoustic microscopy. *IEEE Transactions on Biomedical Engineering* 67, 220-225 (2019).

References related to electrical impulse deconvolution:

30. Chowdhury, K.B., Bader, M., Dehner, C., Jüstel, D. & Ntziachristos, V. Individual transducer impulse response characterization method to improve image quality of array-based handheld optoacoustic tomography. *Optics Letters* 46, 1-4 (2021).

31. Chowdhury, K.B., Prakash, J., Karlas, A., Jüstel, D. & Ntziachristos, V. A synthetic total impulse response characterization method for correction of hand-held optoacoustic images. *IEEE transactions on medical imaging* 39, 3218-3230 (2020).

In conclusion, we do recommend for publication in Nature Communications, provided it fits the length standards.

Reply:

We are grateful for the reviewers' endorsement. We also remain fully committed to any additional editorial adjustments required to meet journal format standards.

Reviewer #2 (Remarks to the Author):

1. While 3D-PanoPACT demonstrates excellent temporal resolution and generally satisfactory tracking outcomes for small-molecule probes, its imaging resolution has significantly decreased overall, rather than the resolution degradation being limited to the edges as described in the results section. Although this resolution compromise might be an acceptable trade-off for rapid monitoring, its potential impact on result validity warrants further investigation. To strengthen this methodology, we recommend acquiring high-resolution data from specific metabolic organs at critical time points using smaller FOVs. If these targeted measurements align with the metabolic trends observed through rapid whole-body monitoring, it would compellingly demonstrate that controlled resolution reduction does not compromise monitoring accuracy. Additionally, providing a supplemental video showing the dynamic metabolic progression of molecular probes would significantly enhance data visualization and methodological transparency.

Reply:

Thank you sincerely for your thorough evaluation of our manuscript and your constructive feedback. 3D-PanoPACT achieves high-spatiotemporal-resolution imaging over a large field of view (FOV) by applying the spatiotemporal-integration (STINT) method. In addition, several critical components contribute to this achievement, including precision hardware design, calibration of transducer element positioning and rotation axis, accurate measurement of the electrical impulse response, and dual-speed-of-sound back-projection reconstruction. By appropriately compromising temporal resolution, the system achieves significantly improved spatial imaging resolution over a large FOV, which has been validated in phantom and whole-body animal tests (Fig. R1). We believe this is an essential prerequisite for whole-body cross-regional probe tracking.

Fig. R1 Spatial resolution improvement within large FOV using STINT method in 3D-PanoPACT.

To further support the methodology of this study and verify the validity of the results, we selected two specific organs—the heart (Fig. R2a) and liver (Fig. R2b)—for high-resolution single-organ reconstruction and compared the relative percentage enhancement of photoacoustic signals in these organs with the results of whole-body tracking results. The consistency between these two sets of results serves as an important validation of our approach. The reconstruction sound speed was fine-tuned for different organs to ensure optimal imaging quality within an FOV size of 12 mm. High-resolution images were reconstructed at four time points ($t = 30$ s, 120 s, 280 s, and 500 s). The mean percentage enhancement in organ signal was compared with the fitted curve of corresponding organ changes in the whole-body probe tracking results. It can be seen that these targeted measurement results are in good agreement with the metabolic trends observed through whole-body monitoring (Fig. R2c and d). The largest deviation between a data point and the fitted curve is 8.26% (at $t = 500$ s in the heart). This can strongly demonstrate that 3D-PanoPACT has high accuracy in whole-body metabolic dynamic monitoring and holds practical value in biomedical research. We have supplemented these contents in the Results section and revised Fig. 6.

Fig. R2 Single-organ high-resolution reconstruction and analysis results. **a** High-resolution reconstructions of the heart at representative time points ($t = 30$, 120 , 280 , 500 s), with pseudo-colored overlay indicating relative signal enhancement ($t = 0$ s referenced). HT, heart. **b** Reconstruction results of the liver. LV, liver. **c** Comparison of heart high-resolution data at specific time points with the fitted curve of corresponding organ signal changes in whole-body probe tracking results. **d** Comparison of liver data.

To further refine the work on A1094 probe metabolic tracking, we introduced phantom experiments before the *in-vivo* tests. A1094 solutions were prepared with a concentration gradient ($50 \mu\text{M}$, $60 \mu\text{M}$, $70 \mu\text{M}$, and $80 \mu\text{M}$) and sequentially injected into Teflon tubes for imaging, with an FOV of 60 mm (Fig. R3a). Analysis at different regions revealed that, due to factors such as non-uniform light illumination and spatial impulse response, the pixel values could not accurately reflect the probe concentration (Fig. R3b). However, the relative percentage change remained almost constant regardless of the regions selected, demonstrating high accuracy (Fig. R3c). Therefore, calculating the relative changes can reflect the metabolic patterns of the probe. We applied this method in our *in vivo* tests. We believe that the inclusion of this part makes the research on A1094 probe tracking more systematic and theoretically grounded. We have added the above content to the Results section and supplemented Fig. S9.

Fig. R3 Validation of A1094 probe concentration tracking using relative change.

a Imaging results of serial dilutions of A1094 in the tube. **b** Image pixel values acquired from regions of interest (colored boxes in **a**) in different concentrations. Data markers are color-coded to correspond with their respective regions. **c** Relative signal percentage change with 50 μM as reference baseline.

As recommended, we have now included a supplementary movie (Movie S6) demonstrating the dynamic metabolic progression of the molecular probes. This video provides a comprehensive visualization of the whole-body metabolic dynamics observed in our study, further improving methodological transparency (Fig. R4). The movie has been uploaded alongside the revised manuscript, and a brief description has been added to the Supplementary Materials. We believe this addition strengthens the accessibility and interpretability of our data.

Fig. R4 The screenshot of the newly added Movie S6 (See the attachment for details).

Revisions:

Results section:

Lines 432-438:

“To validate the A1094 probe tracking methodology, we first conducted phantom experiments prior to *in-vivo* test. Serial dilutions of A1094 (50, 60, 70, and 80 μM) were infused into Teflon tubes and imaged under a 60 mm FOV (Fig. S9a). Regional analysis revealed that while absolute pixel values

failed to linearly correlate with probe concentrations due to non-uniform light illumination and spatial impulse response variations (Fig. S9b), the relative percentage changes remained consistent across the regions (relative standard deviation < 5%) (Fig. S9c), confirming that relative changes could reliably reflect metabolic patterns, a principle subsequently applied to *in-vivo* studies.”

Lines 464-472:

“In Movie S6, the dynamic cross-regional tracking of A1094 across the whole body was visually demonstrated using 3D-PanoPACT. To validate the methodological robustness of our approach, high-resolution single-organ reconstruction (FOV = 12 mm) were performed of two organs - the heart (Fig. 6i) and liver (Fig. 6j). Organ-specific sound speeds were fine-tuned to ensure imaging fidelity. PA relative enhancement was quantified at four critical time points (t = 30, 120, 280, and 500 s) and compared with whole-body tracking results. The high-resolution organ measurements showed strong agreement with whole-body metabolic trends (maximum deviation: 8.26% at t = 500 s in heart), confirming the accuracy of 3D-PanoPACT for whole-body metabolic monitoring (Fig. 6k and l).”

Figures:

Fig. 6 (please refer to the end of this file)

Supplementary Materials:

Fig. S9 (please refer to the end of this file)

Movie S6: Dynamics of high-spatiotemporal-resolution tracking of small molecule A1094 metabolic pathways at whole-body scale (please refer to the attachments in this submission)

2. To better highlight this technique’s breakthroughs in whole-body metabolic monitoring and disease research, citing relevant prior studies in photoacoustics is recommended:

Radiology, 2021, 300(1): 89-97.

Reply:

We sincerely appreciate the reviewer’s insightful suggestion to further contextualize our technique’s advancements. In the revised manuscript, we have now cited the recommended study (Radiology, 2021, 300(1), 89-97) in the Introduction section to highlight 3D-PanoPACT’s breakthroughs in monitoring whole-body cross-regional metabolic dynamics. Thank you for the comments in improving the scholarly rigor of our manuscript.

Revisions:

Introduction section:

Lines 67-68:

“For example, existing studies have already been able to achieve dynamic assessment of metabolic functions in specific organs using 2D-PACT²⁰.”

References:

20. Lv, J., Xu, Y., Xu, L. & Nie, L. Quantitative functional evaluation of liver fibrosis in mice with dynamic contrast-enhanced photoacoustic imaging. Radiology 300, 89-97 (2021).

Reviewer #3 (Remarks to the Author):

Reply:

Thank you for your remarks and for the opportunity to participate in this review process.

Figure Revisions

Figure 6. High-spatiotemporal-resolution tracking of small molecules at whole-body scale using 3D-PanoPACT (Movie S6).

a A schematic of the experimental setup and imaging strategy. **b** High-quality whole-body 3D imaging acquired within 5 s under this imaging strategy. **c** Depth-encoded results. **d** Comparison of the absorption spectra of A1094 with haemoglobin (HbO₂) and deoxygenated haemoglobin (Hb). Imaging wavelength: 1064 nm. **e-g** Relative enhancement ($t = 0$ s referenced) in signal intensity

across different organs following A1094 administration. Raw data are shown in grayscale, while fitted curves are depicted in colors. IA, iliac artery; HT, heart; LV, liver; MA, mesenteric artery; IN, intestines; UT, ureter; BD, bladder. **h** Mapping of whole-body cross-regional enhancement at 0 s, 10 s, 60 s, and 500 s post-A1094 administration, revealing molecular metabolic pathways. **i** High-resolution reconstructions of the heart at representative time points ($t = 30, 120, 280, 500$ s), with pseudo-colored overlay indicating relative signal enhancement ($t = 0$ s referenced). **j** Reconstruction results of the liver. **k** Comparison of heart high-resolution data at specific time points with the fitted curve of corresponding organ signal changes in whole-body probe tracking results. **l** Comparison of liver data.

Supplementary Materials Revisions

Figure S9. Validation of A1094 probe concentration tracking using relative change.

a Imaging results of serial dilutions of A1094 in the tube. **b** Image pixel values acquired from regions of interest (colored boxes in a) in different concentrations. Data markers are color-coded to correspond with their respective regions. **c** Relative signal percentage change with 50 μM as reference baseline.